



# Assessing the lifetime of anthropogenic $CO_2$ and its sensitivity to different carbon cycle processes

Christine Kaufhold[1,2], Matteo Willeit[1], Bo Liu[3,4], and Andrey Ganopolski[1]

[1]Department of Earth System Analysis, Potsdam Institute for Climate Impact Research (PIK), Member of the Leibniz Association, P.O. Box 601203, D-14412 Potsdam, Germany
[2]Institute of Physics and Astronomy, Universität Potsdam, Potsdam, Germany
[3]Universität Hamburg, Hamburg, Germany
[4]Max Planck Institute for Meteorology, Hamburg, Germany

**Correspondence:** Christine Kaufhold (kaufhold@pik-potsdam.de)

**Abstract.** Although it is well-established that anthropogenic $CO_2$ emitted into the atmosphere will persist for a long time, the duration of the anthropogenic climate perturbation will depend on how rapidly the excess $CO_2$ is removed from the climate system by different biogeochemical processes. The uncertainty around the long-term climate evolution is therefore not only linked to the future of anthropogenic $CO_2$ emissions, but to our insufficient understanding of the long-term carbon cycle. Here, we use the fast Earth system model CLIMBER-X, which features a comprehensive carbon cycle, to examine the lifetime of anthropogenic $CO_2$ and its effects on the long-term evolution of atmospheric $CO_2$ concentration. This is done through an ensemble of 100,000 year long simulations, each driven by idealized $CO_2$ emission pulses. Our findings indicate that, depending on the magnitude of the emission, 75% of anthropogenic $CO_2$ is removed within 197–1,820 years after emissions end. Approximately 4.3% of anthropogenic $CO_2$ will remain beyond 100 kyr. We find that the uptake of carbon by land, which has only been marginally considered in previous studies, has a significant long-term effect, storing approximately 4–13% of anthropogenic carbon by the end of the simulation. For the first time, we have quantified the effect of dynamically changing methane concentrations on the long-term carbon cycle, showing that its effects are likely negligible over long timescales. The timescale of carbon removal via silicate weathering is also reassessed here, providing an estimate (80–105 kyrs) that is significantly shorter than some previous studies due to higher climate sensitivity, stronger weathering feedbacks, and the use of a spatially explicit weathering scheme, leading to a faster removal of anthropogenic $CO_2$ in the long-term. Our study highlights the importance of adding model complexity to the global carbon cycle in Earth system models, as to accurately represent the long-term future evolution of atmospheric $CO_2$.

## 1 Introduction

A large amount of research has been dedicated to studying the impact of anthropogenic $CO_2$ emissions on the climate, what oceanographer Roger Revelle coined as "man's great geophysical experiment" (Revelle, 1956; Revelle et al., 1965). The scope of such research is usually limited to centennial timescales due to their relevance for governance and policy. However, there is a growing societal need and an increasing number of scientific inquiries on the long-term future. This particularly concerns the



site selection for nuclear waste disposal and post-closure safety assessments of deep geological repositories, as a number of environmental factors, such as subterranean stress, permafrost, erosion, and subrosion, can potentially compromise long-term
safety (Näslund et al., 2013; Lord et al., 2015a; Lindborg et al., 2018; Turner et al., 2023; Kurgyis et al., 2024).

Previous studies on long-term future climate evolution have primarily focused on one of two, tightly related research areas. Some, for example, have investigated the length of the current interglacial under natural conditions: Berger and Loutre (2002) showed that we were already positioned to experience an unusually prolonged interglacial period as we currently approach a minima in the 100-kyr and 400-kyr eccentricity cycle. A consensus also remains to be seen regarding the timing and duration
of the next glacial cycle; model realizations which satisfy paleoclimatic constraints in Talento and Ganopolski (2021) suggest that, under non-anthropogenic conditions, the next full glacial was expected to occur in approximately 90 kyr to around 150 kyr. A second area of interest is the lifetime of anthropogenic $CO_2$ in the atmosphere[1]: Archer et al. (1997) and Archer et al. (1998) improved previous estimates by including $CaCO_3$ dissolution kinetics, and showed that the return of atmospheric $CO_2$ concentration to pre-industrial conditions would take tens of thousands of years. These two overarching areas of focus
converged in Archer (2005) and Archer and Ganopolski (2005), where they showed that, (1) not only is anthropogenic $CO_2$ expected to survive in our atmosphere for several hundreds of thousands of years, but (2) it could also significantly delay the next glacial period. This was later confirmed by other studies (Ganopolski et al., 2016; Talento and Ganopolski, 2021).

In the past, general circulation models (GCMs) and intermediate complexity models (EMICs) have been used to study the 1000–10,000 year carbon cycle response to anthropogenic $CO_2$ emissions (see: Archer et al., 2009a; Joos et al., 2013). Mod-
elling studies become more scarce when going even further forward in time due to the rising computational cost, and as such, experiments are based on either EMICs (Ridgwell and Hargreaves, 2007; Charbit et al., 2008; Eby et al., 2009; Shaffer, 2010; Goodwin and Ridgwell, 2010; Meissner et al., 2012; Ganopolski et al., 2016; Brault et al., 2017; Jeltsch-Thömmes and Joos, 2020; Duque-Villegas et al., 2022), or models of lower complexity (Sundquist, 1991; Walker and Kasting, 1992; Lenton and Britton, 2006; Tyrrell et al., 2007a; Uchikawa and Zeebe, 2008; Cawley, 2011; Herrero et al., 2013; Talento and Ganopolski,
2021; Couplet et al., 2024). The results of these studies largely agree in some aspects. For example, it has been established that the magnitude of cumulative emissions will predominantly control the long-term response, rather than the emission pathway (Eby et al., 2009; Matthews et al., 2009; Zickfeld et al., 2012; Herrington and Zickfeld, 2014). Contemporary predictions for deep future $CO_2$ concentrations exhibit considerable diversity, however. For illustration, Fig. 1 provides a graphical overview on the current estimates of atmospheric $CO_2$ concentration 10,000 years after present for different cumulative $CO_2$ emissions.

Quantifying the atmospheric lifetime of anthropogenic $CO_2$ is challenging because it is controlled by a number of biochemical and geological mechanisms operating on different timescales up to hundreds of thousands of years. Modelling studies suggest that the terrestrial biosphere will initially uptake anthropogenic $CO_2$ (owing to enhanced vegetation productivity from $CO_2$ fertilization), although increased cellular respiration, soil carbon decomposition, land-use change, and wildfires could eventually turn the land into a carbon source in a warming world. Extensive fossil fuel emissions can therefore overpower the
capacity of conventional land carbon reservoirs to absorb anthropogenic $CO_2$, and only the ocean remains the $CO_2$ sink on long

---

[1]Here, we follow terminology introduced by Archer and Brovkin (2008), where "lifetime of anthropogenic $CO_2$" is understood to be the lifetime of an anomaly in atmospheric $CO_2$ concentration caused by anthropogenic $CO_2$ emissions.



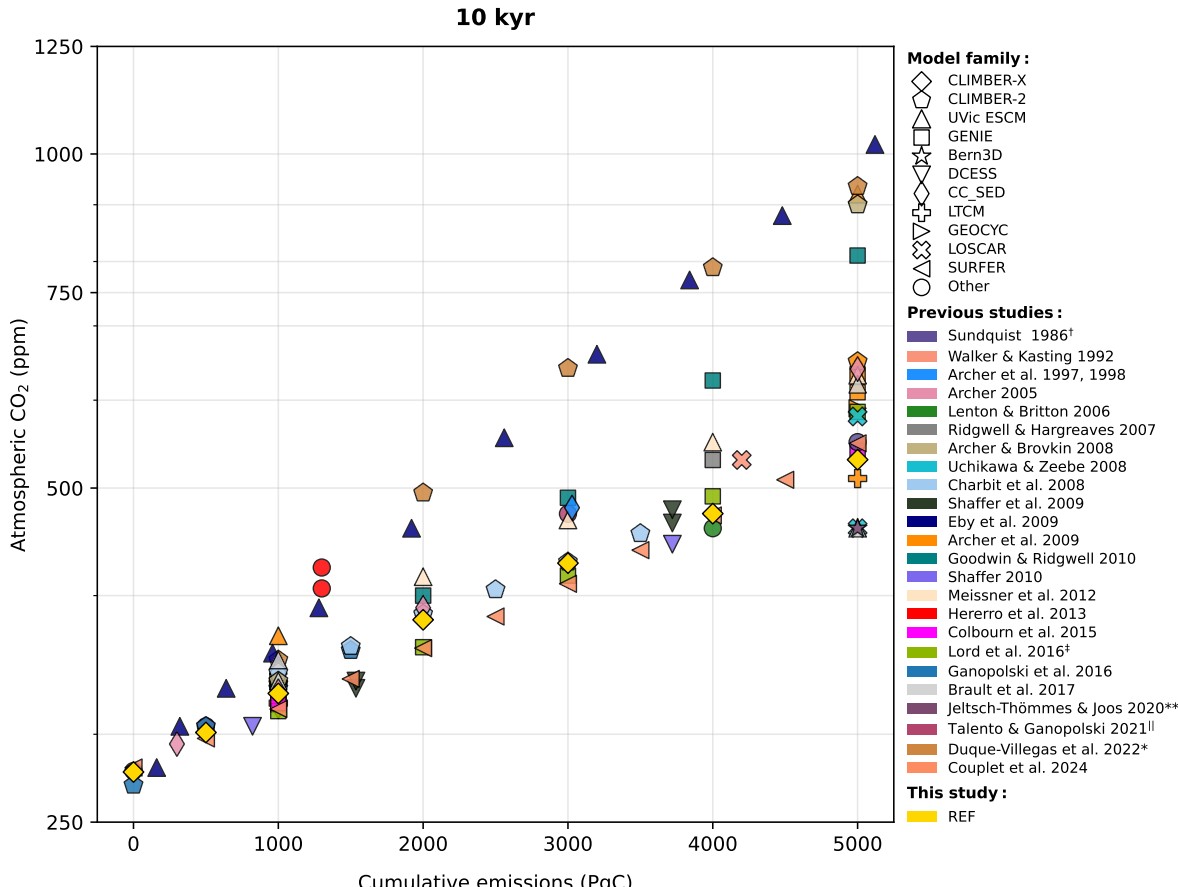

**Figure 1. Estimates on the impact of different cumulative emissions on atmospheric** $CO_2$ **concentration 10** kyr **after the start of the simulation.** Experiments shorter than 10,000 years, or those which considered emissions of 6,000 PgC or larger (i.e., unconventional fossil fuel resources), were omitted. Different experiments were chosen from these publications to highlight that the large diversity is primarily due to how long-term carbon cycle processes are resolved. The reference experiment in this study (yellow) has been plotted for comparison. The data was acquired through visual inspection of graphs, and as a result, small errors may be present. Further information can be found in the respective publications.[†]Data for Sundquist (1991) taken from Tyrrell et al. (2007a). [‡]Data generated from the emulator based on cGENIE results. [**]Data calculated from fraction of emissions remaining. [ǁ]Best solution from the model ensemble. [*]Data calculated from formula provided in the appendix.



timescales through processes such as air–sea $CO_2$ exchange, ocean invasion, seafloor $CaCO_3$ reactions and carbonate accumulation. The substantial influence that marine biogeochemistry has on the long-term carbon cycle has also been recognized by many studies, particularly regarding the reaction between dissolved anthropogenic $CO_2$ and calcium carbonate in deep ocean sediments (Archer et al., 1997; Middelburg et al., 2020). On even longer timescales, carbon exchange with geological reservoirs

becomes important through processes such as sediment burial, chemical weathering of rocks on land, and volcanic degassing. As a consequence of all of these processes, carbon cycle models predict a highly nonlinear removal of anthropogenic $CO_2$ over time. The evolution of anthropogenic $CO_2$ can therefore be understood as a superposition of exponential decays (Maier-Reimer and Hasselmann, 1987; Archer et al., 1997; Archer and Brovkin, 2008; Colbourn et al., 2015; Lord et al., 2015b), with each function representing a different process in the carbon cycle that takes up carbon. This generally produces a "long tail" of

anthropogenic $CO_2$ concentration which has been shown to persist for hundreds of thousands of years.

Due to the complexity and variety of poorly understood processes, the long-term future climate evolution remains highly uncertain even without considering the unpredictability of future anthropogenic emissions (which can be reconciled to some extent using an ensemble of emission scenarios). Given this complexity, it becomes essential to make assumptions based on the timescales involved, the specific research question being addressed, and the resolution of the model employed. Previous studies

have often made significant simplifications to (e.g., Lenton and Britton, 2006), or completely neglected (e.g., Archer et al., 1997; Lord et al., 2015b) certain aspects related to land carbon. Weathering processes have also been frequently simplified (e.g., Charbit et al., 2008; Uchikawa and Zeebe, 2008), or their feedbacks not considered (e.g., Montenegro et al., 2007; Ridgwell and Hargreaves, 2007; Eby et al., 2009). As computational expense is the biggest factor inhibiting high-resolution modelling on longer timescales, these studies also did not consider the interaction between $CO_2$ and ice sheets, methane deposits, and the

effect of future glacial cycles (Archer, 2005).

Here, we provide a set of idealized long-term transient climate change scenarios for the next 100,000 years using the fast Earth system model CLIMBER-X with a comprehensive carbon cycle model. These experiments seek to quantify the degree of uncertainty of the "anthropogenic factor" by employing a wide range of idealized emission scenarios to evaluate the spread in possible climate response. Considering the ongoing and substantial disagreement of future $CO_2$ evolution among different

models, we provide an analysis of the sensitivity of the model response to climate sensitivity and different carbon cycle processes. This is a first step towards a fully-coupled simulation of the next 100 kyr with CLIMBER-X that includes interactive ice sheets. The paper is structured as follows: we first describe our experimental set-up, model, and considerations (Section 2). The results are then contextualized in relation to other studies by analyzing the atmospheric lifetime and subsequent removal timescales of different carbon cycle processes (Section 3). This is followed by a corresponding sensitivity study (Section 4).

We conclude with a brief discussion, summary of our findings and and an outlook on further investigations (Section 5).



## 2 Methods

### 2.1 Model description

We use CLIMBER-X v1.1 (Willeit et al., 2022, 2023), a fast Earth system model designed to simulate the climate evolution on time scales ranging from decades to >100,000 years. CLIMBER-X includes the statistical-dynamical semi-empirical
atmosphere model SESAM (Willeit et al., 2022), the 3D frictional-geostrophic ocean model GOLDSTEIN (Edwards et al., 1998; Edwards and Marsh, 2005), the dynamic-thermodynamic sea ice model SISIM (Willeit et al., 2022), the land surface and dynamic vegetation model PALADYN (Willeit and Ganopolski, 2016) and the ocean biogeochemistry and marine sediments model HAMOCC (Heinze and Maier-Reimer, 1999; Ilyina et al., 2013; Mauritsen et al., 2019). CLIMBER-X operates at a horizontal resolution of $5° \times 5°$. It is computationally efficient and can simulate up to 10,000 years per day, allowing it to
perform a large ensemble of long-term simulations. This is also possible because short term processes (e.g., weather, diurnal cycle) are not resolved. CLIMBER-X can generally well represent historical changes in climate (Willeit et al., 2022) and the carbon cycle (Willeit et al., 2023), and is suited to provide credible simulations for very long timescales into the deep future. Some studies using CLIMBER-X to transiently simulate the next several thousand years with a focus on the stability of the Greenland ice sheet have already been performed (e.g., Höning et al., 2023; Höning et al., 2024).

### 100 2.2 Open carbon cycle

The CLIMBER-X carbon cycle model has already been described in detail (Willeit et al., 2023), and has been shown to effectively represent the cycling of carbon though the atmosphere, vegetation, soils, permafrost, seawater, and marine sediments. For our experiments, we run the simulations using the so-called "open carbon cycle set-up" (Willeit et al., 2023), with interactive marine sediments and weathering on land and the resulting geological sources and sinks of carbon. Carbon is not conserved
in this setup; it is removed from the system through sediment burial and introduced to the system via weathering and volcanic outgassing. However, a conservation of phosphate and silicate in the ocean-sediment system was additionally enforced. This effectively "closes" the silicate and phosphate cycles, which would otherwise complicate the analysis and interpretation of model results. The globally uniform atmospheric $CO_2$ concentration in CLIMBER-X is computed interactively using source and sink terms in the following prognostic equation for the total carbon content stored as atmospheric $CO_2$ ($C_{atm}$):

$$\frac{dC_{atm}}{dt} = F_{lnd} + F_{ocn} + F_{anth} + F_{volc} - F_{weath} \qquad (1)$$

where $F_{lnd}$ is the global net land-to-atmosphere carbon flux (PgC $yr^{-1}$), $F_{ocn}$ is the net sea-air carbon flux (PgC $yr^{-1}$), $F_{anth}$ are anthropogenic carbon emissions (PgC $yr^{-1}$), $F_{volc}$ is the volcanic outgassing flux (PgC $yr^{-1}$), and $F_{weath}$ is the $CO_2$
consumption by carbonate and silicate weathering (PgC $yr^{-1}$) (Willeit et al., 2023). PALADYN includes a rock weathering scheme influenced by runoff and temperature (Hartmann, 2009a; Börker et al., 2020), accounting for 16 different lithologies



as described in Hartmann and Moosdorf (2012). The weathering module computes dissolved inorganic carbon (DIC) and alkalinity fluxes to the ocean based on the release of bicarbonate ions ($HCO_3^-$) into rivers. Spatially explicit carbonate and silicate weathering fluxes take the form:

$$F_{weath}^{carb} = F_{weath,0}^{carb} \sum_{\substack{i=1 \\ lithology}}^{14} e^{\frac{1000\, E_{a,carb}}{R(T \cdot T_0)}(T-T_0)} \times b(i)\beta(i)\alpha(i)R_{off}, \tag{2}$$

$$F_{weath}^{sil} = F_{weath,0}^{sil} \sum_{\substack{i=1 \\ lithology}}^{14} e^{\frac{1000\, E_{a,sil(i)}}{R(T \cdot T_0)}(T-T_0)} \times b(i)\beta(i)(1-\alpha(i))R_{off}, \tag{3}$$

for every lithology except for carbonate sedimentary rocks and loess, which are instead described by the runoff-dependent weathering rates from Amiotte Suchet et al. (2003) and Börker et al. (2020), respectively. In Equations 2 and 3, T is the annual mean near-surface air temperature (K), $T_0$ is 284.2 K, $R_{off}$ is the annual runoff ($kg\,m^{-2}\,yr^{-1}$), b is molality/weathering rate ($molC\,kg^{-1}$ water), $E_{a,carb}$ is the activation energy of carbonates ($14\,kJ\,mol^{-1}$), $E_{a,sil}$ is the activation energy of silicates ($kJ\,mol^{-1}$), R is the molar gas constant ($8.3145\,J\,mol^{-1}\,K^{-1}$), $\alpha$ is the fraction to weather as carbonate rocks, and $\beta$ is the
fraction of a given lithology i in a grid cell. The values for the parameters $\alpha$, b, and $E_a$ are all based on the calibrated run-off and temperature dependent models from Hartmann (2009a) and Hartmann et al. (2014).

Although the Earth experiences a number of external forces, this study generally excludes factors like impact events, changes in tectonic configuration and solar luminosity, as they are either unpredictable or operate on such long timescales that they are not relevant for our investigation. Similarly, we do not resolve the "deep" carbon cycle (i.e., carbon subduction and recycling
in the Earth's mantle) as their effects only become important on timescales longer than our experiment duration (i.e., $10^6 - 10^7$ years), and involves processes that cannot be resolved with Earth system models. As such, we assume that the pre-industrial carbon cycle was in equilibrium, which implies that the constant volcanic outgassing is prescribed and set to half the global silicate weathering rate at the pre-industrial time (0.0738 PgC $yr^{-1}$ or 6.15 TmolC $yr^{-1}$). This condition ensures that the atmospheric $CO_2$ is in equilibrium under pre-industrial conditions (Munhoven and François, 1994; Willeit et al., 2023), which
has been a common assumption in previous investigations which looked at the long-term forced response of the climate (e.g., Colbourn et al., 2015; Lord et al., 2015b; Brault et al., 2017).

## 2.3 Experimental set-up

Model simulations are started from a pre-industrial equilibrium state which has been obtained from a 100,000 year equilibrium spin-up of the carbon cycle model as described in Willeit et al. (2023). To simplify interpretation and ensure comparability with
previous studies, orbital forcing was fixed at present-day values, with the combined effects of anthropogenic and orbital forcing to be explored in a future study. The pre-industrial $CO_2$ concentration is taken to be 280 ppm, whereas the orbital parameter



**Table 1. Overview of the experimental configuration for the simulations performed in this study.** For each experimental configuration, all emission forcings (0 PgC-5000 PgC) were applied to produce an ensemble.

| Experiment Name | Emission Pathway | Land Carbon | Terrestrial Weathering | Climate Sensitivity | Methane |
|---|---|---|---|---|---|
| Reference experiment | | | | | |
| REF | Control | On | Variable | 3.1°C | Constant |
| Emission pathway experiments | | | | | |
| PATH1 | Config 1 | On | Variable | 3.1°C | Constant |
| PATH2 | Config 2 | On | Variable | 3.1°C | Constant |
| Sensitivity experiments | | | | | |
| noLAND | Control | Off | Variable | 3.1°C | Constant |
| noWEATH | Control | On | Constant | 3.1°C | Constant |
| ECS2 | Control | On | Variable | 2°C | Constant |
| ECS4 | Control | On | Variable | 4°C | Constant |
| intCH4 | Control | On | Variable | 3.1°C | Interactive |

eccentricity e is 0.0167, obliquity ε is 23.46°, and the perihelion ω is 100.33°. This corresponds to a maximum summer insolation at 65°N of 480.4 $Wm^{-2}$. In the reference experiment, we set a constant methane concentration of 600 ppb, which is representative for Holocene conditions (Sapart et al., 2012; Mitchell et al., 2013; Beck et al., 2018). Although CLIMBER-X is

capable of simulating ice sheets (e.g., Höning et al., 2023; Talento et al., 2024; Willeit et al., 2024), we conduct our simulations with prescribed present-day Greenland and Antarctica to ensure comparability with earlier studies. Interactive ice sheets will be enabled in a follow-up study. All simulations run for 100,000 years with constant orbital parameters and without any climate acceleration technique.

To investigate the lifetime of anthropogenic $CO_2$ and the effect it has on the long-term evolution of the climate, we introduce

idealized $CO_2$ emission pulses representing cumulative emissions from 0 PgC to 5000 PgC (Fig. E1) to cover a variety of different anthropogenic scenarios. The upper limit of ~5000 PgC has been used in many previous studies (e.g., Archer, 2005; Montenegro et al., 2007; Uchikawa and Zeebe, 2008; Archer et al., 2009a) as it just surpasses the current estimated maximum of conventional fossil fuel reserves (Lal, 2008; McGlade and Ekins, 2015). To simulate a variety of emissions pathways and explore the sensitivity of the results to the duration of the $CO_2$ emission pulse, we designed three different Gaussian functions

with different shapes (with an increasing mean and standard deviation with higher cumulative $CO_2$, Fig. E1). Following the emission pulse, emissions are set to 0 PgC $yr^{-1}$ for the rest of the simulation.

A number of additional experiments were performed to isolate the effect of different carbon cycle processes on the long-term fate of anthropogenic $CO_2$. Specifically, we explored the role of the land carbon cycle response, the impact of enabling interactive methane concentrations in the atmosphere, and the impact of the emissions pathway on the lifetime of anthropogenic

$CO_2$. To isolate the effect of the weathering feedback on the $CO_2$ lifetime, a dedicated set of experiments was conducted





in which weathering fluxes were enforced to remain constant at their pre-industrial rate throughout the simulation (0.2376 PgC yr$^{-1}$ or 19.8 TmolC yr$^{-1}$ for carbonate weathering; 0.1476 PgC yr$^{-1}$ or 12.3 TmolC yr$^{-1}$ for silicate weathering). The sensitivity of the results to different equilibrium climate sensitivities between 2-4 °C was additionally investigated, with the procedure for obtaining these sensitivities described in Appendix A. A summary of all experimental configurations is presented in Table 1.

## 3 Results

### 3.1 Long-term carbon cycle evolution

An overview of the 100 kyr response in atmospheric $CO_2$ concentration and global mean temperature across all emission scenarios in the reference (REF) experiment can be seen in Fig. 2. The initialization of our experimental set-up is demonstrably robust, evidenced by the less than 2 ppm drift (from the pre-industrial 280 ppm) of atmospheric $CO_2$ concentration in the zero emissions scenario over the span of 100 kyr (Fig. 2a). All simulations with anthropogenic emissions show a rapid increase in atmospheric $CO_2$ concentration during the ramp up of emissions, with peak concentration occurring within the first 200 years. In this reference run, peak $CO_2$ concentrations are approximately 360 and 1820 ppm for 500 PgC and 5000 PgC scenarios respectively (Fig. 2a). These values are within the range of magnitude reported by previous studies (e.g., Montenegro et al., 2007; Colbourn et al., 2015). We find that the increase in global mean surface temperature lags that of $CO_2$ concentration (Fig. 2b), and that the size of the temporal lag tends to increase as cumulative emission grows (Ricke and Caldeira, 2014; Zickfeld and Herrington, 2015). Peak temperature anomalies of approximately 0.6 and 6.4 °C are seen for the 500 PgC and 5000 PgC scenarios in the REF ensemble. After 300 years from the start of the experiments, both atmospheric $CO_2$ concentration and global temperatures decrease across the different emission scenarios as the atmospheric burden starts to diminish and carbon is taken up by the land and ocean pools. The only exception to this is the 5000 PgC scenario, where temperatures temporarily stabilize instead of decreasing due to the release of soil carbon into the atmosphere (Fig. 2a-c). Temperatures slowly recover to pre-industrial levels, and after 10 kyr, global warming is reduced by more than half across the different emission scenarios (Fig. 2b). After 100 kyr, atmospheric $CO_2$ concentration still remains around 10 and 100 ppm above pre-industrial levels in the 500 and 5000 PgC scenarios, and temperatures are approximately 0.1 and 1.4 °C larger than the pre-industrial in these scenarios at the end of the simulation.

The fate of the anthropogenic $CO_2$ emissions and how this carbon is distributed among the different Earth system components after the peak atmospheric $CO_2$ concentration is shown in Fig. 4. The land and ocean both rapidly absorb $CO_2$ during the ramp-up of emissions through enhanced vegetation productivity, air–sea $CO_2$ exchange and its subsequent dissolution (Fig. 3). At peak, $CO_2$ concentration, the land and ocean are responsible for between 34–61% of carbon removal (19–39% by land, 15–22% by the ocean, Fig. 4), with the carbon removal fractions decreasing with increasing cumulative emission. During the first 1 kyr after peak $CO_2$ concentrations, oceanic $CO_2$ uptake gradually slows as surface waters approach equilibrium with the atmosphere (Fig. 3b). Although ocean carbon fraction largely increases over the first 1 kyr after peak $CO_2$ concentrations (accounting for 46–59% 1 kyr after emissions cease, Fig. 4), the carbon fraction taken up by the ocean during the latter half

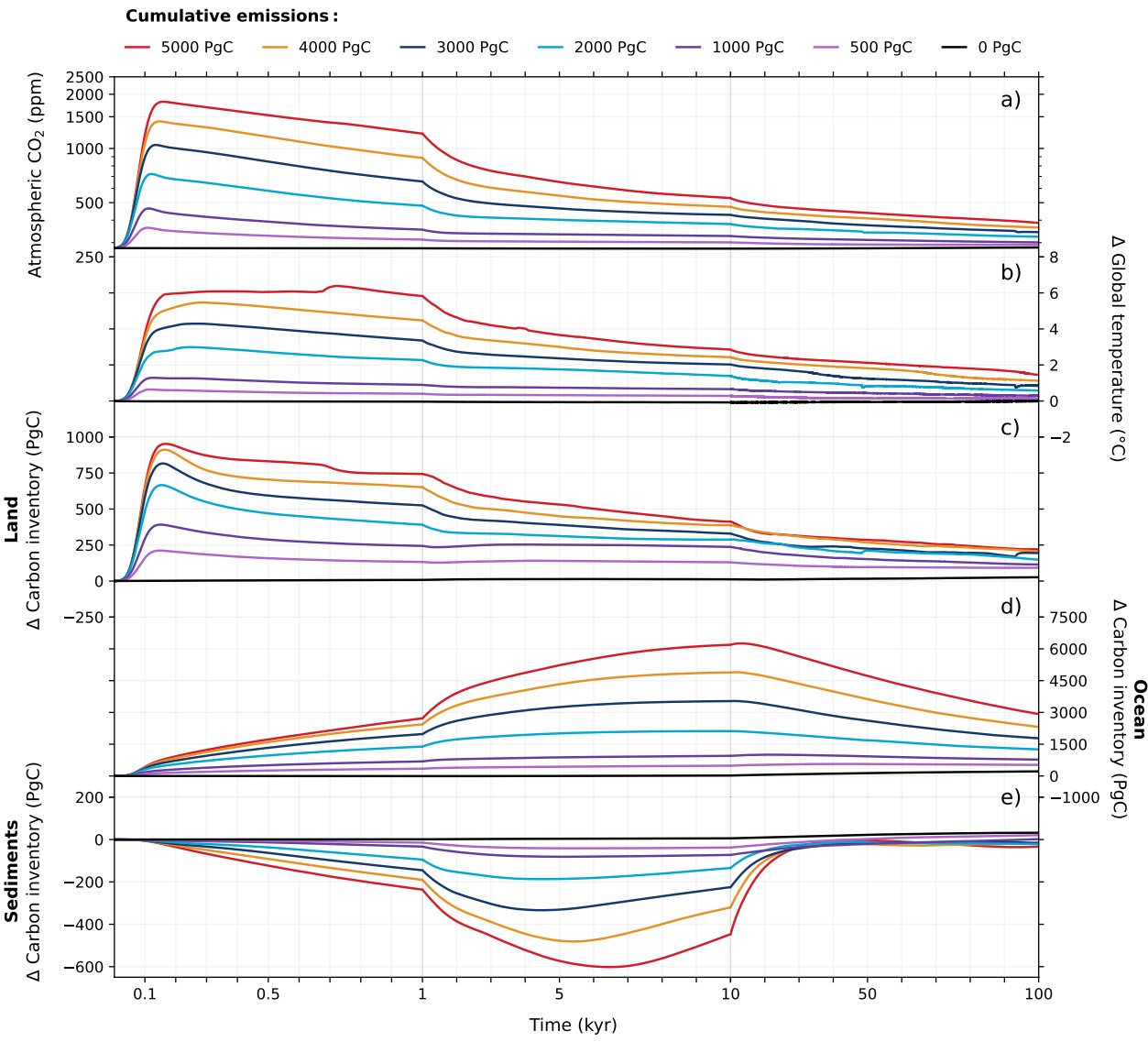

**Figure 2. Response in (a) atmospheric CO$_2$ concentration, (b) global surface air temperature, (c) land carbon inventory, (d) ocean carbon inventory, and (e) sediment carbon inventory to the full ensemble of emission pulses (0-5000 PgC) in the REF experiment for 100 kyr.**





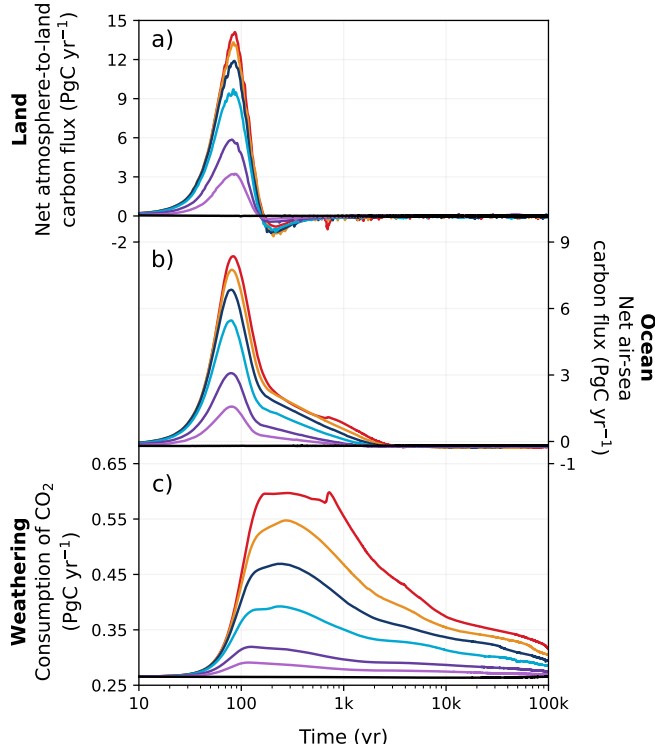

**Figure 3. Changes in (a) net atmosphere-to-land carbon flux, (b) net air–sea carbon flux, and (c) weathering consumption of** $CO_2$ **in different emission scenarios over 100** kyr**.** Colours correspond to the cumulative emission scenarios shown in Fig. 2.

of the millennium appears disproportionately larger than what is taken up during the first 500 years, as land carbon already begins decreasing by this time (Fig. 2c, Fig. 3a). Beyond the first millennium, geological processes (e.g., marine sediments and weathering) largely control the evolution of atmospheric $CO_2$ concentration (Archer and Brovkin, 2008). Silicate weathering is responsible for 11–17% of carbon removal 10 kyr after peak $CO_2$ concentration. However, by 100 kyr, it is responsible for the majority of carbon removal (62–78%), and becomes the dominant removal process on the timescale of hundreds of thousands of years.

### 3.1.1 Land carbon response

Two opposing mechanisms involving vegetation and soil carbon predominantly govern the land carbon response to climate change. The land carbon pool is initially a carbon sink due to increases in vegetation carbon (Brovkin et al., 2013), as $CO_2$-driven fertilization enhances net primary productivity. These increases, however, can be partially or totally offset by warming-enhanced soil respiration, which strongly depends on model parameters, whether permafrost, peatlands, and wetlands are included, and how they are modelled in the carbon cycle model (Zickfeld et al., 2013; Eby et al., 2013). In our simulations, the peak increase in vegetation carbon significantly exceeds and more than compensates for the maximum loss in soil carbon (Fig.





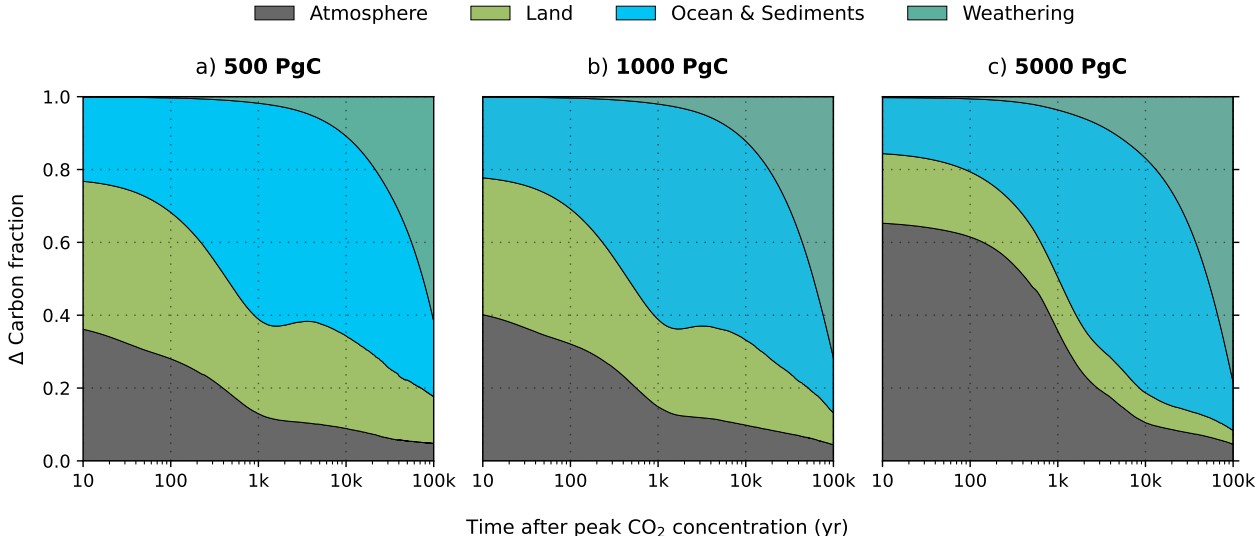

**Figure 4. The relative partition of anthropogenic carbon in the REF experiment after peak atmospheric** $CO_2$ **concentration.** The partition of the atmosphere (grey), land (green), ocean & sediments (blue) and silicate weathering (teal) is shown for the (a) 500, (b) 1000, and (c) 5000 PgC scenarios. These calculation for these percentages is outlined in Appendix B using cumulative fluxes from the atmosphere. The carbon fraction attributed to the ocean & sediments is labeled this way, as the air–sea $CO_2$ flux may indirectly include carbon from sediments through dissolution. However, we do not account for sediment carbon fluxes directly since there is no direct air-sediment flux (see Appendix B for further details on this formulation).

5). Peak land carbon uptake in our simulations falls comfortably between values given by previous studies (Mikolajewicz et al., 2006; Vakilifard et al., 2022). Uncertainties in the land carbon response over the next millennium, including anthropogenic land use change, has been addressed with more detail in Kaufhold et al. (2024).

In the latter half of the first millennium, vegetation carbon decreases as atmospheric $CO_2$ concentrations fall (Fig. 5a). Soil carbon exhibits non-monotonous behaviour with increasing emissions: low emission scenarios (<1000 PgC) have gained up to 100 PgC, and high emission scenarios (>4000 PgC) have lost up to 150 PgC by 1 kyr (Fig. 5b). This is because, in the low emission scenarios, the accumulation of carbon in soil exceeds carbon loss due to enhanced soil respiration (Fig. 6h), preventing the overall decline in soil carbon inventory (Fig. 4a). This is not the case in high emission scenarios, as permafrost
area decreases so significantly that soil respiration counteracts any potential increase when temperatures are sufficiently high (Fig. 6i). Although soil carbon inventory decreases in high emission scenarios by 1 kyr, it recovers over the next 100 kyr as temperatures gradually decrease. In our simulations, the land is a net carbon sink for the entire 100 kyr due to an increase in vegetation carbon and recovery of soil carbon. At the end of the experiments, the land still stores approximately 7% (range 4–13%) of the emitted anthropogenic carbon (Fig. 4).



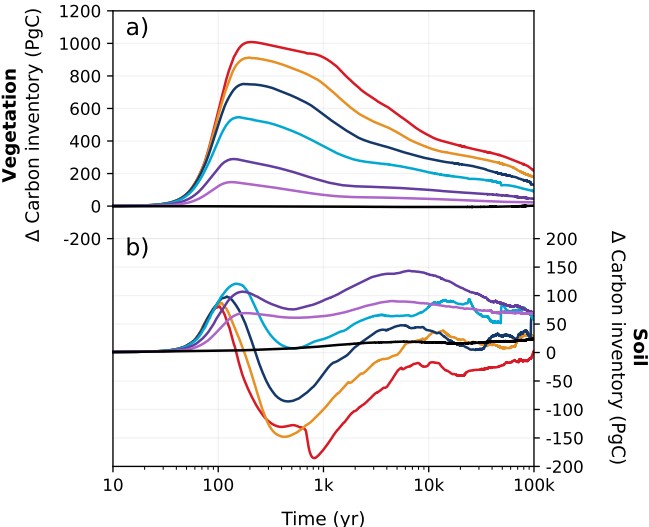

**Figure 5. Changes in (a) vegetation and (b) soil carbon inventories relative to the pre-industrial period in different emission scenarios over 100 kyr.** Colours correspond to the cumulative emission scenarios shown in Fig. 2.

### 3.1.2  Ocean and sediment carbon response

The ocean absorbs and stores the atmospheric $CO_2$ due to a sequence of chemical, biological, physical and geological processes. Under $CO_2$ emission pulses, the ocean initially takes up carbon at an accelerated rate from greater air–sea $CO_2$ exchange (3b). At peak $CO_2$ concentration, the ocean has a net cumulative uptake between ~100–780 PgC depending on the emission scenario (Fig. 2d). Oceanic $CO_2$ uptake yields the production of bicarbonate and hydrogen ions from $CO_2$ dissolution. As a consequence, surface ocean pH decreases at a similar rate that carbon is taken up in the ocean, with larger pH decreases associated with higher cumulative emissions (Fig. 7b). The magnitude of surface ocean pH decrease in our experiments match well with other studies: for example, the peak pH anomaly of approximately -0.7 in the 5000 PgC experiment falls within the range of -0.6 to -0.8 given by other models (Caldeira and Wickett, 2005; Montenegro et al., 2007; Uchikawa and Zeebe, 2008; Eby et al., 2009; Zickfeld et al., 2012; Colbourn et al., 2015). As phytoplankton growth rate strongly depends on temperature in HAMOCC (Ilyina et al., 2013; Willeit et al., 2023), net primary productivity (NPP) generally increases with higher sea surface temperature (SST) under larger cumulative emissions (Fig. 8a). Warming also accelerates the remineralization of organic matter in the ocean, which increases nutrient availability in the surface ocean for primary production. Moreover, reduced sea-ice cover in high latitudes increases light availability and, therefore, enhances local productivity. In our highest emission scenario, the combined effects amount to an approximate ~20% increase in peak NPP under the 5000 PgC scenario compared to the pre-industrial value (53.3 PgC yr$^{-1}$, Fig. 8a).

As the amount of dissolved $CO_2$ in the surface ocean rises, its ability to absorb excess $CO_2$ decreases. This decrease in buffering capacity is illustrated by the Revelle factor ($\xi$, the sensitivity of pCO2 to changes in DIC, Revelle and Suess (1957)),





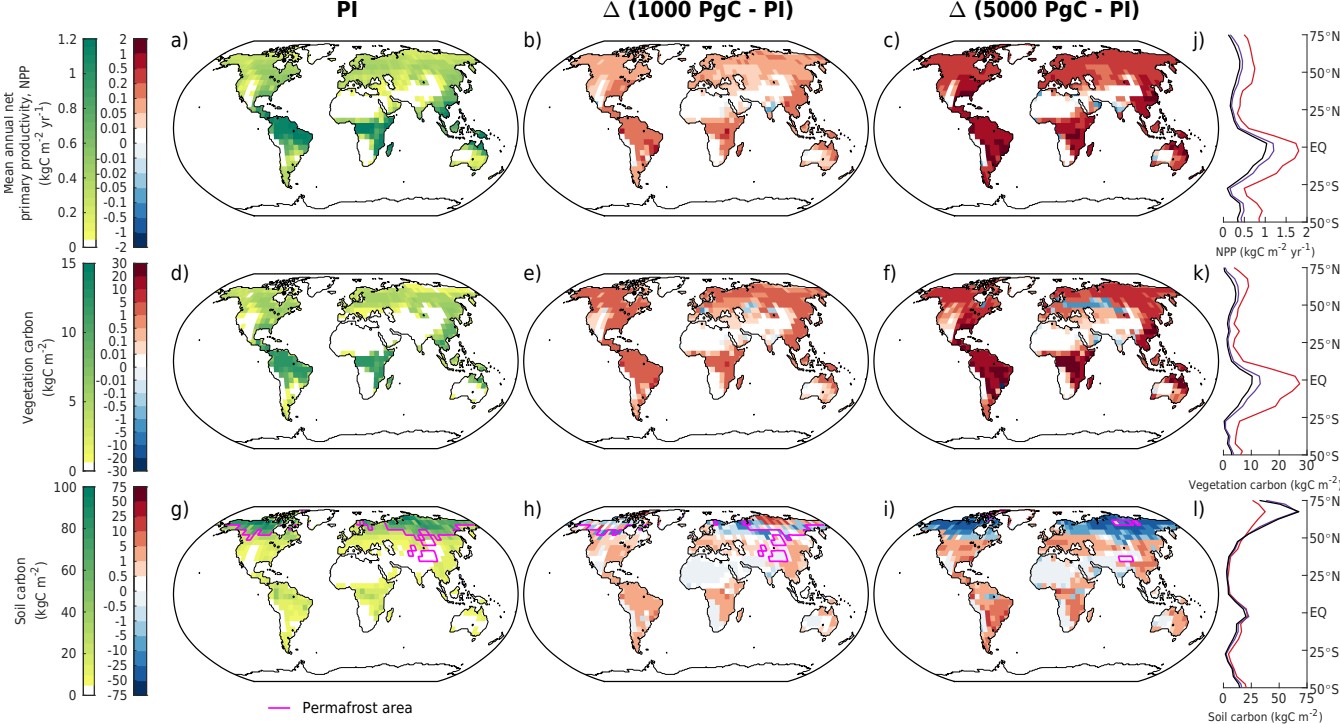

**Figure 6. Mean annual (a-b) net primary productivity, (d-f) vegetation carbon, and (g-i) soil carbon in different emission scenarios at 1 kyr.** The columns show (a,d,g) absolute values in the zero emissions scenario (equivalent to the pre-industrial), (b,e,h) 1000 PgC scenario relative to the pre-industrial, and (c,f,i) 5000 PgC scenario relative to the pre-industrial. Absolute values for the zonal mean are additionally shown for (j) NPP, (k) vegetation and (l) soil carbon for the zero emissions (black), 1000 PgC (purple), and 5000 PgC (red) scenarios. Magenta lines (g-i) display the permafrost area.

which increases with higher emissions (Fig. 7a). The pre-industrial Revelle factor of ∼9.4 matches reasonably well with that calculated by Jiang et al. (2019), and the peak Revelle factor in the 500 PgC scenario (∼10.2) is close to that calculated using present-day GLODAPv2 observations (Terhaar et al., 2022). The evolution of $\xi$ reversely follows that of pH (Fig. 7b), as the latter is a function of the ratio of DIC to alkalinity (Egleston et al., 2010). Within the first century, $\xi$ rapidly increases with surface DIC concentration (Fig. 7c), when surface alkalinity remains hardly changed (Fig. 7d). During this period, the surface DIC rise mainly results from the large air–sea $CO_2$ flux (Fig. 3b) and the consequential accumulation of anthropogenic carbon in the upper ocean before it is sufficiently mixed with the deep waters.

Between 100 yr and 1 kyr, $\xi$ slowly decreases, owing to the gradual decrease of surface DIC (except for the 5000 PgC scenario) and the increase of surface alkalinity. The decrease of surface DIC results from the combined effect of the downward transport of anthropogenic carbon in the surface ocean and the enhanced biological production (Fig. 8a), which is partly counteracted by the increase of the global DIC inventory (Fig. 7c). Both DIC and alkalinity inventories increase during this period due to high carbonate weathering rates (input as $HCO_3^-$ into the ocean, see Fig. 9a) and enhanced refluxes from sediment.





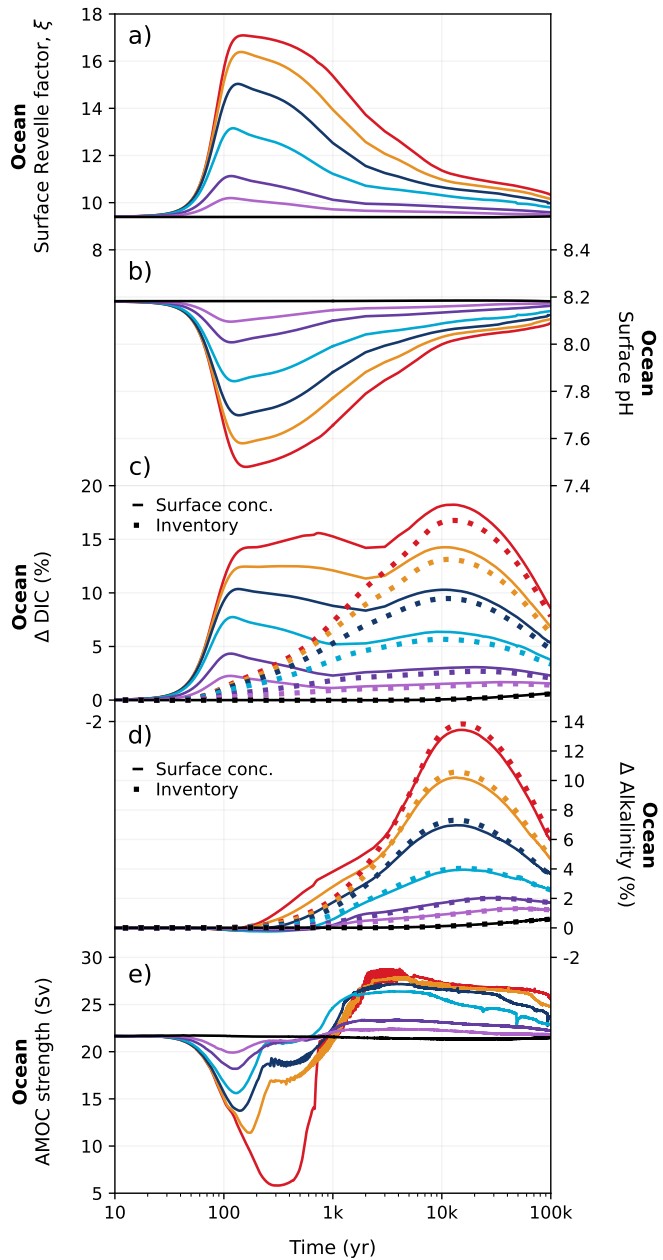

**Figure 7. Changes in (a) ocean buffering capacity (Revelle factor), (b) global mean surface pH, (c) relative changes in surface DIC concentration and global ocean DIC inventory, (d) relative changes in surface alkalinity concentration and global alkalinity inventory, and (e) strength of the Atlantic Meridional Overturning Circulation (AMOC) in different emission scenarios over 100** kyr. Colours correspond to the cumulative emission scenarios shown in Fig. 2. The surface ocean Revelle factor shown in (a) has been calculated via Eq. C1 in Appendix C.





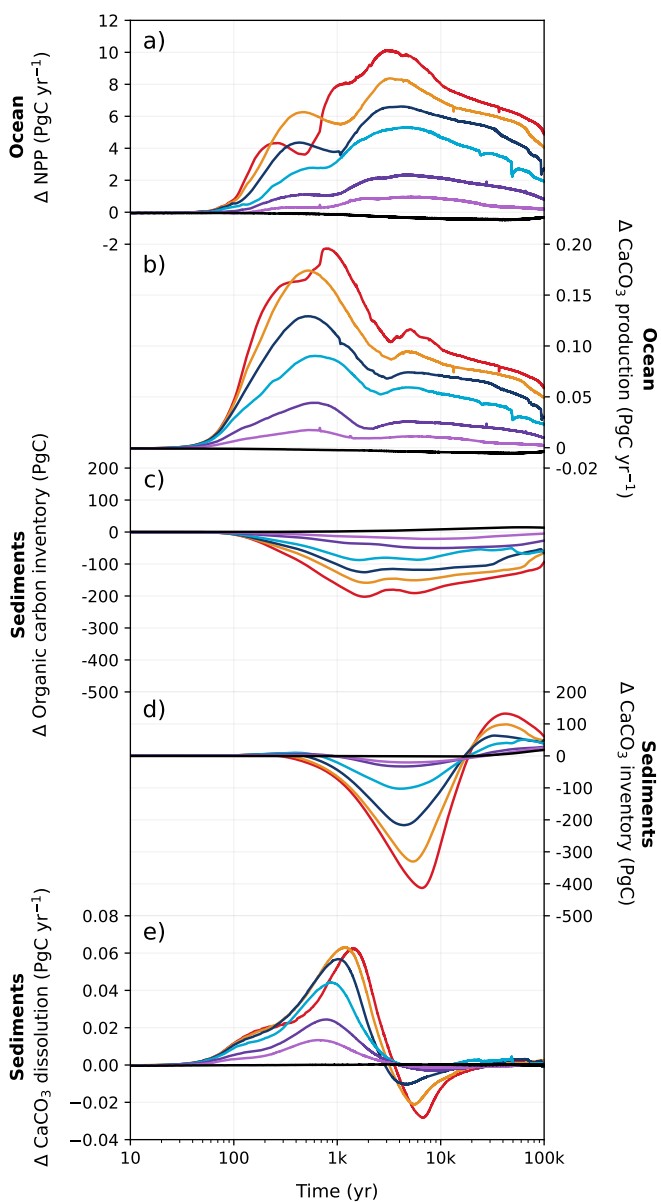

**Figure 8. Changes in (a) marine net primary production, (b)** $CaCO_3$ **production, (c) sediment particulate organic carbon inventory, (d) sediment** $CaCO_3$ **inventory, and (e) sediment** $CaCO_3$ **dissolution rate in different emission scenarios over 100** kyr**.** Colours correspond to the cumulative emission scenarios shown in Fig. 2.



Although NPP increases with warming, the loss of organic carbon to the sediment (not shown) and the organic carbon content in the sediment (Fig. 8c) are reduced due to higher remineralization rates in both the water column and sediment. Deep convection, which transports surface waters with decreased pH and carbonate ion concentrations into the interior ocean, together with increased organic matter remineralization, causes carbonate ion concentrations in the interior ocean to fall with increasing emissions (Fig. E6). In response, the lysocline and carbonate compensation depth (CCD) shoal and sediment $CaCO_3$ dissolves,

driving the increase of alkalinity inventory. In the 5000 PgC scenario, surface DIC increases between years 1100 to 1700, which is likely due to extended long period of AMOC decline (Fig. 7e) that inhibits deep convection and the downward carbon transport.

Surface DIC shows diverse variations between 1 and 10 kyr depending on the emission scenario, whereas surface alkalinity generally increases and is responsible for the recovery of the buffering capacity (shown by the continuous decline of ξ). The

variation of surface alkalinity generally follows that of the global inventory, with the latter controlled by the weathering input and net loss to the sediment. The sediment $CaCO_3$ continues to dissolve and with a higher rate than the first millennium (Fig. 8e). Beyond 10 kyr, both surface DIC and alkalinity decline, following the respective global inventory, as a response of decreased carbonate weathering (Fig. 9a) and increased loss to the sediment (Fig. 8c,d). Due to silicate weathering (Section 3.1.3) which reduces atmospheric $CO_2$, the ocean becomes a carbon source, contributing to the decrease of surface DIC and

recovering of buffering capacity.

The impact of carbonate compensation and seafloor $CaCO_3$ neutralization on restoring the buffering capacity on millennium timescales shown in these simulations are in line with previous studies (Archer et al., 1997, 1998; Tyrrell et al., 2007a; Tyrrell, 2007b). The recovery of sediment $CaCO_3$ inventory occurs roughly around the same time as predicted by Lenton and Britton (2006). However, the timing of peak $CaCO_3$ dissolution in CLIMBER-X is earlier than in a CLIMBER-2 study by Montenegro

et al. (2007), who found peak $CaCO_3$ dissolution around 1500–2500 years after the start of their simulation. In CLIMBER-X, the biological production of $CaCO_3$ increases with emissions as it is proportional to the production of organic detritus (and thus proportional to net primary production), leading to more $CaCO_3$ export to the interior ocean. Such changes in $CaCO_3$ production is not represented in CLIMBER-2, contributing to the different timings of peak dissolution between Montenegro et al. (2007) and this study.

**3.1.3 Weathering response**

While the pre-industrial weathering rates in our simulations (19.8 Tmol C yr$^{-1}$ for carbonate, 12.3 Tmol C yr$^{-1}$ for silicate; Fig. 9) fall within, but toward the higher end of, observational estimates (Meybeck, 1987; Gaillardet et al., 1999; Munhoven, 2002; Amiotte Suchet et al., 2003), they are significantly higher than previous studies investigating the long-term future $CO_2$ evolution (Uchikawa and Zeebe, 2008; Colbourn et al., 2015; Lord et al., 2015b). This is at least partly due to the spatially

explicit weathering scheme employed in our model, as switching from a 0D to a 2D scheme has been shown to increase weathering rates (Colbourn et al., 2013; Brault et al., 2017). Chemical weathering on land increases in high emission scenarios, due to warmer and wetter conditions, drawing down more carbon from the atmosphere (Fig. 9). Both carbonate and silicate weathering rapidly increase over the first 200 years, as a response to elevated temperatures and increased runoff (Lenton and





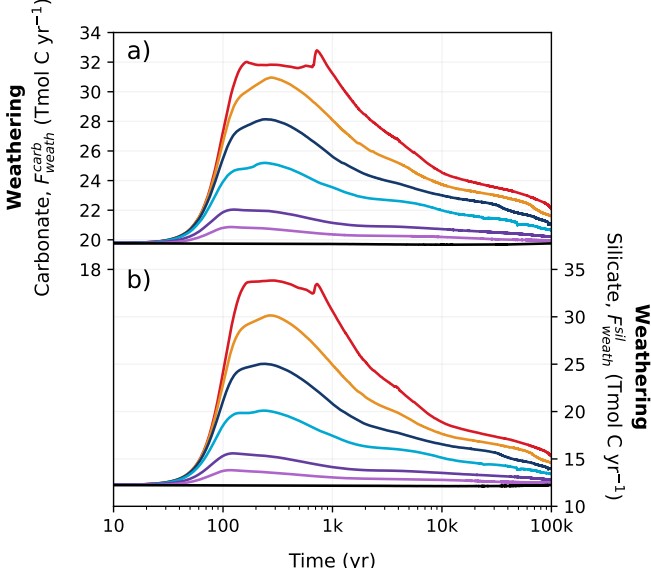

**Figure 9. Changes in (a) carbonate and (b) silicate weathering rates in different emission scenarios over 100** kyr. Colours correspond to the cumulative emission scenarios shown in Fig.2.

Britton, 2006). Carbonate weathering flux increases to 20.8-32.0 Tmol C $yr^{-1}$ in the different scenarios around the same time as

peak atmospheric $CO_2$ concentration (Fig. 9a). Peak silicate weathering occurs at the same time, but has even larger differences from the pre-industrial, nearly tripling under the high emission scenario (Fig. 9b). After peak weathering rates are reached in the REF experiments, they begin to decrease within the first millennium as atmospheric $CO_2$ decreases. Silicate weathering removes approximately 3% of anthropogenic $CO_2$ 1 kyr after peak concentrations, but this increases to approximately 73% over the course of 100 kyr (Fig. 4).

The sensitivity of carbonate and silicate weathering to climate change in CLIMBER-X is compared to estimates from earlier investigations in Fig. 10a,b. Generally, the CLIMBER-X response falls in between that what is estimated by previous studies (Sundquist, 1991; Uchikawa and Zeebe, 2008; Colbourn et al., 2015; Lord et al., 2015b), with the response of carbonate weathering falling in the middle, and silicate weathering tending to the stronger side. In both cases, however, weathering response is higher than simulated by the equations used in Lord et al. (2015b). In addition to this, the more sophisticated

weathering scheme in CLIMBER-X produces a narrow hysteresis in the response to the change in global mean temperature (Fig. 10a,b), as weathering rates decrease slower than global mean temperature. This hysteresis is primarily due to a lag between the evolution of precipitation and temperature. The combination of rather large pre-industrial silicate weathering flux and large sensitivity of silicate weathering to climate in the model results in a weathering feedback that is stronger than in some previous studies (Lord et al., 2015b; Colbourn et al., 2015). This has implications for the lifetime of atmospheric $CO_2$ (Section 3.2).

Most previous studies on deep future $CO_2$ concentrations did not spatially resolve weathering, instead relying on global mean values. The few studies which did use spatially explicit weathering employed either the GKWM (Bluth and Kump,



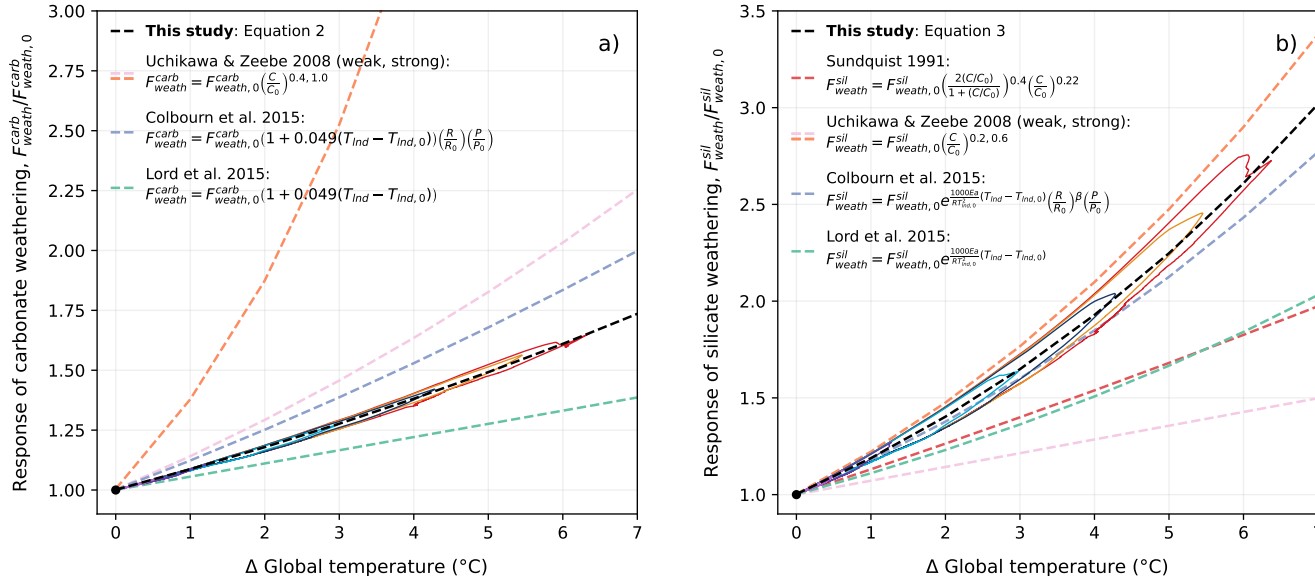

**Figure 10. Relationship between (a) carbonate weathering, (b) silicate weathering, and temperature in previous studies and CLIMBER-X.** Solid coloured lines corresponding to the cumulative emission scenarios seen in Fig. 2 represent the weathering response in the REF experiment. The black dashed line in (a) and (b) show an exponential fit of the trajectory of the weathering response. Calculated trajectories have additionally been plotted for the equations used by Sundquist (1991), Uchikawa and Zeebe (2008), Colbourn et al. (2015) (with runoff and productivity parametrized as a function of $CO_2$ as in Colbourn et al. (2013)), and Lord et al. (2015b) specifically using CLIMBER-X variables for atmospheric $CO_2$ and land surface temperature. An exponential fit of these trajectories have been plotted above.

1991) or GEM-CO2 (Amiotte Suchet et al., 2003) lithological maps, which are quite coarse resolution compared to the GLiM (Hartmann, 2009a) lithological map used by CLIMBER-X. The pre-industrial carbonate and silicate weathering distributions are shown in Fig. 11a,d. These distributions align quite closely with the calculated distribution of $CO_2$ consumption from

Hartmann et al. (2009b). However, not all details can be reproduced given the 5°×5°resolution of CLIMBER-X and biases in precipitation (Fig. 11g, see Willeit et al. (2022)). Weathering rates rise with increasing temperature, particularly in regions where significant weathering is already taking place. Runoff, which indirectly depends on precipitation through soil infiltration and drainage (Willeit and Ganopolski, 2016), drives some of these changes (Fig. 11g,h,i). As emissions grow, silicate weathering sees a significant increase in the Tropical rain belt area due to higher precipitation. These include regions such as South

East Asia, Central Africa, and Brazil (Fig. 11k). For carbonate weathering, large changes are not only limited to the equatorial regions, although the highest and lowest weathering rates in South East Asia and Central Asia can also be explained by increases and decreases in precipitation (Fig. 11b,c). Such spatial patterns in silicate and carbonate weathering under idealized 1000 and 5000 PgC scenarios have also been seen by Brault et al. (2017). In our simulations, highly active weathering regions are dominant contributors to the global weathering flux. About 5% of land area across the different emission scenarios is re-

sponsible for ∼37% of the total carbonate weathering and ∼32% of the total silicate weathering (Fig. 11b,c,e,f). These values





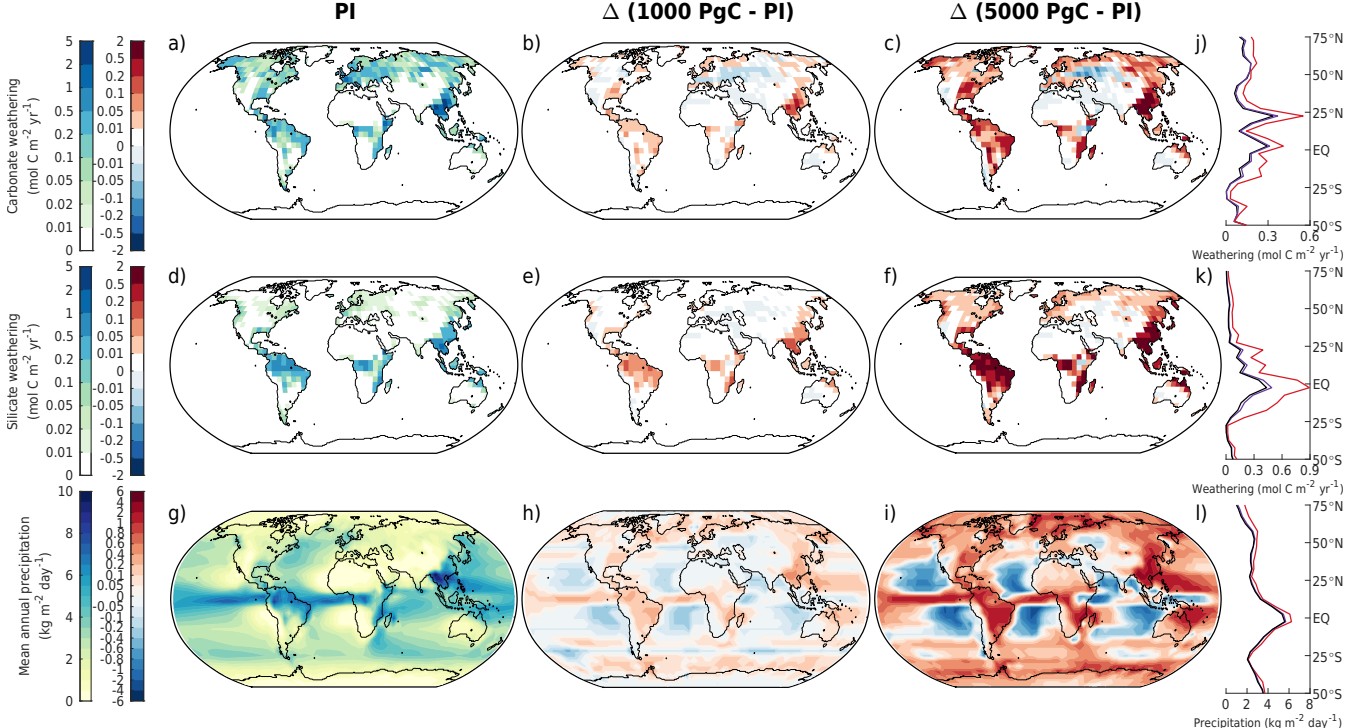

**Figure 11. Mean annual (a-c) carbonate weathering, (d-f) silicate weathering and (g-i) precipitation in different emission scenarios at 1 kyr.** The columns show (a,d,g) absolute values in the zero emissions scenario (equivalent to the pre-industrial), (b,e,h) 1000 PgC scenario relative to the pre-industrial, and (c,f,i) 5000 PgC scenario relative to the pre-industrial. Absolute values for the zonal mean are additionally shown for (j) carbonate weathering, (k) silicate weathering, and (l) precipitation for the zero emissions PgC (black), 1000 PgC (purple), and 5000 PgC (red) scenarios.

generally align with present-day estimates by Hartmann et al. (2009b). Given that the simulated weathering rates for both carbonate and silicate rocks are quite heterogeneous spatially, simplifications used by previous studies to calculate weathering rates (e.g., weathering as a function of global mean land surface temperature) could be insufficient to describe the feedback between climate and weathering.

## 3.2 Atmospheric lifetime of anthropogenic CO$_2$

The atmospheric lifetime of anthropogenic CO$_2$ across the different scenarios is estimated using an impulse response function (IRF). This is a first-order approximation on the long-term effects of a short term perturbation. It can be applied to changes in atmospheric CO$_2$ concentration (normalized by cumulative CO$_2$ emissions) such that it represents the fraction of emissions remaining (Maier-Reimer and Hasselmann, 1987; Joos et al., 2013; Jeltsch-Thömmes and Joos, 2020). In this formulation, it is functionally no different than airborne fraction, but we distinguish the two by assuming the IRF is only valid for time after peak CO$_2$ concentration (CO$_2(t_0)$ = CO$_2^{max}$). This can be written as the following:



$$\text{IRF}_{\text{CO}_2}(t) = \frac{\Delta\text{CO}_2(t) \times 2.124}{E} \tag{4}$$

where E is cumulative emissions in PgC, $\Delta\text{CO}_2(t)$ is the change in atmospheric $\text{CO}_2$ concentration at time t compared to the pre-industrial concentration (280 ppm), and 2.214 PgC ppm$^{-1}$ is the conversion factor between concentration and mass (Friedlingstein et al., 2023). The IRF for the REF ensemble is shown in Fig. 12. Unlike previous studies (e.g., Archer et al., 2009a; Joos et al., 2013), we do not deploy an instantaneous pulse, and our fractions of emissions remaining do not start at 1 as carbon sequestration already began during the ramp up of emissions. As this is an important consideration, the IRF

for the PATH1 and PATH2 experiments has additionally been plotted in Fig. 12b to show the relative impact of the emissions pathway. Although the emissions pathway has a small impact on the IRF within the first millennium, this effect becomes largely negligible after ∼3 kyr. This supports the long-term path independence found by Zickfeld et al. (2012) and Herrington and Zickfeld (2014), meaning that our results are fully comparable with those in the Long-Term Model Intercomparison Project (LTMIP) in Archer et al. (2009a).

Modelling studies have suggested that the majority (up to 80%) of anthropogenic $\text{CO}_2$ is quickly taken up by fast carbon processes, leading to a mean lifetime of only a few centuries (Archer, 2005; Montenegro et al., 2007; Archer and Brovkin, 2008). In the REF experiment, the mean lifetime is 390 (range 0-955) years (Fig. 12c). This value depends on the cumulative emission loaded to the atmosphere (Fig. 12c), and to some extent, the pathway of the emission (Fig. 12b). This explains why the lifetime determined by the timing of $\text{IRF}_{\text{CO}_2}(t) = 1/e$ is close to zero in the 500 PgC scenario, as the majority of carbon is

taken up from the atmosphere prior to reaching peak atmospheric $\text{CO}_2$ concentration in our simulations (Fig. 4a). On average, it takes between 197–1,820 years after peak $\text{CO}_2$ concentration to remove 75% of anthropogenic emissions, depending on the emission scenario. The most comprehensive assessment to date indicates that between 20-35% of $\text{CO}_2$ remains in the atmosphere after 200–2,000 years (Archer et al., 2009a). Our results generally agree with this estimate (Fig. 12c). After 10 kyr, it has been previously reported that $\text{CO}_2$ emissions in the atmosphere would remain somewhere between ∼5–20% for

the 1000 PgC scenario and ∼10–35% for 5000 PgC scenario (Archer et al., 2009a). In all experiments, atmospheric $\text{CO}_2$ concentration has decreased to approximately 10% of its initial value after 10 kyr across the different emission scenarios (Fig. 12c). Our values are on the lower end of the range provided by Archer et al. (2009a) due to the combined effect of a higher pre-industrial weathering rate and stronger weathering feedbacks. After 100 kyr from the start of the experiments, the fraction of emissions remaining is on average 4.3% (range 4.0–4.4%) across the different emission scenarios (Fig. 12a). This falls in

between estimates by previous studies, which have reported that between 2–7% of emissions will remain in the atmosphere beyond 100 kyr (Archer, 2005; Lenton and Britton, 2006; Jeltsch-Thömmes and Joos, 2020).





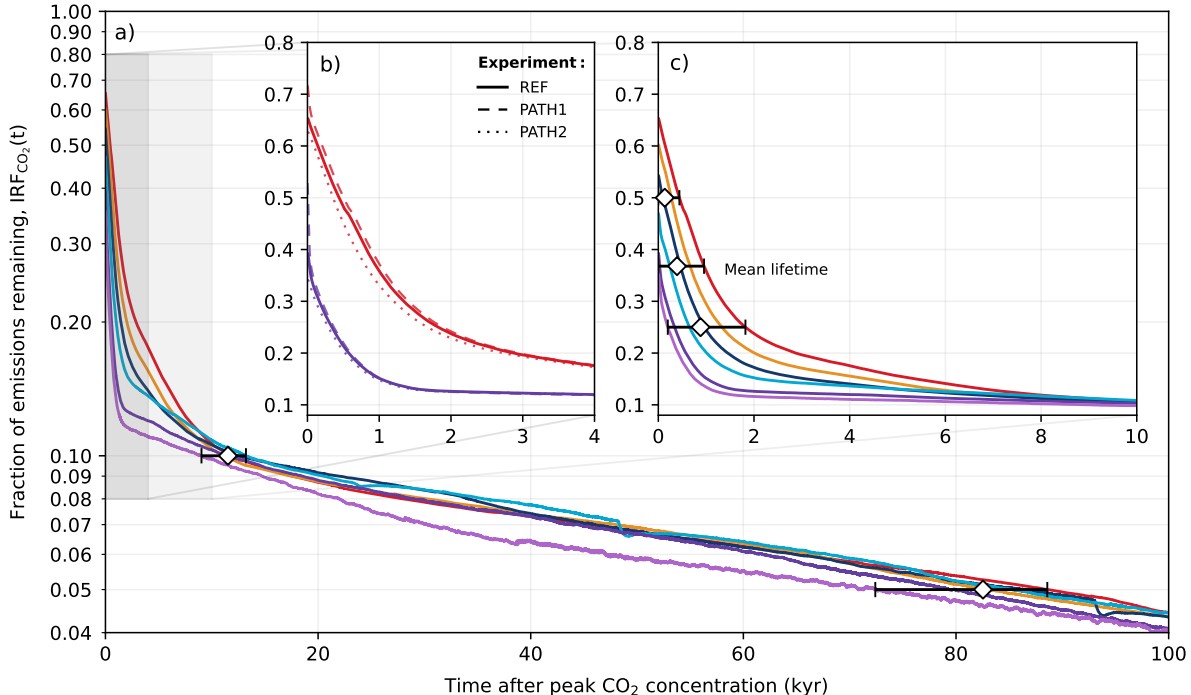

**Figure 12. Atmospheric lifetime of anthropogenic** $CO_2$ **described by** $\mathbf{IRF_{CO_2}(t)}$ **in the REF, PATH1, and PATH2 experiments.** Colours correspond to the cumulative emission scenarios shown in Fig. 2. (a) Fraction of emissions remaining in the atmosphere over the course of 100 kyr for the different emission scenarios. (b) Same as (a), but only showing the 1000 PgC and 5000 PgC experiments for the first 4 kyr after peak $CO_2$ concentration. Here, the PATH1 and PATH2 experiments have additionally been plotted to show the effect of the emissions pathway on the IRF. (c) Same as (a), but for 10 kyr. The diamond markers and horizontal bars represent the mean time and spread in which the different emission scenarios reach 50%, ≈ 1/e % (mean lifetime), 25%, 10% and 5% of its original cumulative emission.

### 3.3 Multi-exponential timescale analysis

We conducted a least-squares fit of a multi-exponential decay function to tease out different timescales which can be otherwise associated with long-term processes that take up excess $CO_2$. This analysis was exclusively done on the REF and noLAND 365 ensembles; the latter so that our results could be directly compared to those in Lord et al. (2015b). The Python function curve_fit in Scipy.Optimize was used, employing nonlinear least squares to fit the following function to our data:

$$CO_2(t) = 280 + \Delta CO_2^{max} \sum_{i=1}^{n} A_i \cdot \exp\left(-\frac{t}{\tau_i}\right) \tag{5}$$





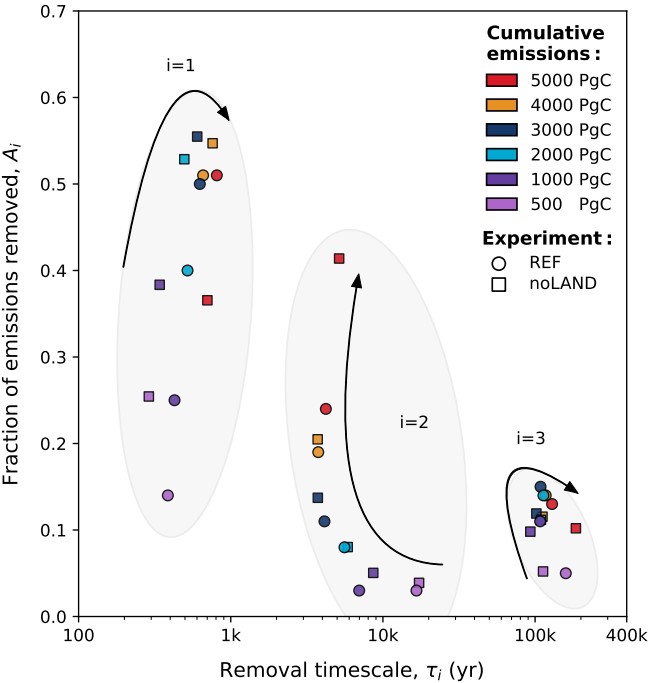

**Figure 13. Dependence of fitting parameters ($A_i$ and $\tau_i$) as determined in the multi-exponential decay analysis of $CO_2$ concentration in the REF and noLAND experiments.** Different markers have been used to convey different experiments. The arrows in the figure indicate how the ability of individual processes to remove anthropogenic $CO_2$ change as total emissions increase.

where $CO_2(t)$ is the $CO_2$ concentration at time t (where $CO_2(t_0) = CO_2^{max}$), and $\Delta CO_2^{max}$ is peak $CO_2$ concentration compared to pre-industrial concentration (280 ppm). This function represents the superposition of exponential decay functions each with their own time constant, $\tau_i$, and amplitude, $A_i$ (Maier-Reimer and Hasselmann, 1987). Here, the former represents the timescale of decay (i.e., the removal timescale), and the latter represents the fraction of emissions removed over that timescale. Since an instantaneous emission pulse was not used, the first ∼200 years of our simulations were dedicated to the ramp up and

ramp down of atmospheric $CO_2$ concentration (Fig. E1). Therefore, we decided to fit our data starting from peak concentration in atmospheric $CO_2$. Several superimposed exponential functions were used to fit the data (from n=1 to 6), but fast convergence times and significantly high (>0.99) $R^2$-values were found for n = 3.

We should first highlight that it is not surprising that n = 3 yielded the best fit. Although the equilibrium timescales for "fast" carbon processes have been thoroughly examined in previous literature (e.g., Eby et al., 2009), Archer and Brovkin (2008)

outlined that long-term carbon uptake would be predominately controlled by the slow processes of (1) ocean invasion, (2) reactions with $CaCO_3$, and (3) reactions with igneous (i.e., silicate) rocks. These are sometimes more broadly grouped into "ocean" and "geological" processes, with $CaCO_3$ reactions further split into sea floor and terrestrial $CaCO_3$ neutralization. Archer et al. (1997) found an optimal fit with n = 4, where $CaCO_3$ reactions were split into these two components. Colbourn et al. (2015)





**Table 2. Fitting parameters used for a multi-exponential decay function describing atmospheric $CO_2$ concentration in the REF and noLAND experiment ensembles.** Here, the fraction of emissions ($A_i$) taken up over the removal timescales ($\tau_i$) associated with ocean invasion ($i = 1$), $CaCO_3$ reactions ($i = 2$) and silicate weathering ($i = 3$) are shown. We highlight that this is the total fraction of emissions removed after peak $CO_2$ concentrations have been reached. Anthropogenic $CO_2$ removed during the ramp up of emissions is not considered in this analysis, however, fraction of emissions removed during the ramp up of emissions (via surface ocean processes) can be broadly estimated from $1 - \sum A_i$.

| Cumulative Emissions, E (PgC) | $CO_2$ Concentration Peak, $CO_2^{max}$ (ppm) | i = | 1 | 2 | 3 | Fraction Removed, $\sum A_i$ | $R^2$-Value |
|---|---|---|---|---|---|---|---|
| REF experiment | | | | | | | |
| 500 | 363 | $A_i$ | 0.14 | 0.03 | 0.05 | 0.22 | 0.9989 |
| | | $\tau_i$ | 386 | 16612 | 159105 | | |
| 1000 | 465 | $A_i$ | 0.25 | 0.03 | 0.11 | 0.39 | 0.9992 |
| | | $\tau_i$ | 427 | 6982 | 107893 | | |
| 2000 | 722 | $A_i$ | 0.4 | 0.08 | 0.14 | 0.62 | 0.9983 |
| | | $\tau_i$ | 520 | 5587 | 113421 | | |
| 3000 | 1047 | $A_i$ | 0.50 | 0.11 | 0.15 | 0.76 | 0.9992 |
| | | $\tau_i$ | 626 | 4128 | 108592 | | |
| 4000 | 1415 | $A_i$ | 0.51 | 0.19 | 0.14 | 0.84 | 0.9991 |
| | | $\tau_i$ | 658 | 3758 | 117405 | | |
| 5000 | 1819 | $A_i$ | 0.51 | 0.24 | 0.13 | 0.88 | 0.9992 |
| | | $\tau_i$ | 810 | 4216 | 129309 | | |
| noLAND experiment | | | | | | | |
| 500 | 426 | $A_i$ | 0.25 | 0.04 | 0.05 | 0.34 | 0.9994 |
| | | $\tau_i$ | 289 | 17284 | 112811 | | |
| 1000 | 594 | $A_i$ | 0.38 | 0 .05 | 0.1 | 0.53 | 0.9996 |
| | | $\tau_i$ | 341 | 8638 | 93042 | | |
| 2000 | 974 | $A_i$ | 0.53 | 0.08 | 0.1 | 0.72 | 0.9996 |
| | | $\tau_i$ | 495 | 5868 | 109293 | | |
| 3000 | 1381 | $A_i$ | 0.55 | 0.14 | 0.12 | 0.81 | 0.9996 |
| | | $\tau_i$ | 602 | 3733 | 101937 | | |
| 4000 | 1804 | $A_i$ | 0.55 | 0.20 | 0.12 | 0.87 | 0.9992 |
| | | $\tau_i$ | 760 | 3725 | 111838 | | |
| 5000 | 2234 | $A_i$ | 0.37 | 0.41 | 0.10 | 0.88 | 0.9993 |
| | | $\tau_i$ | 702 | 5155 | 185413 | | |



found a fit using n = 6, corresponding to both "fast" and "slow" processes of ocean invasion/carbonate chemistry, mixed-layer
mixing, deep ocean mixing, sediment dissolution, carbonate weathering, and silicate weathering. Lord et al. (2015b) found
an optimal fit with n = 5, however, they categorize the first three terms as part of a broader ocean sink classification. Jeltsch-
Thömmes and Joos (2020) also found a fit using n = 5, but attribute at least part of the shorter timescales to land processes.
In the REF experiment, removal timescales of 386–810 years for i = 1, approximately 4–17 kyr for i = 2, and approximately
108–159 kyr for i = 3 were found across the different emission scenarios (Table 2). For the noLAND experiment, these ranges
change to 289–702 years for i = 1, approximately 4–17 kyr for i = 2, and approximately 93-185 kyr for i = 3.

The centennial magnitude of the first removal timescale (i = 1) suggests the involvement of ocean circulation, which we
interpret as deep ocean processes responsible for transporting $CO_2$ away from the surface. The range determined in both the
REF and noLAND experiments fit well within previous estimates: for example, Archer et al. (2009a) found a range of 250–450
years, while Lord et al. (2015b) found 230-880 years. The ability of this process to remove carbon with increasing emissions
is nonlinear (Fig. 13), as the fraction of emissions removed increases up to ∼3000–4000 PgC cumulative emissions (Fig. 13).
A major cause of this behaviour is the slowdown of ocean circulation during the first millennium, as seen by the weakening of
AMOC. In our experiments, both the duration and magnitude of AMOC weakening increase with larger cumulative emissions
(Fig. 7e), followed by a subsequent strengthening of the AMOC after emissions cease which overshoots the PI value. This
promotes the downward transport of anthropogenic carbon and therefore increases the oceanic carbon uptake capacity (Mon-
tenegro et al., 2007; Lord et al., 2015b). In the 5000 PgC scenario, however, the behavior of this timescale differs significantly
between the two experiments (Fig. 13). While deep ocean processes in the REF experiment stabilize at a similar removal ca-
pacity as in the 4000 PgC scenario (∼51%, Table 2), its ability is notably reduced in the noLAND experiment (∼37%). In
the REF experiment, AMOC does not begin recovery until 200–300 years after emissions end in the 5000 PgC scenario (Fig.
7e), leading to large amount of anthropogenic carbon accumulating in the upper ocean by 1 kyr (Fig. E4). However, AMOC
eventually recovers and follows a similar trajectory to that of the 4000 PgC experiment. Meanwhile, AMOC fails to recover
with 5000 PgC in the noLAND experiment, shutting down for the majority of the simulation (not shown). This results in the
decreased ability of deep ocean processes to absorb additional anthropogenic $CO_2$.

Previous studies have given a range of 1–6 kyr for sediment dissolution, and 6–14 kyr for carbonate weathering (Sundquist,
1991; Archer et al., 1997; Ridgwell and Hargreaves, 2007; Colbourn et al., 2015; Lord et al., 2015b; Jeltsch-Thömmes and Joos,
2020). However, it has been emphasized that these processes should not be considered separately given their interdependence
(Lord et al., 2015b). We attribute the second calculated timescale (i = 2), which is similar in both sets of experiments and falls
within the estimated range, to the combined influence of both processes. As emissions increase, the removal timescale shortens,
and a higher fraction of emissions is removed (Fig. 13). The enhanced ability of this timescale to remove anthropogenic $CO_2$
with larger cumulative emissions is primarily driven by the resulting increase in carbonate weathering, which leads to more
$HCO_3^-$ in the ocean (Fig. 9a). In low emission scenarios, the calculated removal timescale is closer to approximately 17 kyr,
suggesting that terrestrial weathering of carbonate rocks is the dominant process of carbon removal. But considering that
this timescale generally has a limited capacity to remove additional anthropogenic $CO_2$ (∼3-4%, Table 2), it suggests that the





combined effect of enhanced carbonate weathering (Fig. 9a) and CaCO$_3$ dissolution (Fig. 8e) is relatively weak in low emission scenarios.

In high emission scenarios, however, the timescale shortens to about 4 kyr; closer to that which is estimated for marine sediment dissolution. Sediment dissolution from a shoaling lysocline and CCD in the 3000-5000 PgC scenarios reach a similar maximum of $\sim$0.17 PgC yr$^{-1}$ shortly after the first millennium (Fig. 8e). As this process restores buffering capacity to the ocean, and is responsible for the continued removal of anthropogenic CO$_2$, it is perhaps not so surprising that there is a discernible threshold in the removal timescale at these levels of emissions. Again, the removal ability of this timescale in the

5000 PgC scenario differs significantly between the REF and noLAND experiments, as the latter accounts for the removal of nearly twice as much anthropogenic CO$_2$. This shows that a weakening ability of deep ocean processes to take up CO$_2$ is compensated by an increasing ability of sediment dissolution, to some extent (similarly shown in Archer et al., 1997), and is because seafloor CaCO$_3$ neutralization will act to restore buffering capacity that has been depleted from highly alkaline deep water (Fig. E5c,g,k).

The last timescale to be considered (i = 3) pertains to silicate weathering. Various estimates have been proposed for this timescale. In knowing that the effect of silicate weathering only becomes apparent on long timescales, some studies have just held silicate weathering rates constant as to investigate the role of carbonate weathering on the removal of anthropogenic CO$_2$ (e.g., Montenegro et al., 2007; Ridgwell and Hargreaves, 2007). In other cases, the timescale of silicate weathering was estimated to be somewhere between 200–400 kyr (Archer et al., 1997; Archer, 2005). First attempts at simulating the effect

of the enhanced weathering feedback from increased temperature and net primary productivity on land showed that silicate weathering works on timescales of one hundred thousand years (e.g., Lenton and Britton, 2006). Some more recent studies have attempted to quantify the timescale of the silicate weathering feedback by incorporating a weathering dependence on temperature, runoff and productivity. Yet, when applied to deep future modelling, processes were not fully resolved (e.g., 0D weathering, lack of coupling with a land module), requiring additional parametrizations which lead to a persistent estimate of

$\sim$250 kyr for the silicate weathering timescale.

    The multi-exponential analysis shown here provides an estimate on the silicate weathering timescale which is significantly shorter than what has been previously reported. For the first time, a non-monotonous dependence of the silicate weathering timescale and capacity for carbon removal to cumulative emissions is found. This trend is seen in both the REF and noLAND experiments, although the noLAND experiments have a wider range in the timescale. In the 500 PgC scenario, silicate weath-

ering is responsible for removing around 5% of anthropogenic CO$_2$ (Table 2), whereas the removal timescale is approximately 159 kyr in the REF experiment and 113 kyr in the noLAND experiment – the latter nearly a factor 2 less than estimated by Archer et al. (1997), Colbourn et al. (2015), and Lord et al. (2015b). The ability of this process to remove carbon increases up to a maximum of 15% (seen in the REF experiment) as emissions increase. At the same time, the removal timescale of silicate weathering in the REF experiment decreases to approximately 107 kyr in the 1000 PgC scenario, before generally increasing

to approximately 129 kyr in the 5000 PgC scenario (Fig. 13). The initial shortening of the silicate weathering timescale with increasing emissions can be partially explained by spatially explicit weathering (Colbourn et al., 2013; Brault et al., 2017), and strong weathering feedbacks (Section 3.1). This is perhaps not unexpected given that differences in weathering parametriza-





tions have been found to change the estimated timescale of silicate weathering by more than 100 kyr (Uchikawa and Zeebe, 2008; Colbourn et al., 2015).

The behavior of the silicate weathering timescale mirrors that of $CaCO_3$ reactions, as the timescale generally decreases as emissions rise until approximately 3000–4000 PgC (Fig. 13). The threshold of $\sim$3000–4000 PgC is also seen for deep ocean processes, which increase in removal ability until that point. The non-monotonic behaviour of the silicate weathering timescale and carbon removal ability can be attributed to the same physical mechanisms responsible for the nonlinearity observed in deep ocean processes and $CaCO_3$ reactions. If removal by either of these processes is strong, it can diminish the effectiveness of silicate weathering by reducing the amount of atmospheric $CO_2$ available for it to act upon. As such, AMOC likely plays a significant role in the removal timescale of silicate weathering, as experiments with a delayed AMOC recovery (Fig. 7e), or a complete shutdown (not shown), show an increase in the timescale. Since emission scenarios greater than 5000 PgC were not explored, it is unclear whether this trend will continue with increasing emissions. Despite differences in our estimated timescale of silicate weathering, the ability of silicate weathering to remove carbon ($\sim$5–15% across all experiments and scenarios, Table

2) falls reasonably close to previous studies, which has reported values between 4–10% (Lenton and Britton, 2006; Ridgwell and Hargreaves, 2007; Archer, 2005; Colbourn et al., 2015; Lord et al., 2015b).

## 3.4    Continuum of removal timescales

The multi-exponential decomposition of atmospheric $CO_2$ (Section 3.3), is, to some extent, dependent on how well a discrete number of functions can describe the continuous decline in atmospheric $CO_2$. This drawback has been recognized and dis-

cussed by other studies (e.g., Jeltsch-Thömmes and Joos, 2020). Therefore, we have derived an alternative method to calculate the removal timescale of anthropogenic $CO_2$ emissions as a function of time. With this method, the following first-order exponential decay equation describing the decline of atmospheric $CO_2$ (Eq. 6) is solved to estimate the removal timescale, $\tau(t)$. This was done by estimating the derivative using a central difference approach:

$$\frac{d\Delta CO_2(t)}{dt} = \frac{-\Delta CO_2(t)}{\tau(t)} \tag{6}$$

$$\tau(t) = \frac{-2\Delta t \Delta CO_2(t)}{\Delta CO_2(t+\Delta t) - \Delta CO_2(t-\Delta t)} \tag{7}$$

where $\Delta CO_2(t)$ is the change in atmospheric $CO_2$ concentration at time t (where $CO_2(t_0) = CO_2^{max}$) compared to the pre-industrial concentration (280 ppm). In doing this, the time-dependence of the calculated removal timescales could be examined,

along with how it changes for different levels of cumulative $CO_2$ emissions. We employed a rolling mean with an increasing window size over time, such that larger removal timescales are averaged over longer time periods. This has been more thoroughly described in Appendix D. Like Sections 3.2 and 3.3, this analysis was carried out for data after peak $CO_2$ concentration had been reached ($CO_2(t_0) = CO_2^{max}$).



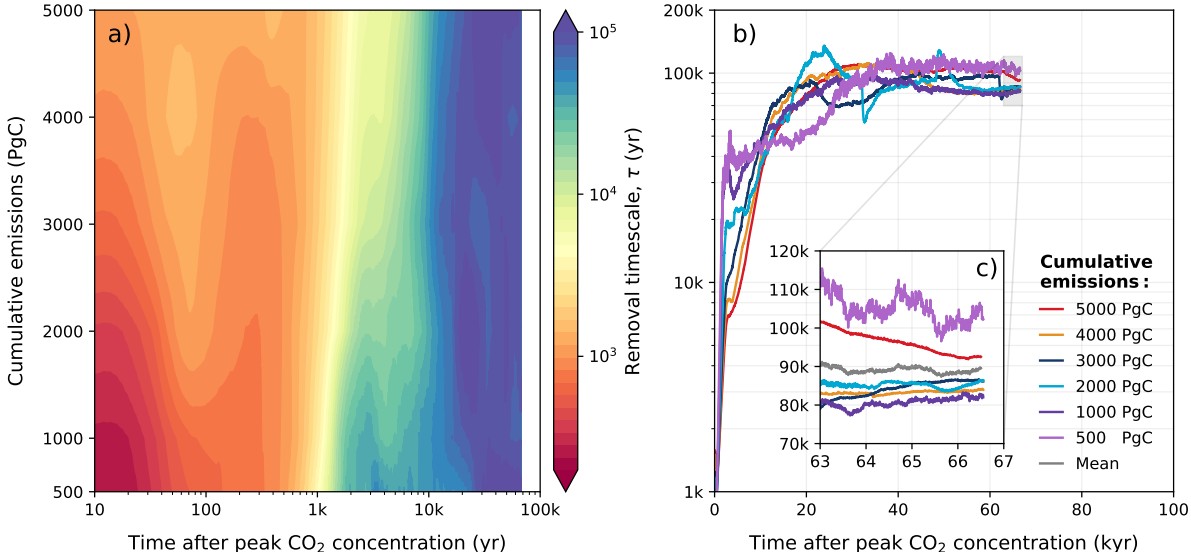

**Figure 14. Estimated removal timescale $\tau$ at any given time as determined from Eq. 7 for the different emission scenarios in the REF experiment.** Data has been filtered with the variable moving window discussed in Appendix D for visibility, and is why the estimated removal timescale is not available for the full 100 kyr.

The calculated removal timescale for the REF experiments, which includes changes in land carbon, are shown in Fig. 14a,b.
As expected, the removal timescale is generally short in the beginning. At peak $CO_2$ concentration, the estimated removal timescale ranges between $\sim$0.3–1.6 kyr in the REF experiment (Fig. 14a), with smaller emission scenarios having shorter removal timescales. The removal timescales in the 500 and 1000 PgC scenarios increase over the first 1 kyr in the REF experiment. In the 2000-5000 PgC scenarios, the removal timescales initially exhibit a similar increase over the first $\sim$100 years after peak $CO_2$ concentration. However, the timescales start to decrease again after that, reaching a local minimum
approximately 300 years after peak $CO_2$ concentration (Fig. 14a). This is the result of changes in soil carbon. For the high emission scenarios, thawing permafrost causes the soil to become a source of carbon around 100 years after peak atmospheric $CO_2$ concentration (Fig. 5b). This temporarily adds carbon to the atmosphere, keeping concentrations marginally inflated, and explains the initial increase in removal timescale at this time as $\tau(t) \propto \frac{1}{\frac{dCO_2(t)}{dt}}$. Soil carbon begins to recover under high emission scenarios $\sim$300 years after peak atmospheric $CO_2$ concentration (Fig. 5b). At this time, the opposite effect occurs: the
soil becomes a carbon sink, absorbing carbon at a rate greater than in the low emission scenario. This accelerates the decline in atmospheric $CO_2$ concentrations and temporarily shortens $\tau$. For low emission scenarios, the effect of soil carbon is not reflected in the removal timescale, as temperatures are not sufficiently high to cause warming-induced soil respiration, making soil carbon a net sink in the experiment (Fig. 5b).

Between 1–10 kyr, there is a period in time in which removal timescale temporarily increases for the 500 and 1000 PgC
scenarios (Fig. 14a). This behaviour in removal timescales, similarly discussed before for soil carbon, is here due to multi-



millennial processes which temporarily stabilize atmospheric $CO_2$ concentrations for a brief period. As this occurs around the same time as when $CaCO_3$ reactions are particularly significant for the removal of anthropogenic $CO_2$ (Fig. 13), this is likely related to the interplay between deep-sea carbonate dissolution and terrestrial weathering. The ability of seafloor $CaCO_3$ neutralization to remove anthropogenic $CO_2$ is particularly weak for low emission scenarios (Fig. 13, discussed in Section 3.3), as the deep water is not sufficiently alkaline to induce large $CaCO_3$ dissolution (Fig. 8e, Fig. E5b,f,j). This temporarily reduces the drawdown of excess atmospheric $CO_2$, which, in turn, increases the removal timescale.

At 50 kyr, the effect of carbonate weathering becomes minimal and the removal timescale primarily reflects silicate weathering. Interestingly, the removal timescale for the ∼1000–4000 PgC scenarios converge to a value lower than that in the 500 and 5000 PgC scenarios (Fig. 14c). This generally confirms that the non-monotonic behaviour of the silicate weathering timescale seen in the multi-exponential analysis for both the REF and noLAND experiments (Fig. 13) is not just an artifact of the fit. Averaging over the last 5,000 years of τ(t) yields a mean removal timescale of approximately 89 kyr for the REF experiment (range 80–105 kyr, Fig. 14c). This method generally provides estimates on the silicate weathering timescale that are slightly lower and exhibit less spread compared to those obtained in Section 3.3. The multi-exponential analysis is helpful for understanding the role and removal ability of each process individually. However, this second method provides more confidence in calculating the silicate weathering timescale, as it is not influenced by factors such as the number of superimposed exponential functions (n), or the goodness of fit.

## 4   Sensitivity experiments

To explore the sensitivity of our results to different carbon cycle processes and climate sensitivity, we have performed a number of additional experiments (Table 1). The results from these additional experiments are discussed below. To provide a linear analysis of our sensitivity experiments, we limit our emission scenarios to ≤3000 PgC. This is largely because high levels of cumulative emissions make it difficult to ignore secondary and cascading processes (e.g., shutdown of AMOC), which would introduce nonlinearities and provide additional uncertainties.

### 4.1   Land carbon sink

A complete disregard of the land carbon pool results in higher atmospheric $CO_2$ concentrations at the start of our simulations (Fig. 15a). Peak atmospheric $CO_2$ concentrations are between ∼65-340 ppm higher without the land carbon pool, depending on the emission scenario. After peak $CO_2$, the difference in atmospheric $CO_2$ concentrations between the noLAND and REF experiments begin to decrease for all emission scenarios up to 3000 PgC. Conversely, carbon inventory of the ocean increases to approximately 2300 PgC by 1 kyr, which is roughly 300 PgC more than in the REF experiment (Fig. 16d). The majority of this carbon uptake in the ocean occurs at peak $CO_2$ concentration from increased air–sea $CO_2$ flux. However, at least some of this carbon is from sediment dissolution (Fig. 16g), which contributes approximately 50 PgC more by 1 kyr. At the beginning of the noLAND experiment, less $CO_2$ is absorbed as a whole, as it is taken up by relatively slow ocean and weathering processes. This leads to higher fraction of emissions remaining compared to the REF experiment (Fig. 17a).



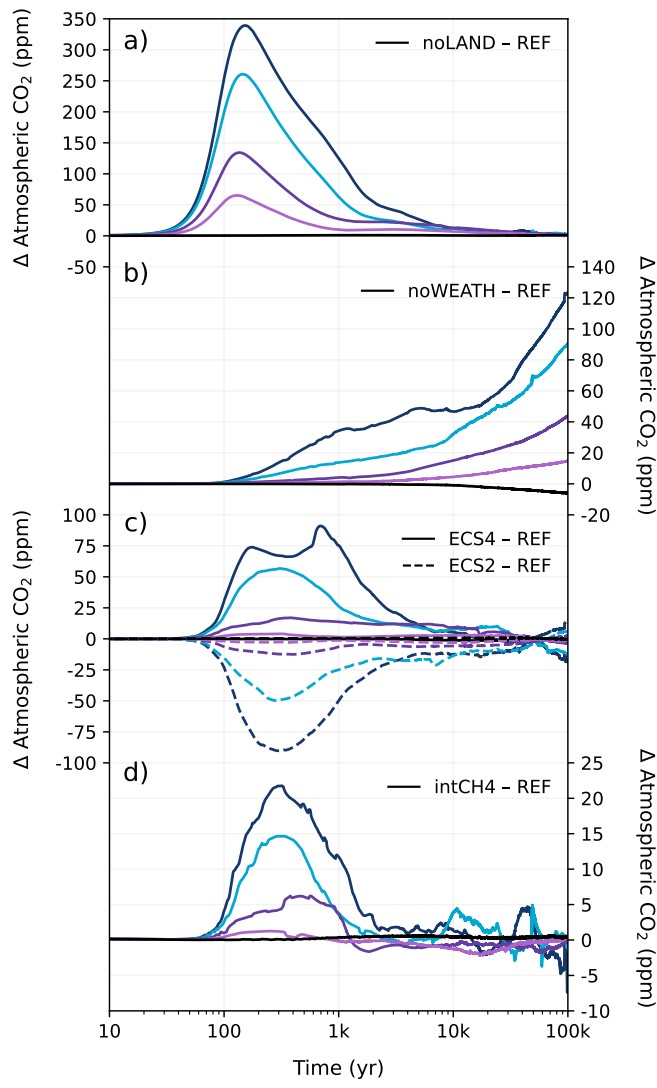

**Figure 15. Difference in atmospheric CO$_2$ concentration in the different sensitivity experiments compared to the REF experiment.**
(a) The effect of disabling the land carbon pool. (b) The effect of constant weathering (i.e., removal of the weathering feedback). (c) The effect of a lower (2 °C) and a higher (4 °C) equilibrium climate sensitivity compared to the standard 3.1 °C in CLIMBER-X. (d) The effect of interactive methane. Negative anomaly values represent a larger atmospheric concentration in the REF experiment. Colours correspond to the cumulative emission scenarios shown in Fig. 2.




It has long been recognized that land plays a crucial role in modulating the effects of weathering (Nugent and Matthews, 2012; Brault et al., 2017). However, the influence of land carbon on the long-term removal of anthropogenic $CO_2$ through

silicate weathering remains largely understudied. Since atmospheric $CO_2$ concentrations are higher in the noLAND experiment (Fig. 15a), peak temperatures and runoff are also elevated. As weathering in CLIMBER-X is driven by changes in runoff and temperature (see Eq. 2 and 3), the noLAND experiment produces higher weathering rates compared to the REF experiment. This leads to a faster decline of atmospheric $CO_2$ concentrations in the long term and a shorter silicate weathering timescale. Averaging over the last 5,000 years of $\tau(t)$ in Fig. 17b yields an average value of approximately 79 kyr for the noLAND

experiment. By 100 kyr, when silicate weathering becomes the dominant carbon removal process, the higher silicate weathering in the noLAND experiment brings atmospheric $CO_2$ concentrations closer to those in the REF experiment (Fig. 15a). As a result, the average fraction of emissions remaining between the REF and noLAND experiments differ by only 0.2% at this time (Fig. 17a). The higher weathering in the noLAND experiment therefore partially compensates both higher initial atmospheric $CO_2$ concentrations and a missing land carbon pool on long timescales.

**4.2  Silicate weathering feedback**

In the absence of a silicate weathering feedback, atmospheric $CO_2$ concentrations cannot be restored back to pre-industrial conditions (Fig. 17a). Initially, atmospheric $CO_2$ concentrations are quite similar between the REF and noWEATH experiments, as weathering is only responsible for taking up approximately 3% of anthropogenic $CO_2$ within the first 1 kyr after emissions end (Fig. 4). However, the two experiments quickly diverge as atmospheric $CO_2$ is additionally consumed by en-

hanced weathering in the REF experiment. The difference in atmospheric $CO_2$ concentration between the REF and noWEATH grows to ~120 ppm over the course of 100 kyr (Fig. 15). This is because the value to which atmospheric $CO_2$ concentrations returns to in the noWEATH experiments increases with larger emissions (with final values between ~300–470 ppm compared to the pre-industrial 280 ppm). Across the different emission scenarios, the last ~14% (range 13-15%) of anthropogenic $CO_2$ in the noWEATH experiment cannot be removed without the weathering feedback (Fig. 17). This estimate is larger than what

was calculated for the fraction of emissions removed by the silicate weathering timescale in the noLAND experiment (range 5–12%, Table 2) and range provided by previous studies (range 4–10%; Lenton and Britton, 2006; Ridgwell and Hargreaves, 2007; Archer, 2005; Colbourn et al., 2015; Lord et al., 2015b; Jeltsch-Thömmes and Joos, 2020). As atmospheric $CO_2$ concentrations do not decrease beyond ~300–470 ppm in the noWEATH experiments, temperatures stay significantly elevated (and near constant) until the end of the simulation. Climate is therefore very different in the noWEATH experiment, resulting

in notably different behaviour in the land, ocean, and sediment carbon pools compared to the REF experiment (see Fig. 16 for further details).

**4.3  Climate sensitivity**

The best estimate of equilibrium climate sensitivity is reported as 3 °C by the IPCC (Canadell et al., 2021), which is close with the estimated ECS of 3.1 °C in CLIMBER-X. However, the IPCC also reports that the very likely range is 2 °C to 5 °C,

so the effect of a lower and higher ECS (corresponding to values of 2 °C and 4 °C, Appendix A) was also tested. As seen

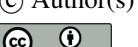

**Figure 16. Net cumulative uptake of carbon in the (a-c) land, (d-f) ocean, and (g-i) sediment carbon pools across the different sensitivity experiments and emission scenarios.** The value at which each stacked bar reaches represents the magnitude of carbon uptake as the stacked bar plot is cumulative. Positive values represents carbon into the land, ocean and sediment carbon pools. The time slices of (a,d,g) 1 kyr, (b,e,h) 10 kyr, and (c,f,i) 100 kyr were chosen to capture the effects of the different timescales.




in Fig. 15c, a lower ECS is associated with lower atmospheric concentrations of $CO_2$ (and vice versa), and higher levels of cumulative emissions produce larger differences from the REF experiment. The differences in atmospheric $CO_2$ concentration can largely be traced back to changes in land carbon inventory (Fig. 16a). In the case of ECS4, terrestrial vegetation takes up approximately the same amount of carbon as in the REF experiment (not shown). However, increased warming causes

soil carbon to become a larger source of carbon, resulting in higher atmospheric $CO_2$ concentrations and lower land carbon across the different emission scenarios over the 100 kyr. For ECS2, both the vegetation and soil carbon pools are net sinks of carbon for the entire duration of the simulation regardless of the emission scenario (not shown). Despite significant variations in the partitioning of carbon between the land and ocean carbon pools (Fig. 16), differences in atmospheric $CO_2$ concentration between the ECS2/ECS4 experiments and the REF experiment are within ~5 ppm by 10 kyr, and remain low for the rest of

the simulation (Fig. 15c).

At first glance, this might suggest that the impact of ECS on atmospheric $CO_2$ concentration is minimal after 10 kyr regardless of the cumulative emission scenario. This behaviour was similarly seen in Shaffer et al. (2009). However, this conclusion is only valid for a short period. In Fig. 17c,d, higher ECS values leads to a faster decline in atmospheric concentration and a shorter removal timescale of silicate weathering. Although atmospheric $CO_2$ concentrations are initially higher in the ECS4

experiment, they fall below those of the REF experiment around 50 kyr due to substantial consumption by weathering (Fig. 15c). The reverse is true for ECS2. Averaging over the last 5,000 years for $\tau(t)$ results in a mean silicate weathering timescale of 153 kyr for an ECS of 2 °C, and 54 kyr for an ECS of 4 °C. This generally corroborates findings by Colbourn et al. (2015), who found that a lower ECS would give a longer timescale for silicate weathering. With that in mind, the default ECS in Colbourn et al. (2015) and Lord et al. (2015b) was 2.64 °C, indicating that climate sensitivity could account for at least some

of the differences in our estimated silicate weathering timescale (Fig. 13).

### 4.4 Interactive methane

Global warming from anthropogenic $CO_2$ emissions can cause additional methane emissions from natural sources that can result in an additional positive feedback loop. Although methane has a relatively short atmospheric lifetime compared to our experiment duration (~9.8 years, Voulgarakis et al., 2013), it has a higher global warming potential compared to $CO_2$. An

increase in methane concentration could therefore effect the capacity of different Earth system components to take up carbon over time. For the first time, we investigate the impact of interactive methane on the long-term carbon cycle. A simple methane cycle has been implemented in CLIMBER-X that explicitly simulates the natural methane emissions from land resulting from anaerobic respiration in wetlands and peatlands (Willeit and Ganopolski, 2016). Atmospheric methane concentration is then computed assuming a constant atmospheric lifetime of methane (Willeit et al., 2023).

In the zero emissions scenario, $CH_4$ concentration stays largely around 600–700 ppb (close to the constant 600 ppb set in the REF experiment, Fig. E2). For the other scenarios (500–3000 PgC), atmospheric $CH_4$ concentrations rise over the first ~200 years, reaching a peak of approximately 1620 ppb in the 3000 PgC experiment. With interactive methane, atmospheric $CO_2$ concentrations can be up to 20 ppm higher than in the REF experiment (Fig. 15d). During much of the first 1 kyr, the difference in atmospheric $CO_2$ concentration between the intCH4 and REF experiments increase with higher emissions. However, this





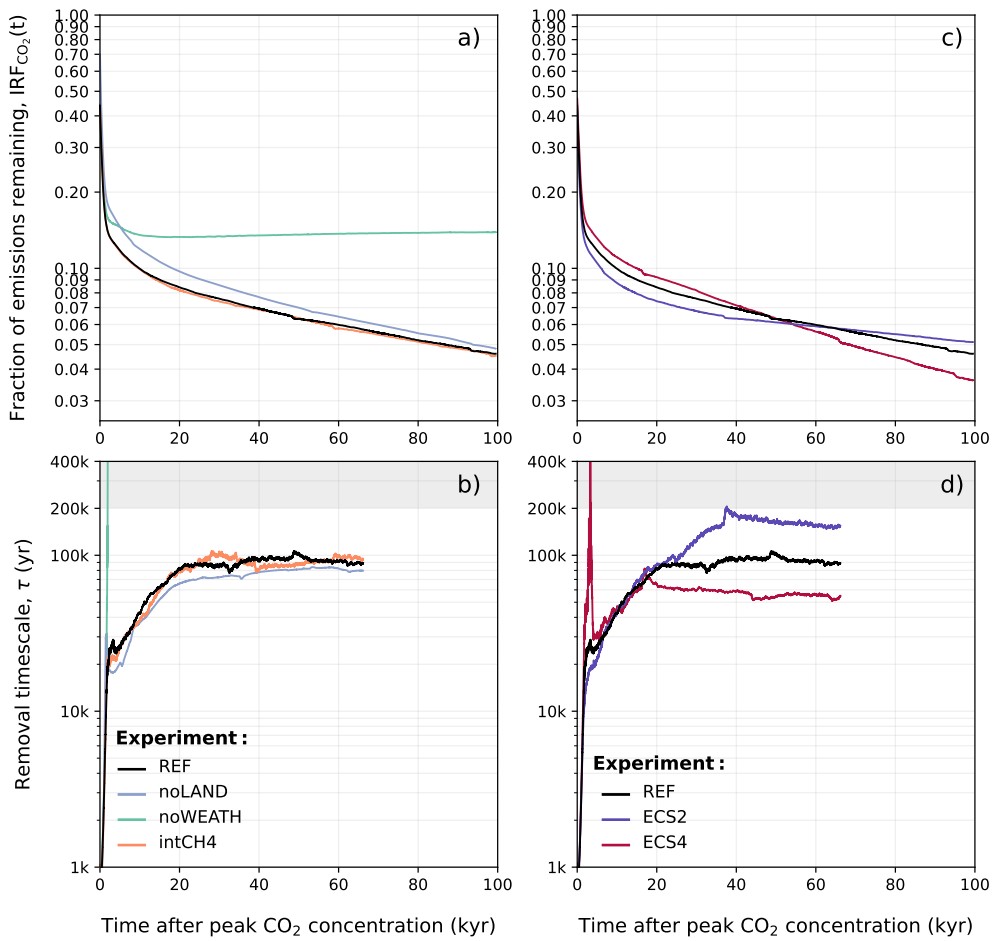

**Figure 17. Comparison of (a,c) the atmospheric lifetime of anthropogenic** $CO_2$ **described by** $IRF_{CO_2}(t)$ **and (b,d) removal timescale of anthropogenic** $CO_2$ **in the different sensitivity experiments.** The mean of the 500-3000 PgC scenarios is shown. Data in (b,d) has been filtered with the variable moving window discussed in Appendix D for visibility, and is why the estimated removal timescale is not available for the full 100 kyr.



pattern changes as atmospheric $CO_2$ concentrations decline, and does not hold beyond the first 1 kyr. Global mean temperature in the 3000 PgC scenario is up to 0.5 °C higher in the intCH4 experiment, but the difference remains largely negligible in the low emission scenarios. While total vegetation carbon is similar between the REF and intCH4 experiments, soil carbon in the intCH4 experiments is notably lower across the different emission scenarios, causing the land carbon pool to take up less than the REF experiment overall (Fig. 16a-c). Carbon stored in peatland is responsible for some of these differences, as carbon is lost from warming-induced soil respiration. Beyond the first 5 kyr, interactive methane has a ±5 ppm effect on atmospheric $CO_2$ concentration (Fig. 15d). In this way, the lifetime of anthropogenic $CO_2$ and removal timescales in the intCH4 experiments do not differ significantly from the REF experiment over long timescales (Fig. 17a,b), suggesting that the indirect effect of $CH_4$ on the climate (through $CO_2$) is likely negligible over such periods.

## 5   Discussion and conclusions

The impacts of anthropogenic climate change will likely be felt for tens of thousands of years, if not hundreds of thousands of years – long after the anthropogenic emission of $CO_2$. By simulating the long-term response to idealized emission scenarios, we highlighted how processes that remove carbon from the atmosphere change with different levels of cumulative emissions. With this, we have examined the capacity and extent of these processes to take up carbon on different timescales. To some extent, the results of this study align with previous investigations. For example, we have shown that the magnitude of cumulative emissions will predominantly control the long-term response, supporting the path independence found by previous work (Matthews et al., 2009; Zickfeld et al., 2012; Herrington and Zickfeld, 2014). The decline in anthropogenic $CO_2$ over time seen here generally falls within estimates given by the multi-model assessment in Archer et al. (2009a) and other studies: approximately 10% of anthropogenic emissions will persist longer than 10,000 years, and ∼4% beyond 100,000 years. The response of deep ocean processes and $CaCO_3$ reactions, as well as their ability to remove anthropogenic $CO_2$ at different levels of cumulative emissions, also supports earlier investigations (Lord et al., 2015b). However, the extensive sensitivity analysis, along with the higher resolution and complexity of CLIMBER-X, have lead to several new findings:

*The removal timescale for silicate weathering is significantly shorter than previously estimated.* The two methods used in this study provided significantly shorter removal timescales for silicate weathering compared to previous research, providing a best estimate of 80–105 kyrs depending on the emission size. This discrepancy is likely due to differences in climate sensitivity, weathering feedbacks, a spatially explicit weathering scheme and a land carbon pool. Given that it has been ∼20-30 years since the constant 200-400 kyr range was first established for the silicate weathering timescale using lower complexity models (Sundquist, 1991; Archer et al., 1997, 1998), our results suggest that it is time to re-examine this estimate. Considering the progress made in model resolution and process descriptions since the last LTMIP both in our model and others, another intercomparison project could be beneficial.

*The removal timescale for silicate weathering is dependent on the magnitude of cumulative emissions.* A major finding in Lord et al. (2015b) was that the removal ability and timescale of silicate weathering was approximately invariant with respect to cumulative emissions. This is challenged here, as it was shown that both are non-monotonous with increasing emissions.





In both the REF and noLAND experiments, there is an initial shortening of the silicate weathering timescale with increasing cumulative emissions until ∼3000-4000 PgC. However, for the 5000 PgC scenario, it was found (via two methods) that the timescale increases. Whether this trend persists for emissions beyond 5000 PgC remains to be determined. Future research would benefit from exploring the connection between extended AMOC decline and the long-term removal of anthropogenic $CO_2$ via silicate weathering.

*Land carbon has a significant long-term influence on the future evolution of atmospheric* $CO_2$. Despite the variation in vegetation and soil carbon response with increasing emissions, the land is a net carbon sink for the duration of our simulations, and is responsible for storing approximately 4–13% of anthropogenic carbon by the end of the simulation. The land's capacity to absorb anthropogenic $CO_2$ in the simulations buffers higher $CO_2$ concentrations and, therefore, peak global temperatures. As a consequence, the presence of land reduces weathering rates, thereby increasing the removal timescale of silicate weathering. Experiments that exclude land carbon would show a shorter silicate weathering timescale than those that include land carbon. As silicate weathering is the ultimate control on the removal of anthropogenic $CO_2$, the extent to which land buffers peak temperature in a model can significantly influence the atmospheric lifetime of anthropogenic $CO_2$.

*The removal timescale for silicate weathering is significantly influenced by a number of factors.* In this study, we quantified the effects of climate sensitivity, the land carbon pool, weathering feedbacks, and dynamically changing methane concentrations on the atmospheric lifetime of anthropogenic $CO_2$. The sensitivity analysis revealed that climate sensitivity has the most significant influence on the silicate weathering timescale (Fig. 17). However, most factors were generally found to impact the silicate weathering timescale, albeit in inverse ways. Although the land carbon pool is expected to increase the silicate weathering timescale, the higher climate sensitivity, stronger silicate weathering feedbacks, and spatially explicit weathering scheme reduce this timescale in CLIMBER-X, ultimately compensating for this effect. This means that, in theory, the same silicate weathering timescale could be obtained with no land carbon and either weaker weathering feedbacks/lower ECS. This emphasizes the importance of incorporating as many components of the carbon cycle as possible, and increasing model complexity in Earth system models as to more accurately represent the long-term future behavior of atmospheric $CO_2$. Furthermore, the global heterogeneity of carbonate and silicate weathering with increasing emissions (Fig. 11) also showed the importance of highly active weathering regions to the global weathering flux (Section 3.1.3), highlighting the need for spatially explicit schemes. Improving the silicate weathering timescale would require more accurate constraints on both these factors and the global silicate weathering rate, along with a deeper understanding of how weathering responds to increasing global temperatures.

*The methane cycle will likely have a negligible effect on long timescales.* Rising global temperatures will increase methane levels, potentially amplifying positive feedbacks and impacting how Earth system components absorb carbon over time. These experiments show that a dynamic methane cycle in CLIMBER-X can contribute up to an additional ∼25 ppm in atmospheric $CO_2$ concentration. However, this effect is mainly limited to the first millennium after emissions end, and is largely negligible on long timescales. As such, the methane cycle does not contribute to substantially different removal timescales, leading to the same rate of anthropogenic $CO_2$ removal over time as in the REF experiment.





There are several assumptions in our study which should be considered when generalizing the long-term response of the climate-carbon system. Certain aspects of the carbon cycle in CLIMBER-X are simplified or not considered, as current computational restrictions limit the inclusion of all carbon-cycle processes across all timescales. For example, CLIMBER-X does not consider deep-sea methane hydrates or nitrogen limitation on tropical or boreal forests, despite their potential impacts on future carbon uptake (Archer et al., 2009b; Norby et al., 2010). Some general uncertainties and poor empirical constraints should also be considered, as they can inhibit an accurate description of some processes. One such example is the strength of the biological pump in a warming world. Although state-of-the-art Earth system models do not agree on the overall response of marine primary production to increasing $CO_2$ levels (Laufkötter et al., 2015), widespread ocean acidification is still expected to have some influence (Mora et al., 2013). This is not considered in our model, which relies on an extended NPZD (nutrients–phytoplankton–zooplankton–detritus) description, typical for models at this level of complexity. The weathering scheme used in CLIMBER-X is significantly more advanced than in other models of the same class, however, it does not explicitly account for changes in vegetation cover or net primary production. While this could be an important simplification, it is presently unclear to what extent it may affect long-term $CO_2$ dynamics. As 5% of the land area in our simulations is responsible for a third of global carbonate and silicate weathering fluxes, small changes in the controlling parameters in these areas can also significantly impact total weathering fluxes (Hartmann et al., 2009b). These limitations highlight potential directions for further work.

Few studies (Lenton and Britton, 2006; Lord et al., 2015b; Couplet et al., 2024) have looked at idealized emission scenarios with more than ∼6000 PgC corresponding to unconventional fossil fuels. This was not considered here, partially because the potential of unconventional fossil fuels is still speculative, but also because such high levels of cumulative emissions make it difficult to ignore additional changes in the Earth system, such as sea level rise. Simulations in this study were carried out using prescribed present-day ice sheets to maintain consistency with previous research. However, the melting of the Greenland and West Antarctic ice sheets, inevitable in high emission scenarios, can introduce additional feedback mechanisms in the carbon cycle (Wadham et al., 2019). These effects could be significant for the range of emissions considered here. In addition to this, it has been assumed that the pre-industrial global carbon cycle was in equilibrium, such that the silicate weathering feedback will eventually return $CO_2$ concentrations to 280 ppm after an anthropogenic perturbation in the carbon cycle. This assumption can also be explicitly seen in the zero emissions scenario in Fig. 2. While this is a standard assumption for investigations into the long-term response of the climate-carbon system to perturbations in the carbon cycle, it is perhaps not so appropriate for very long time scales considered in this work. The implications of this assumption will be considered in a forthcoming paper. Future work with the CLIMBER-X model will investigate the long-term effects of anthropogenic $CO_2$ on the dynamic evolution of Northern Hemisphere ice sheets.



## Appendix A: Emulation of different climate sensitivities

Previous studies have implemented straightforward ways to scale ECS (e.g., Colbourn et al., 2015). In CLIMBER-X, different climate sensitivities can be achieved by scaling the equivalent $CO_2$ in the long-wave radiation scheme in the following way:

$$CO_2^{eq,ecs} = \exp(\alpha \ln CO_2^{eq} + (1 - \alpha) \ln CO_2^{ref}), \tag{A1}$$

where $\alpha$ is the scaling coefficient, and $CO_2^{ref} = 280$ ppm. The equivalent $CO_2$ concentration for radiation is $CO_2^{eq}$, which also includes the effect of other greenhouse gases (Willeit et al., 2023). It was determined a priori that $\alpha = 0.66$ yielded an ECS of 2 °C, whereas an $\alpha = 1.33$ yielded an ECS of 4 °C. This method has been more extensively applied in Höning et al. (2024) and Kaufhold et al. (2024). A comparison of this ECS scaling technique in CLIMBER-X with state-of-the-art Earth system models
is also examined in Kaufhold et al. (2024).

## Appendix B: Fraction of cumulative fluxes from the atmosphere

The fraction of the different carbon reservoirs which contribute to the removal of anthropogenic $CO_2$ over a span of 100 kyr is assessed using Eq. B3. This is similar to calculations for airborne, land-borne, ocean-borne and sediment-borne fractions (Jones et al., 2013; Vakilifard et al., 2022), as the cumulative carbon fluxes of the atmosphere, land, ocean, and weathering are
710 normalized by cumulative anthropogenic emissions, E (PgC). However, we focus strictly on changes in cumulative carbon flux from the atmosphere relative to the pre-industrial, instead of changes in carbon inventory, as our experiments were conducted with an open carbon cycle. From Eq. 1 we get:

$$F_{anth} = \frac{dC_{atm}}{dt} - F_{lnd} - F_{ocn} + F_{weath} - F_{volc}, \tag{B1}$$

which, integrated in time gives:

$$\int_0^t F_{anth} dt = (C_{atm}(t) - C_{atm}(0)) - \int_0^t F_{lnd} dt - \int_0^t F_{ocn} dt + \int_0^t F_{weath} dt - \int_0^t F_{volc} dt. \tag{B2}$$

As we only consider times after the cessation of anthropogenic emissions (i.e., $\int_0^t F_{anth} dt = E$, total cumulative emissions), the equation above becomes:

$$1 = \frac{1}{E} \left( (C_{atm}(t) - C_{atm}(0)) - \int_0^t F_{lnd} dt - \int_0^t F_{ocn} dt + \int_0^t \left( F_{weath}^{carb} + F_{weath}^{sil} \right) dt - \int_0^t F_{volc} dt \right) \tag{B3}$$

The individual components seen in Fig. 4 are then computed as follows:

$$\Phi_{atm}(t) = \frac{1}{E} (C_{atm}(t) - C_{atm}(0)) \equiv IRF_{CO_2}(t), \tag{B4}$$

$$\Phi_{lnd}(t) = -\frac{1}{E} \int_0^t F_{lnd} dt, \tag{B5}$$



$$\Phi_{\text{ocn}}(t) = -\frac{1}{E}\left(\int_0^t F_{\text{ocn}}dt - \int_0^t F_{\text{weath}}^{\text{carb}}dt\right), \tag{B6}$$

$$\Phi_{\text{weath}}^{\text{sil}}(t) = \frac{1}{E}\left(\int_0^t F_{\text{weath}}^{\text{sil}}dt - \int_0^t F_{\text{volc}}dt\right). \tag{B7}$$

Fluxes (in PgC yr$^{-1}$) are relative to the average flux over the reference simulation to account for model drift. We add and subtract $F_{\text{weath}}^{\text{carb}}$ from $F_{\text{ocn}}$ and $F_{\text{weath}}$ respectively, as carbonate weathering additionally transfers carbon into the ocean and causes ocean outgassing. Volcanic outgassing, $F_{\text{volc}}$, was additionally subtracted from $F_{\text{weath}}$ to isolate silicate weathering consumption.

## Appendix C: Calculation of the Revelle factor

The Revelle factor $\xi$ quantifies the ocean's ability to absorb $CO_2$. It essentially compares the ratio of fractional changes of $CO_2$ by fractional changes in DIC (Sarmiento and Gruber, 2006), making it a useful metric for assessing the sensitivity of our simulations to anthropogenic $CO_2$ perturbations (Broecker et al., 1979). Given that there are competing definitions in literature (see: Egleston et al., 2010), we explicitly calculate the Revelle factor in the following way:

$$\xi = \frac{\partial \ln pCO_2}{\partial \ln DIC} \tag{C1}$$

## Appendix D: Rolling mean with variable window size

One disadvantage in the formulation for $\tau(t)$ in Section 3.4, is that it is sensitive to small-scale variability and noise in $CO_2(t)$. A simple rolling mean is not sufficient to smooth this, as the removal timescale is expected to increase over time, eventually converging to an $\mathcal{O}(10^{5-6})$ timescale which is representative of silicate weathering. For this reason, we use a forward rolling mean to remove noise on both short and long timescales. While a rolling mean is generally defined with a fixed window size, we modified the standard equation to have a variable window size. This can be written as the following:

$$CO_2(t) = \frac{1}{w(t)}\sum_{i=t}^{t+w(t)} CO_2(i), \quad \text{where } CO_2(t_0) = CO_2^{\text{max}} \tag{D1}$$

where $w(t)$ is the window size as a function of time, t is time, $CO_2(t)$ is atmospheric $CO_2$ concentration at time t, and $CO_2^{\text{max}}$ is peak $CO_2$ concentration. Different functions were tested for $w(t)$, but a following simple, linearly increasing window of $w(t) = 0.5t - 70$ provided the best results, as it still preserved the overall trend on multiple timescales. Given data limitations, the rolling window could not extend beyond 100 kyr due to the forward rolling technique. For this reason, the data is cut around approximately $t \approx 67$ kyr, when the window size becomes approximately $w(t) \approx 33$ kyr, and is why data for the full 100 kyr are not shown in both Fig. 14 and 17b,d.



**Appendix E: Supporting figures**

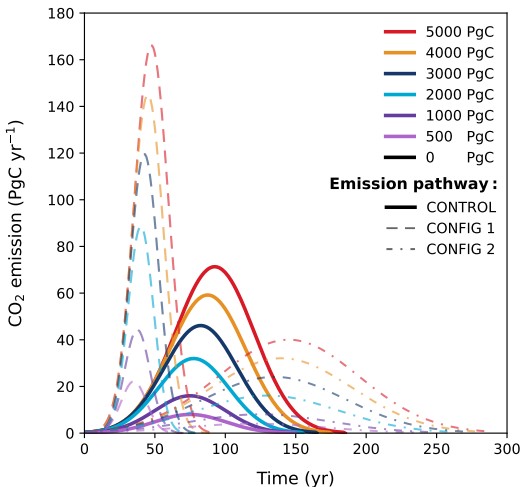

**Figure E1. Pathways of emitted carbon as described by 3 different functions.** The config 1 and 2 functions describe scenarios that effectively emit $CO_2$ faster/slower than the control function.

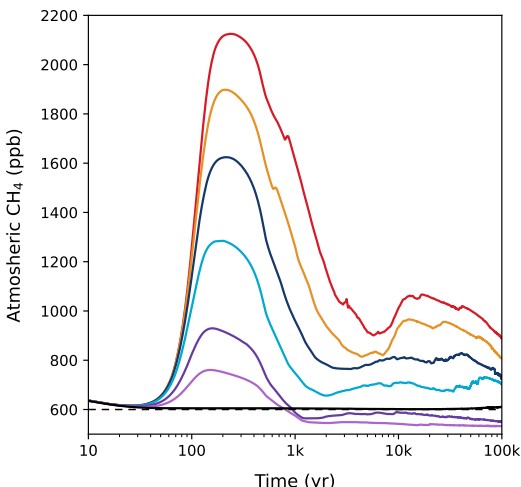

**Figure E2. Atmospheric** $CH_4$ **concentration in the intCH4 experiment.** The black dashed line here shows the constant $CH_4$ concentration used in the REF experiment.



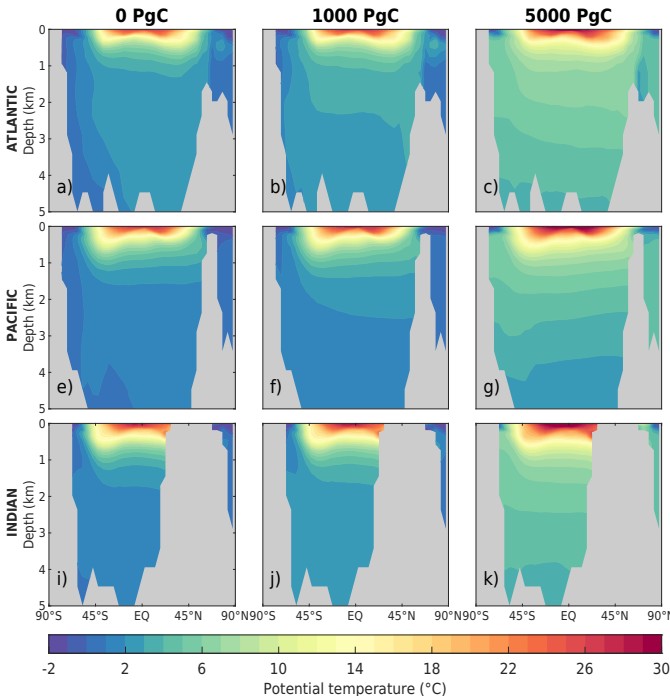

**Figure E3. Annual mean potential temperature in 3 oceanic basins at 1** kyr**.** The top row (a-c) shows the zonal mean cross section of the Atlantic ocean, the middle row (d-f) shows the Pacific ocean, and the bottom row (g-i) shows the Indian ocean.





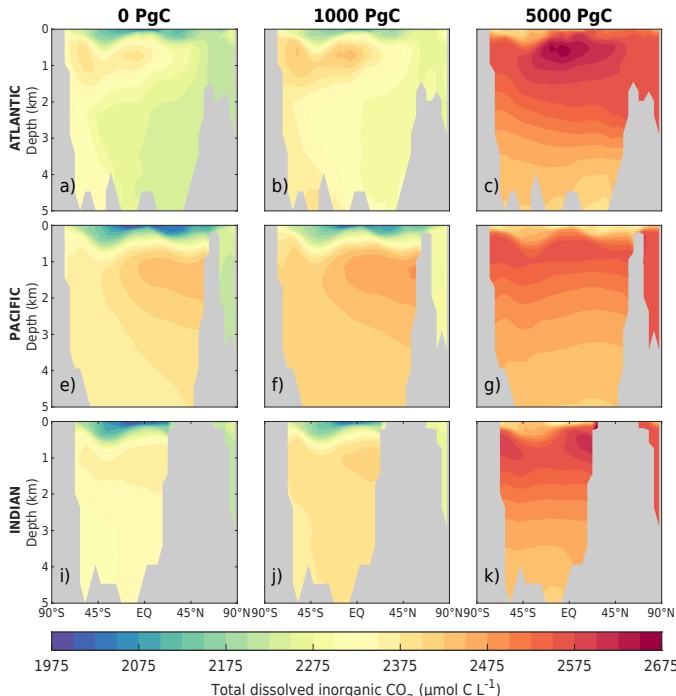

**Figure E4. Annual mean dissolved inorganic** $CO_2$ **in 3 oceanic basins at 1** kyr. The top row (a-c) shows the zonal mean cross section of the Atlantic ocean, the middle row (d-f) shows the Pacific ocean, and the bottom row (g-i) shows the Indian ocean.



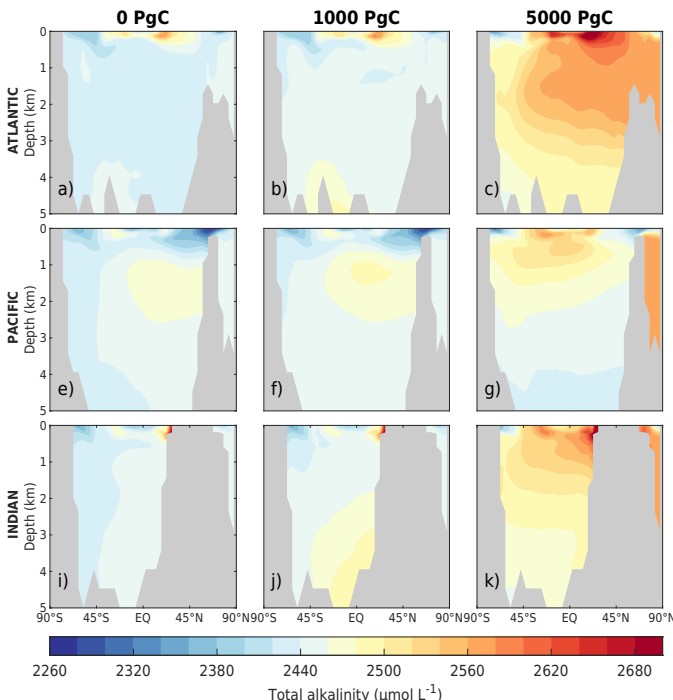

**Figure E5. Annual mean alkalinity in 3 oceanic basins at 1** kyr**.** The top row (a-c) shows the zonal mean cross section of the Atlantic ocean, the middle row (d-f) shows the Pacific ocean, and the bottom row (g-i) shows the Indian ocean.



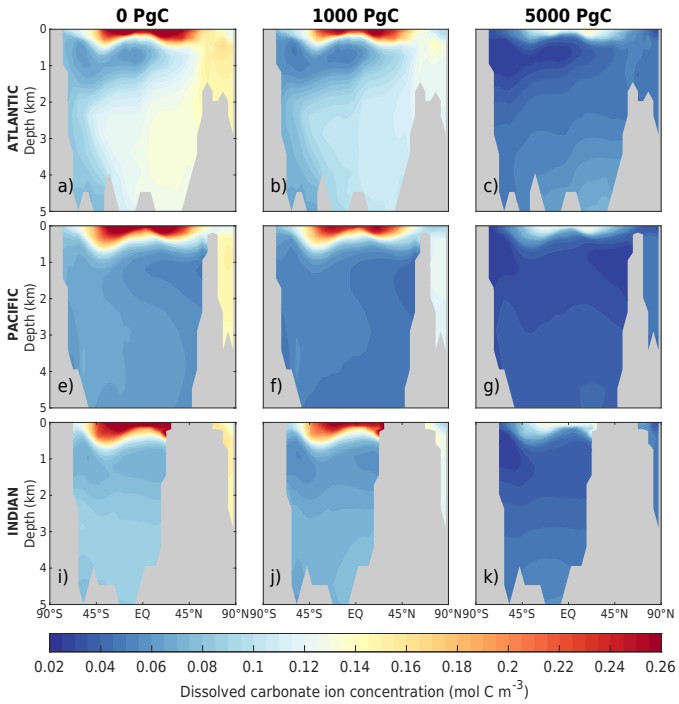

**Figure E6. Annual mean dissolved carbonate ion concentration in 3 oceanic basins at 1** kyr. The top row (a-c) shows the zonal mean cross section of the Atlantic ocean, the middle row (d-f) shows the Pacific ocean, and the bottom row (g-i) shows the Indian ocean.



*Data availability.* Relevant data will be made available on a public data repository (e.g., Zenodo) upon publication.

*Author contributions.* Experimental set-up and design was led by MW and AG. CK conducted the simulations and wrote the manuscript with support from MW, BL, and AG. All authors contributed to the analysis, discussion, and the general quality of the text.

*Competing interests.* The authors declare that they have no conflict of interest.

*Acknowledgements.* CK is funded by the Bundesgesellschaft für Endlagerung through the URS project (research project no. STAFuE-21-4-Klei). MW is supported by the German paleoclimate modelling initiative PalMod (grant nos. 01LP1920B, 01LP1917D, 01LP2305B). BL is supported by the German paleoclimate modelling initiative PalMod (grant nos: 01LP1919B, 01LP2328A). PalMod is part of the Research for Sustainable Development initiative (FONA) funded by the German Federal Ministry of Education and Research (BMBF). The authors gratefully acknowledge the European Regional Development Fund (ERDF), the German Federal Ministry of Education and Research, and

the Land Brandenburg for supporting this project by providing resources on the high-performance computer system at the Potsdam Institute for Climate Impact Research. We additionally thank Victor Couplet for helpful discussions, and for providing SURFER model output.



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
