# Peer review of "Assessing the lifetime of anthropogenic $CO_2$ and its sensitivity to different carbon cycle processes"

_EGUsphere, 2024_

## Author Comment (AC1)

**Response to RC1**

We are grateful to the reviewer for their positive and encouraging assessment of our manuscript. We are delighted that our study was well received and have carefully considered their comments to improve our work. As an overview, and in response to all reviewers, we will address the following aspects in the revised manuscript:

- Shortening the paper and re-organizing our results: We will make considerable effort to shorten our paper (and focus on the novelties of this study), by identifying content in Sections 3.1.1, 3.1.2, 3.2, 3.3, and 3.4 that does not signficantly contribute to our main points. We will move Table 2 to the appendix. We will remove Fig. E2 as it is not referenced in text, and Fig. 14a as it more or less shows the same as 14b-c. We will make Fig. 2 a single column, similar to Fig. 3, 5, 7, 8 and 9. We will re-organize and re-write certain statements as per the reviewers' comments.

- Enhancing the discussion: We will add a PULSE experiment to Sections 3.2-3.3, attribute weathering fluxes to either temperature or runoff in Section 3.1.2, and expand on the potential caveats in our study (related to erosion limitation and tipping points) in Section 5. We will also further discuss the additional positive climate-carbon cycle feedback caused by $CH_4$ in Section 4.4.

- Adding additional figures and tables: We will add subplots to Fig. 13 for $A_i$ vs E and $\tau_i$ vs E, and a figure for temperature evolution in the intCH4 experiment in the appendix. We will also provide a table of lithological values (e.g., activation energy of silicates for the different lithologies), as well as the fitting parameters for the multi-exponential decay fit of the PULSE experiment in the appendix.

- Clarifying certain aspects: We will clarify the treatment of organic carbon in the revised manuscript. Additionally, we will expand on the weathering scheme by including a table of lithological values for Equations 2 and 3, and we will incorporate the equations for carbonate weathering specific to loess and carbonate sedimentary rocks. Furthermore, we will clarify in our figure captions whether time is counted from the beginning of the pulse, or from maximum $CO_2$ concentration.

Please find our point-to-point responses to the individual comments given by Reviewer #1 below (reviewer comment in black, our response in blue).

In their paper 'Assessing the lifetime of anthropogenic CO2 and its sensitivity to different carbon cycle processes' Kaufhold et al. apply the CLIMBER-X Earth system model of intermediate complexity to investigate the atmospheric lifetime and the removal processes of cumulative CO2 emission in a set of 100 kyr experiments.

They include a variety of sensitivity experiments to further investigate the role of the landbiosphere and weathering feedbacks in removing atmospheric CO2 perturbations.

Overall, the paper is well written and provides interesting results and nicely addresses the question of the landbiosphere and weathering feedbacks for the removal of atmospheric CO2 perturbations and provides and a wealth of figures. The spatially explicit weathering scheme of CLIMBER-X is an important addition to the investigation! In summary, the study is well suited for publication in Biogeosciences.

I have three more general aspects and a few minor comments the authors may address during revision.

We would like to thank the reviewer for their positive comments on our paper, and will make improvements where highlighted.

**General:**

1) Presentation of the C-perturbation
In the paper, the authors alternate between presenting the atmospheric C-perturbation (or the perturbation of other reservoirs) in absolute values (i.e. ppm CO2 or PgC) and as fraction of the maximum CO2 perturbation. For comparison with other studies and to address non-linearities it would, in my eyes, be much easier to show results as fractions of the CO2 perturbation rather than as absolute values (for example also in Fig. 1). The fact that the atmospheric CO2 perturbation (in ppm) is larger for larger cumulative emissions is not surprising and it could be interesting to investigate the non-linearities in more detail instead. An alternative could also be to normalize the results by the response to a certain pulse size to highlight the non-linearities (e.g. Fig. 2).
We acknowledge that alternating between absolute values (as in Section 3.1) and fractional values (as in Sections 3.2-3.4) could be confusing, and that expressing some responses as fractions of cumulative emissions may be more interesting for exploring nonlinearities. In principle, we could provide a version of Fig. 1 that shows the change in atmospheric $CO_2$ as a fraction of cumulative emisions. However, the method used to determine atmospheric $CO_2$ concentration from other publications (i.e., visual inspection) is not accurate enough, meaning that the figure is semi-qualitative and mostly serves an illustrative purpose.We will provide a statement on this in the figure caption of the revised manuscript. Furthermore, we are cautious about modifying the current presentation of these results in Fig. 2 (either by normalizing it by cumulative emissions, or by a certain pulse size) as it could make it difficult for future comparisons with our work, especially given the ramp up period. We have aimed to remain consistent with how values have been presented in prior studies. For instance, atmospheric $CO_2$ concentrations have consistently been reported in absolute values (e.g., Archer 1998, Lenton & Britton 2006, Ridgwell & Hargreaves 2007, Archer & Brovkin 2008, Archer et al. 2009, Lord et al. 2015, etc.), and emissions removed or remaining given in such normalized values (e.g., Archer 1997, Eby et al. 2009, Lord et al. 2015, etc.). There are a few exceptions to this, but these are typically made for specific reasons. For instance, Joos et al. (2013) report atmospheric $CO_2$ only as a fraction of emitted emissions, but their study's objective was to compare the response of different models.

2) how emissions are prescribed
The way emissions are prescribed in this study (as Gaussian function) complicates the comparison with studies featuring a pulse-like release of carbon (as often done) to a certain degree. This leads in this study, for example, in the case of small total cumulative emissions, to a large fraction of the atmospheric CO2 perturbation already having been removed before reaching the maximum atm. CO2 perturbation and also to less timescales required when fitting the response as a sum of exponentials (section 3.3) as compared to studies with an instantaneous pulse-like emission. While this is acknowledged in the text, I think it should be made more clear, especially for the discussion of the timescales in sections 3.2-3.4. Further, it might be interesting to add one additional emission pathway sensitivity experiment, where all the carbon is emitted in the first timestep, as done in many of the studies discussed in the

paper. In light of how fast the CLIMBER-X model is (10'000 years per day), this might be doable.

This is a very good point. We agree that this complicates comparisons to other studies, especially as removing the ramp up of emissions may affect the calculated timescales in Section 3.3. We will make the limitations of our analysis (for the REF experiment) more clear going forward. Originally, we chose to not use a pulse-like $CO_2$ perturbation due to concerns about model stability. However, following the reviewer's suggestion, we promptly initiated such experiments for a new emissions pathway ensemble called 'PULSE.' Preliminary results indicate that these experiments behave as expected, with atmospheric $CO_2$ converging to the REF experiment after some time (without any stability issues). At this stage, it is not feasible to revise all results to use the PULSE experiments as the reference. However, assuming no unforeseen issues arise in the PULSE ensemble, we will aim to integrate its results into Sections 3.2–3.3. We also believe this addition will enhance the manuscript by providing greater robustness to the estimated fraction of emissions remaining and timescales.

3) Length of the paper

While the paper does a very nice job in thoroughly describing processes and visualizing a lot in figures, I found it quite lengthy to read. Maybe during revisions this could be kept in mind. For example, in my opinion, sections 3.2-3.4 could be merged and shortened with a focus on the novelties of this study (timescale of the silicate weathering feedback).

We acknowledge that the paper is quite lengthy (as Reviewer #2 also pointed out). We agree that some of the sections mentioned can be shortened and will do so in the revised manuscript. We will also look for other areas that may not significantly contribute to the discussion (e.g., in Section 3.1) as to reduce the overall length. However, we are hesitant to fully merge Sections 3.2–3.4, as we believe maintaining their separation helps preserve structure in the analysis. Furthermore, we believe that our manuscript is stronger when it includes the complete analysis related to determining the atmospheric lifetime of anthropogenic $CO_2$ and its removal timescales. Considering the additional requests for a PULSE experiment, the attribution of weathering fluxes to either temperature or runoff, and an expanded discussion on erosion limitations and tipping elements, it may be challenging to significantly reduce the overall length, however, we will make considerable effort to do so.

**Minor comments:**

- p. 5, l. 105: why is the conservation of phosphate and silicate enforced and how is it done?

As mentioned in the CLIMBER-X description paper (Willeit et al. 2022, 2023), the weathering module includes equations for silicate weathering fluxes. Riverine fluxes of silicate have been disabled in the model set-up here, as they would introduce additional challenges related to nutrient conservation in the ocean. Instead, the budgets for silicate and phosphorus (as well as organic carbon, not mentioned in the manuscript) are balanced by assuming that sediment burial fluxes are returned in remineralized form to the surface ocean. Spatially, these surface fluxes are distributed proportionally to annual runoff. This simplified approach ensures the conservation of phosphorus and silica inventories within the ocean–sediment system throughout the simulation.

We will provide more information clarifying this (as well as that of organic carbon) in the revised manuscript, as this issue was also raised by Reviewer #3.

- p. 6, l. 127-130: please provide values for the parameters
We agree that it would be logical to provide values for α, b, and $E_a$, and will provide these values in the revised manuscript. We will specify in the revised manuscript that i represents the different lithologies to sum over, and explicitly define b(i), $E_{a,sil}$(i), and α(i), and as functions of lithology. We will also add the following table to the Appendix in the revised manuscript for reference:

Table A1: Lithological classes in GLiM (Hartmann & Moosdorf 2012) and their parameters. These lithological classes are summed using the Arrhenius equation. The lithological classes of loess (lo) an carbonate sedimentary rock (sc) are not shown here. The evaporites class (ev) is used only to compute phosphorus fluxes, and is therefore not considered here.

| | Lithological code | Lithological class | Molality / weathering rate, b ((1/12)×molC/kg water) | Activation energy of silicates, $E_{a,sil}$ (kJ/mol) | Fraction to weather as carbonate rocks, α |
|---|---|---|---|---|---|
| 1 | mt | Metamorphics | 0.007626 | 60 | 0.75 |
| 2 | pa | Acid plutonic rocks | 0.005095 | 60 | 0.42 |
| 3 | pb | Basic plutonic rocks | 0.007015 | 50 | 0 |
| 4 | pi | Intermediate plutonic rocks | 0.007015 | 60 | 0.42 |
| 5 | py | Pyroclastics | 0.0061 | 46 | 0 |
| 6 | sm | Mixed sedimentary rocks | 0.012481 | 60 | 0.76 |
| 7 | ss | Siliciclastic sedimentary rocks | 0.005341 | 60 | 0.36 |
| 8 | su | Unconsolidated sediments | 0.003364 | 60 | 0 |
| 9 | va | Acid volcanic rocks | 0.002455 | 60 | 0 |
| 10 | vb | Basic volcanic rocks | 0.007015 | 50 | 0 |
| 11 | vi | Intermediate volcanic rocks | 0.007015 | 50 | 0 |

- p. 7, l. 149ff: looking forward to the interactive ice-sheet simulations!
Thank you for the positive comment, we are also excited to share these simulations in a follow up study.

- p. 7, l. 155: 'pulse' might be a misleading term, maybe replace with 'idealized CO2 emission histories'?
This is a good point, especially with the potential inclusion of the 'PULSE' experiment ensemble. We will change this to 'function' in the revised manuscript.

- section 3.1.3: very nice!
Thank you for the positive comment on this section.

- Fig. 13: maybe clarify in the figure caption as well, why the smaller cumulative emissions lower fractions removed (-> more already taken up by other reservoirs before reaching the max. CO2 perturbation)
We will added a sentence clarifying this, as suggested.

- section 4: I really liked this section!
Thank you for the positive comment on this section.

- Fig. 16: check caption text, not fully clear
We assume that the reviewer is referring to the description of the cumulative stacked barplot in Figure 16. Our intention of this description was to communicate that the barplot is cumulative. By default, plotting a stacked barplot in Python would be non-cumulative, and the length of each bar would represent the magnitude of carbon uptake. For example, as it is shown now, Figure 16a REF shows that land in the 500 PgC scenario takes up ~130 PgC carbon whereas land in the 1000 PgC scenario takes up ~240 PgC. Should the stacked bar plot be non-cumulative, the 500 PgC scenario would still be the same, but the 1000 PgC bar would instead reach ~370 PgC. However, we understand how it is currently written may cause confusion, so we will clarify this by rewriting the second sentence in the Figure caption.

- p. 32, l. 590: 'effect' -> 'affect'?
Thanks, we will change this in the revised manuscript.

- p. 35, l. 646ff: move this part to the other statements about silicate weathering before
Thanks, we will make this change.

**References:**

Archer, D. & Brovkin, V. (2008a). The millennial atmospheric lifetime of anthropogenic $CO_2$. *Climatic Change*, 90(3), 283–297. https://doi.org/10.1007/s10584-008-9413-1

Archer, D., Eby, M., Brovkin, V., Ridgwell, A., Cao, L., Mikolajewicz, U., Caldeira, K., Matsumoto, K., Munhoven, G., Montenegro, A. & Tokos, K. (2009). Atmospheric lifetime of fossil fuel carbon dioxide. *Annual Review of Earth and Planetary Sciences*, 37(1), 117–134. https://doi.org/10.1146/annurev.earth.031208.100206

Archer, D., Kheshgi, H. & Maier-Reimer, E. (1997). Multiple timescales for neutralization of fossil fuel $CO_2$. *Geophysical Research Letters*, 24(4), 405–408. https://doi.org/10.1029/97gl00168

Archer, D., Kheshgi, H. & Maier-Reimer, E. (1998). Dynamics of fossil fuel $CO_2$ neutralization by marine $CaCO_3$. *Global Biogeochemical Cycles*, 12(2), 259–276. https://doi.org/10.1029/98gb00744

Eby, M., Zickfeld, K., Montenegro, A., Archer, D., Meissner, K. J. & Weaver, A. J. (2009). Lifetime of anthropogenic climate change: Millennial time scales of potential $CO_2$ and surface temperature perturbations. *Journal of Climate*, 22(10), 2501–2511. https://doi.org/10.1175/2008jcli2554.1

Hartmann, J. & Moosdorf, N. (2012). The new global lithological map database GLiM: A representation of rock properties at the Earth Surface. *Geochemistry, Geophysics, Geosystems*, 13(12). https://doi.org/10.1029/2012gc004370

Joos, F., Roth, R., Fuglestvedt, J. S., Peters, G. P., Enting, I. G., von Bloh, W., Brovkin, V., Burke, E. J., Eby, M., Edwards, N. R., Friedrich, T., Frölicher, T. L., Halloran, P. R., Holden, P. B., Jones, C., Kleinen, T., Mackenzie, F. T., Matsumoto, K., Meinshausen, M., … Weaver, A. J. (2013). Carbon dioxide and climate impulse response functions for the computation of Greenhouse Gas Metrics: A multi-model analysis. *Atmospheric Chemistry and Physics*, 13(5), 2793–2825. https://doi.org/10.5194/acp-13-2793-2013

Lenton, T. M. & Britton, C. (2006). Enhanced carbonate and silicate weathering accelerates recovery from fossil fuel $CO_2$ perturbations. *Global Biogeochemical Cycles*, 20(3). https://doi.org/10.1029/2005gb002678

Lord, N. S., Ridgwell, A., Thorne, M. C. & Lunt, D. J. (2015). An impulse response function for the "long tail" of excess atmospheric $CO_2$ in an Earth system model. *Global Biogeochemical Cycles*, 30(1), 2–17. https://doi.org/10.1002/2014gb005074

Ridgwell, A. & Hargreaves, J. C. (2007). Regulation of atmospheric $CO_2$ by deep-sea sediments in an Earth system model. *Global Biogeochemical Cycles*, 21(2). https://doi.org/10.1029/2006gb002764

Willeit, M., Ganopolski, A., Robinson, A. & Edwards, N. R. (2022). The Earth System Model CLIMBER-X v1.0 – Part 1: Climate model description and validation. Geoscientific Model Development, 15(14), 5905–5948. https://doi.org/10.5194/gmd-15-5905-2022

Willeit, M., Ilyina, T., Liu, B., Heinze, C., Perrette, M., Heinemann, M., Dalmonech, D., Brovkin, V., Munhoven, G., Börker, J., Hartmann, J., Romero-Mujalli, G. & Ganopolski, A. (2023). The Earth System Model CLIMBER-X v1.0 – Part 2: The global carbon cycle. *Geoscientific Model Development*, 16(12), 3501–3534. https://doi.org/10.5194/gmd-16-3501-2023

---

## Author Comment (AC2)

**Response to RC2**

We sincerely thank the reviewer for their thoughtful and constructive review of our manuscript. We are encouraged that they found merit in our study and confident that incorporating their suggestions will enhance the quality of the work. As an overview, and in response to all reviewers, we will address the following aspects in the revised manuscript:

- Shortening the paper and re-organizing our results: We will make considerable effort to shorten our paper (and focus on the novelties of this study), by identifying content in Sections 3.1.1, 3.1.2, 3.2, 3.3, and 3.4 that does not signficantly contribute to our main points. We will move Table 2 to the appendix. We will remove Fig. E2 as it is not referenced in text, and Fig. 14a as it more or less shows the same as 14b-c. We will make Fig. 2 a single column, similar to Fig. 3, 5, 7, 8 and 9. We will re-organize and re-write certain statements as per the reviewers' comments.

- Enhancing the discussion: We will add a PULSE experiment to Sections 3.2-3.3, attribute weathering fluxes to either temperature or runoff in Section 3.1.2, and expand on the potential caveats in our study (related to erosion limitation and tipping points) in Section 5. We will also further discuss the additional positive climate-carbon cycle feedback caused by $CH_4$ in Section 4.4.

- Adding additional figures and tables: We will add subplots to Fig. 13 for $A_i$ vs E and $\tau_i$ vs E, and a figure for temperature evolution in the intCH4 experiment in the appendix. We will also provide a table of lithological values (e.g., activation energy of silicates for the different lithologies), as well as the fitting parameters for the multi-exponential decay fit of the PULSE experiment in the appendix.

- Clarifying certain aspects: We will clarify the treatment of organic carbon in the revised manuscript. Additionally, we will expand on the weathering scheme by including a table of lithological values for Equations 2 and 3, and we will incorporate the equations for carbonate weathering specific to loess and carbonate sedimentary rocks. Furthermore, we will clarify in our figure captions whether time is counted from the beginning of the simulation, or from maximum $CO_2$ concentration.

Please find our point-to-point responses to the individual comments given by Reviewer #2 below (reviewer comment in black, our response in blue).

This study assesses the lifetime of anthropogenic CO2 and its dependence on total emissions and sensitivity to different carbon cycle processes. The authors use the fast EMIC CLIMBER-X to simulate an ensemble of 100,000-year simulations. They thoroughly analyze which carbon reservoir takes how much carbon and when. Special attention is paid to the timescales of silicate weathering and this study provides a shorter estimate of this timescale compared to previous studies. Next sensitivity experiments are discussed to assess how sensitive the results are to several important processes.
The study is interesting and textually well written. I believe most of the reasoning, as well as the used methodology, are sound. I have a couple of major comments, minor comments and specific comments for the authors to address.
We would like to thank the reviewer for their positive comments on our paper.

**Major comments:**

My first main comment is that the paper is quite long and has a lot of figures. I think the paper would benefit from being a bit shorter. I will leave it to the authors to decide exactly what to change, but I have provided some suggestions below:

We acknowledge that the manuscript is quite lengthy; also noted by Reviewer #1. We agree that some of the sections mentioned can be condensed and will make these adjustments in the revised manuscript. Additionally, we will review other areas, such as parts of Section 3.4, to identify content that may not significantly contribute to the discussion as to reduce the overall length. Considering the additional requests for a PULSE experiment, the attribution of weathering fluxes to either temperature or runoff, and an expanded discussion on erosion limitations and tipping elements, it will be challenging to significantly reduce the overall length, however, we will make considerable effort to do so.

- Section 5 mainly focuses on the weathering processes, and I believe this is also the main novelty in this study. Section 3 is much more elaborate. Is it necessary to, for example, put 3.1.1 and 3.1.2 in the main text or could they also go into the supplementary?

We disagree that weathering processes are the main novelty of our work. The main novelty of our work is that we have carried out long-term simulations with a comprehensive Earth system model, and have analyzed the response of the carbon cycle on different time scales – not just for weathering processes. Therefore, we disagree that Sections 3.1.1 and 3.1.2 do not contribute significantly to the main text. While it is true that the weathering processes and their behavior under different cumulative emission scenarios are a key novelty of our manuscript, the spatially explicit response of the land biosphere is also critical. In previous studies, the land has often been completely omitted (e.g., Archer et al. 1997; Lord et al. 2015; Köhler 2020), simulated with very low complexity and left unreported (e.g., Lenton & Britton 2006), or modelled with similar complexity but not over such long timescales (e.g., Brault et al. 2017). As highlighted by Reviewer #3, the non-monotonous response of soil carbon over long timescales observed in our study may even represent a novel finding. Furthermore, as ocean processes and carbonate chemistry are responsible for the majority of anthropogenic $CO_2$ removal, we also believe it would not be appropriate to completely omit this section. However, we agree that this section in particular could be streamlined and will make considerable efforts to shorten it where possible.

- Is it possible to combine some figures? E.g. Figures 7 and 8, and/or Figures 12 and 13.

We think it would be difficult to combine Fig. 7 and 8 without potentially omitting the response of a few variables. We also think it would be difficult to merge Fig. 12 and 13 considering the proposed changes to Fig. 13 suggested by Reviewer #3. However, we can remove Fig. 14a, as it more or less shows the same as Fig. 14b, and present Fig. 2 in a single column, similar to Fig. 3, 5, 7, 8 and 9.

- In the introduction the authors mention the next glacial cycle. Is it necessary to include this in the introduction?

While it is not necessary to mention the next glacial cycle, we believe it is important in discussing the current state of modelling the long-term future climate evolution. The inclusion of two examples may be excessive, however, so we will omit Lines 29-32.

- Are all figures necessary to go into the main text or can they go into the supplementary? E.g. Figure 1?

We have made full use of the appendix for figures that are not immediately essential at first glance. Indeed, our manuscript has many figures, but we believe our manuscript is stronger when including the responses of the different components (both spatially explicit and timeseries), the associated analyses, and the sensitivity experiments. We are inclined to leave Fig. 1 in the introduction as, like the previous comment, we believe it is important for establishing the current state of the field for long-term future climate modelling.

One thing that remains a bit unclear to me is how the time period where the emissions occur is treated in the analysis. I think it is necessary to mention this in a clear and explicit way. Is the period with emissions included in the 100,000 years? For what analyses is it included, and for what not?

Thank you for the suggestion. We agree that it would be helpful to clarify how the period with emissions is treated in the analysis. For figures showing the time evolution, t=0 corresponds to the beginning of the pulse. However, for all processing and further analysis (i.e., results not focused on the model response to emissions), the ramp-up period of emissions is excluded. We attempted to convey this by noting at the start of certain sections that the analyses were conducted for the time after the peak $CO_2$ concentration (i.e., $CO_2(t_0) = CO_2^{max}$), and to explicitly make a distinction between time after peak $CO_2$ concentration (yr) vs. time (yr) in the figures. In the revised manuscript, we will make sure it is clear that the emission period is part of the simulated 100,000 years and address any places where there could be ambiguities (e.g., caption of Fig. 4). Furthermore, as per Reviewer #1's suggestion, we will also incorporate results from the PULSE ensemble (where all the carbon is emitted in the first timestep), which would avoid complications arising from the emission phase duration.

Following on this comment, I also suggest including in the figures when there are CO2 emissions and/or when there is an atmospheric CO2 maximum. I think this will make the interpretation of the figures next to the text easier.

Thanks for the suggestion. We will make sure that this is explained in the figure captions, but will also try to find a way to emphasize this more clearly in the figure itself.

**Minor comments:**

Abstract: Is it possible to add how the range relates to the cumulative emission range?

We assume the reviewer is referring to the percentage range presented in the abstract that indicates land carbon uptake over 100 kyr. We will add a sentence linking this to the cumulative emission range in the revised manuscript.

Line 51: this statement misses a reference.
We will add a reference to Archer et al. (2009), which corroborates our statement here.

Line 105, 106: and through emissions.
That's a good point, thanks. We will add that as suggested.

Line 118, 119: I suggest mentioning here explicitly what processes the weathering scheme depends on. Since the weathering plays a major role in the manuscript, I think it is important that the weathering scheme is as explicit and clear as possible which also makes it easier to compare it to previous studies using different weathering schemes.
This has already been explicitly mentioned on Page 5, Line 116-117 of the manuscript: "PALADYN includes a rock weathering scheme influenced by runoff and temperature (Hartmann, 2009a; Börker et al., 2020), accounting for 16 different lithologies as described in Hartmann & Moosdorf (2012)." However, as per Reviewer #3's recommendation, we will also explicitly include the equations for loess and carbonate sedimentary rock weathering in Equation 2, and as per Reviewer #1's recommendation, we will add a table focused on weathering parameters for the different lithologies. We will also correct the "16 lithologies" as this was erroneous and should have been 13. This will help further clarify our weathering scheme.

Line 138: where do the numbers come from? Is there a source?
The reviewer here is asking where the values given for volcanic outgassing (0.0738 PgC yr$^{-1}$ or 6.15 TmolC yr$^{-1}$) come from. We set volcanic outgassing to half the global silicate weathering rate at the pre-industrial time (this is already mentioned in the manuscript on Page 6, Lines 137-138). The numbers come from the 100,000 year equilibrium spin-up of the carbon cycle model as described in Willeit et al. (2023), which is mentioned a few lines later (Page 6, Lines 143-144). This value is not based on observations (although it is consistent with the range of observational estimates), but is determined from the condition ensuring that the carbon cycle model is in equilibrium at the pre-industrial climate state, which requires that volcanic outgassing is half the rate of simulated silicate weathering under pre-industrial climate conditions (Munhoven & François 1994; Willeit et al. 2023).

Line 185: I do not see a large response in atmospheric CO2 concentrations, whereas the response in temperature and land carbon are much clearer. I interpret this as that it is not the CO2 concentrations that cause the warming. What does cause this warming?
Many thanks for bringing this to our attention. Reviewer #3 also highlighted this issue. Upon re-evaluating the data, we recognize that this statement was erroneous. Our analysis suggests that the temperature stabilization observed in the 5000 PgC scenario during the first millennium is not driven by the release of soil carbon into the atmosphere, but rather by ocean dynamics and the AMOC. The extended decline in AMOC results in a cooling in the Northern Hemisphere which prevents global mean temperature (GMT) from rising after year ~150. After some time, this cooling is offset by Southern Hemisphere warming via the bipolar seesaw, and explains why GMT stabilizes during the better part of the first millennium. This behaviour continues until the abrupt AMOC recovery, which triggers a rapid increase in GMT (and there is a small bump in global temperature at this time). The role

of AMOC on temperature, rather than $CO_2$ (radiative forcing, $\log(CO_2)$) is demonstrated in Fig. A1 of the "Additional material" section at the end of this document. We will revise this statement accordingly in the updated manuscript.

2d, e are not referenced.
While Fig. 2d is already referenced in Section 3.1.2, Page 12, Line 229, we acknowledge that Fig. 2e was indeed not referenced in the text. We will reference Fig. 2e in the revised manuscript.

4 is referenced before Fig. 3.
Thanks, we will switch these two.

Section 3.1: The sediments are not treated as explicitly as the other reservoirs. Is this for a reason?
Section 3.1 largely introduces the general response of our experiments and the relative partition of anthropogenic carbon into the different reservoirs. The reviewer here asks why we did not treat sediments as their own reservoir.
This was intentionally done here as we performed our experiments with an open carbon cycle. As mentioned in the Fig. 3 caption (and in Appendix B), we focus on changes in cumulative carbon flux from the atmosphere relative to the pre-industrial, instead of changes in carbon inventory (as typically done), as a way to get around this issue. Given that there is no direct air–sediment flux, any anthropogenic $CO_2$ absorbed by sediments must first go through the ocean. This is why we conglomerated the two as one reservoir.

Line 214: Is it possible to give a one or two sentence summary of Kaufhold et al. (2024) here?
We will add more information about Kaufhold et al. (2024) here.

Line 219: Fig. 4a is referenced, but there is not explicit treatment of soil carbon in Fig. 4a. Is the right figure referenced?
Thanks, instead of "soil carbon", this should have referred to "land carbon" inventory, as it explains how land carbon in low emission scenarios is prevented from declining as quickly as in high-emission scenarios, thanks to the sustained levels of soil carbon stocks.

Figure 7a and b show more or less the same thing. Is it necessary to show them both?
Indeed, the Revelle factor and pH are related to ocean chemistry and $CO_2$ absorption, but they focus on different aspects of the carbon cycle and acidification process. The Revelle factor indicates how easily the ocean can absorb $CO_2$, relative to the relative increase in atmospheric $CO_2$. Surface ocean pH indicates the acidity of sea water, which is affected by the actual amount of $CO_2$ absorbed. Although the Revelle factor is a critically important metric for communicating buffering capacity, it is unfortunately often not reported in such long-term studies on the uptake of anthropogenic $CO_2$ (both in terms of peak magnitude and duration across the different emission scenarios). We thought that calculating this explicitly might be useful for future studies to reference.

Line 313: Is it possible to determine how much of the changes in weathering rates are because of changes in temperature, and how much due to changes in run off? I think this would make for a nice addition.

This is something which we had already considered, but did not attempt as the manuscript was already quite long at that point. However, we agree that it would be a nice addition, and we will quantify the relative contribution of temperature and runoff to changes in the weathering rate in the revised manuscript.

Line 315-317: I do not fully understand this sentence.

The reviewer is referring to the following sentence: "For carbonate weathering, large changes are not only limited to the equatorial regions, although the highest and lowest weathering rates in South East Asia and Central Asia can also be explained by increases and decreases in precipitation (Fig. 11b,c)".

Here, we mean that large increases in carbonate weathering with increasing emissions is not only limited to the equatorial regions (as with silicate weathering), but it is more globally distributed. However, the change in carbonate weathering (where it increases and decreases) can also be explained by precipitation. We will clarify this in the revised manuscript.

Section 4.1: I suggest mentioning the noLAND term earlier.

Thanks for the suggestion. Reviewer #3 also raised a similar point. We will introduce the term in the experimental section of the text, remind the reader that this refers to the experiment with land carbon disabled, and include a reference to the experiment table.

Figure 16: Would it be beneficial to also construct panels for weathering?

We considered this as well but ultimately decided against it, as weathering is not a true carbon pool like the others (land, ocean, and sediments). Fig. 16 effectively shows the magnitude of carbon uptake of the different pools across the different emission scenarios at different timeslices. The main novelty here is to distinguish how the different sensitivity experiments change the magnitude of carbon uptake. As weathering consumes more $CO_2$ over time, we decided that it would not provide additional information, as it would only increase in relative magnitude between 1 kyr and 100 kyr.

Line 592: Is permafrost treated in the methane model?

Permafrost is implicitly treated in the methane model through the representation of anaerobic decomposition in saturated soils (which can occur in permafrost regions). In these areas, carbon in the active layer may decompose under anaerobic conditions, leading to methane emissions. However, methane emissions from permafrost regions are generally small compared to emissions from low-latitude wetlands.

Line 597: How does temperature evolve in intCH4 compared to REF as there is quite a strong increase in CH4 concentrations?

We initially included a more detailed discussion on this topic but later condensed it to maintain the manuscript's conciseness. In general, interactive methane increases peak temperature by up to approximately 0.5 °C, with higher emission scenarios exhibiting greater differences in peak temperature compared to the REF experiment. This is already noted on Page 34, Lines 600–602. However, temperatures in the intCH4 experiments tend to converge

to the REF experiments around 1 kyr (similar to $CO_2$ concentrations, Fig. 15). While there are some differences in simulated temperature, the intCH4 and REF experiments largely follow each other over time. At what time temperature is larger/smaller in the intCH4 experiment compared to the REF experiment is primarily influenced by the non-monotonic response of soil carbon to increasing emissions (see Fig. A2-A4 in the "Additional material" section at the end of this document). We will add a figure to the appendix of our revised manuscript showing the temperature evolution in the intCH4 experiment.

Line 674: I think it is good to explicitly mention that ESMs do not agree on centennial timescales.
Thanks, we will change this as suggested.

I suggest adding a discussion on how tipping points (might) affect the estimation of the timescales. The AMOC is already mentioned a couple of times in the text, but I think it would be good to reiterate that, and other tipping points, in Section 5.
Potential tipping points are inherently accounted for in our simulations, as CLIMBER-X already incorporates all "fast" tipping elements (e.g., AMOC, sea ice, permafrost, boreal forests, and Amazonian forests, etc.). The only exception to this are ice sheets, as the Greenland and Antarctic ice sheets are prescribed by their present day configurations in our study. However, we know that the Antarctic ice sheet will not entirely melt in our scenarios (Winkelmann et al. 2015). Only the West Antarctic Ice Sheet will (likely) melt, though most of its area lies over the ocean. As the impact of its melting on the AMOC remains uncertain (Wunderling et al. 2024), it is difficult to predict how this might affect the removal timescale of ocean invasion, and its subsequent influence on other timescales. On the other hand, the Greenland Ice Sheet (GrIS) is projected to melt under the strongest scenario, which could influence weathering rates. This effect has been previously investigated using CLIMBER-2 (Munhoven et al. 2007), but it was shown to be nearly negligible.
Without experiments explicitly simulating the crossing of tipping points (and all analyses therein), the influence of these events on the removal timescales of anthropogenic $CO_2$ cannot be assessed and remains highly uncertain. However, we can provide a statement acknowledging this uncertainty, and noting that the crossing of such tipping elements (although not seen in our experiments) might affect the long-term capacity different components to absorb carbon over time.

**Specific comments:**

Line 61, 62: It is not obvious how non-linear translates to the exponential functions. They are linked here through the word 'therefore', suggesting an obvious connection. I suggest either explaining why non-linear means exponential in this case or rewriting the second sentence a bit.
We will rewrite the second sentence to the following: "The decline in anthropogenic $CO_2$ is usually presented as a superposition of exponential decays (Maier-Reimer and Hasselmann, 1987; Archer et al., 1997; Archer and Brovkin, 2008; Colbourn et al., 2015; Lord et al., 2015b), with each function representing a different process in the carbon cycle that takes up carbon.".

Line 109: I suggest mentioning the units of Catm (i.e. PgC).
Thanks for the suggestion, we will add the unit for Catm as recommended.

Line 194: 'At peak CO2 concentrations, …'
We will change it as suggested.

Line 493: the double fraction does not look so nice in the text.
This is a good point. We will change this to $\tau(t) \propto \left(\frac{dCO_2(t)}{dt}\right)^{-1}$.

Line 532: remaining where? In the atmosphere?
Thanks for pointing this out. You are right that we did not clarify, and that it should indeed be
"…leads to higher fraction of emissions remaining [in the atmosphere]". We will change this
in the revised manuscript.

Line 566: I suggest rewriting this sentence a bit. I first thought that it meant that if a
simulation has lower CO2 concentrations, it has a lower ECS.
We apologize for the miscommunication. We will change "a lower ECS is associated with
lower atmospheric concentrations of $CO_2$ (and vice versa)" to "simulations using a lower
ECS produce lower atmospheric concentrations of $CO_2$ (and vice versa)" in the revised
manuscript.

Line 642: 'the presence of land' feels a bit awkward here.
We will change "the presence of land…" to "the inclusion of land carbon cycle processes
effectively…".

Fig. A1: Role of radiative forcing and AMOC on the evolution of global mean surface temperature. Trajectories have been plotted for the entire 100,000 years.

[Figure]

Fig. A2: Change in global mean surface temperature in the 0-3000 PgC emission scenarios. Colours here correspond to the cumulative emission scenarios shown in Fig. 2 of the manuscript. The response in temperature is shown here for two experiments: solid line for the intCH4 experiment and dashed lines for the REF experiment.

[Figure]

Fig. A3: Change in vegetation carbon inventory in the 0-3000 PgC emission scenarios. Colours here correspond to the cumulative emission scenarios shown in Fig. 2 of the manuscript. The response in vegetation carbon is shown here for two experiments: solid line for the intCH4 experiment and dashed lines for the REF experiment.

[Figure]

Fig. A4: Change in soil carbon inventory in the 0-3000 PgC emission scenarios. Colours here correspond to the cumulative emission scenarios shown in Fig. 2 of the manuscript. The response in soil carbon is shown here for two experiments: solid line for the intCH4 experiment and dashed lines for the REF experiment.

[Figure]

**References:**

Archer, D., Eby, M., Brovkin, V., Ridgwell, A., Cao, L., Mikolajewicz, U., Caldeira, K., Matsumoto, K., Munhoven, G., Montenegro, A. & Tokos, K. (2009). Atmospheric lifetime of fossil fuel carbon dioxide. *Annual Review of Earth and Planetary Sciences*, 37(1), 117–134. https://doi.org/10.1146/annurev.earth.031208.100206

Archer, D., Kheshgi, H. & Maier-Reimer, E. (1997). Multiple timescales for neutralization of fossil fuel $CO_2$. *Geophysical Research Letters*, 24(4), 405–408. https://doi.org/10.1029/97gl00168

Brault, M.-O., Matthews, H. D. & Mysak, L. A. (2017). The importance of terrestrial weathering changes in multimillennial recovery of the global carbon cycle: A two-dimensional perspective. *Earth System Dynamics*, 8(2), 455–475. https://doi.org/10.5194/esd-8-455-2017

Köhler, P. (2020). Anthropogenic $CO_2$ of high emission scenario compensated after 3500 years of ocean alkalinization with an annually constant dissolution of 5 Pg of Olivine. *Frontiers in Climate*, 2. https://doi.org/10.3389/fclim.2020.575744

Lenton, T. M. & Britton, C. (2006). Enhanced carbonate and silicate weathering accelerates recovery from fossil fuel $CO_2$ perturbations. *Global Biogeochemical Cycles*, 20(3). https://doi.org/10.1029/2005gb002678

Lord, N. S., Ridgwell, A., Thorne, M. C. & Lunt, D. J. (2015). An impulse response function for the "long tail" of excess atmospheric $CO_2$ in an Earth system model. *Global Biogeochemical Cycles*, 30(1), 2–17. https://doi.org/10.1002/2014gb005074

Munhoven, G. & François, L.M. (1994). Glacial-Interglacial Changes in Continental Weathering: Possible Implications for Atmospheric CO2 . In: Zahn, R., Pedersen, T.F., Kaminski, M.A., Labeyrie, L. (eds) Carbon Cycling in the Glacial Ocean: Constraints on the Ocean's Role in Global Change. NATO ASI Series, vol 17. Springer, Berlin, Heidelberg. https://doi.org/10.1007/978-3-642-78737-9_3

Munhoven, G., Brovkin, V., Ganopolski, A. & Archer, D. (2007). Impact of future Greenland deglaciation on global weathering fluxes and atmospheric $CO_2$ [Paper presentation]. 17th V. M. Goldschmidt Conference 2007, Cologne, Germany.

Winkelmann, R., Levermann, A., Ridgwell, A. & Caldeira, K. (2015). Combustion of available fossil fuel resources sufficient to eliminate the Antarctic Ice Sheet. *Science Advances*, 1(8). https://doi.org/10.1126/sciadv.1500589

Wunderling, N., von der Heydt, A. S., Aksenov, Y., Barker, S., Bastiaansen, R., Brovkin, V., Brunetti, M., Couplet, V., Kleinen, T., Lear, C. H., Lohmann, J., Roman-Cuesta, R. M., Sinet, S., Swingedouw, D., Winkelmann, R., Anand, P., Barichivich, J., Bathiany, S., Baudena, M., Bruun, J. T., Chiessi, C. M., Coxall, H. K., Docquier, D., Donges, J. F., Falkena, S. K. J., Klose, A. K., Obura, D., Rocha, J., Rynders, S., Steinert, N. J. & Willeit, M. (2024). Climate tipping point interactions and cascades: A Review. *Earth System Dynamics*, 15(1), 41–74. https://doi.org/10.5194/esd-15-41-2024

---

## Author Comment (AC3)

**Response to RC3**

We appreciate the reviewer's positive assessment of our manuscript and are pleased that they found the study valuable. We thank Pierre Maffre for his detailed and constructive feedback, which has certainly strengthened our work. As an overview, and in response to all reviewers, we will address the following aspects in the revised manuscript:

- Shortening the paper and re-organizing our results: We will make considerable effort to shorten our paper (and focus on the novelties of this study), by identifying content in Sections 3.1.1, 3.1.2, 3.2, 3.3, and 3.4 that does not signficantly contribute to our main points. We will move Table 2 to the appendix. We will remove Fig. E2 as it is not referenced in text, and Fig. 14a as it more or less shows the same as 14b-c. We will make Fig. 2 a single column, similar to Fig. 3, 5, 7, 8 and 9. We will re-organize and re-write certain statements as per the reviewers' comments.
- Enhancing the discussion: We will add a PULSE experiment to Sections 3.2-3.3, attribute weathering fluxes to either temperature or runoff in Section 3.1.2, and expand on the potential caveats in our study (related to erosion limitation and tipping points) in Section 5. We will also further discuss the additional positive climate-carbon cycle feedback caused by $CH_4$ in Section 4.4.
- Adding additional figures and tables: We will add subplots to Fig. 13 for $A_i$ vs E and $\tau_i$ vs E, and a figure for temperature evolution in the intCH4 experiment in the appendix. We will also provide a table of lithological values (e.g., activation energy of silicates for the different lithologies), as well as the fitting parameters for the multi-exponential decay fit of the PULSE experiment in the appendix.
- Clarifying certain aspects: We will clarify the treatment of organic carbon in the revised manuscript. Additionally, we will expand on the weathering scheme by including a table of lithological values for Equations 2 and 3, and we will incorporate the equations for carbonate weathering specific to loess and carbonate sedimentary rocks. Furthermore, we will clarify in our figure captions whether time is counted from the beginning of the simulation, or from maximum $CO_2$ concentration.

Please find our point-to-point responses to the individual comments given by Reviewer #3 below (reviewer comment in black, our response in blue).

Kaufhold et al. manuscript addresses the question of the fate of anthropogenic CO2 and climate in the long-term future (100 thousand years). The authors used an Earth system model of intermediate complexity, which has several new implemented processes compared to previous similar studies. They clearly explain the novelties of their study, and the new findings. The manuscript is very well written, and well organized. It is well-suited for publication in Biogeosciences, with some minor revisions.
We would like to thank the reviewer for their positive comments on our paper, and will make improvements where highlighted.

**Major comments:**

I only have one major comment, which concerns the silicate weathering sensitivity to climate.

The authors emphasize their re-estimation of the timescale of carbon removal by silicate weathering, to shorter values than previously thought. They partly attribute this finding to a stronger weathering feedback, which is compared to several estimations (Fig. 10b) and found to fall within the range, though on the upper part (doubling of weathering flux at +4°C, that is +18% per °C of warming).

Among the processes not represented in CLIMBER-X weathering model is the erosion limitation of weathering, or the "soil shielding effect", which is a different point of view of the exact same process. Soil shielding was extensively discussed in Hartmann et al. (2014) (cited in the manuscript), but wasn't yet implemented in Hartmann et al. (2009). Actually, soil shielding is not explicitly represented in any of the model presented in Fig. 10b.

I admit that there is no consensus on how this would affect the sensitivity of weathering to global climate (i.e., the weathering feedback strength), which is the point of interest here. Yet, there are several clues that it would significantly reduce the feedback strength:

- Godderis et al., Geoderma, 2008 (10.1016/j.geoderma.2008.01.020) showed that the sensitivity of tropical weathering to runoff is largely overestimated (~ 5-fold) if considered similar than for temperature climates. Indeed, in the present manuscript, tropical environments dominate the weathering flux, and its response to global warming.
- Maher & Chamberlain, Science, 2014 (10.1126/science.1250770), who also addressed the issue of erosion limitation, suggested a "maximal" weathering sensitivity, in actively eroding mountains, of +5% per °C of warming (which is lowest estimate presented on current Fig. 10b), and an average sensitivity of +1.2% per °C of warming.
- Another weathering model taking into account erosion limitation, and that is spatially explicit, Maffre et al., Clim. Past, 2023 (10.5194/cp-19-1461-2023), suggests a global weathering sensitivity of ~ +9% per °C of warming, though it is unclear if the best fitting functional form should be exponential or linear.

Given the absence of consensus on a value for weathering sensitivity, I do not consider that the present results should be revisited. Simply, I vividly recommend the authors to add more nuances on their statement about weathering timescale (which is one of their main conclusions), and to provide more discussion about weathering sensitivity, the large uncertainty that exists in the literature concerning its value, and how it should affect the weathering timescale.

We are aware of the soil shielding effect, and it was commented on in the CLIMBER-X carbon cycle description paper (Willeit et al., 2023): "The effect of soil shielding on the weathering rate suggested by Hartmann et al. (2014) has not been considered since information on soil shielding is not readily available for periods beyond the recent past."

As the reviewer correctly identifies, the effect of soil shielding has not been considered by our model (and others) largely because there is no consensus on how it would effect the weathering feedback. However, we do not dismiss the possibility it could significantly change –and potentially weaken— the strength of the weathering feedback. In saying this, we will add a paragraph discussing this potential caveat. We also appreciate the compilation of references; they will be added to our manuscript and will give considerable depth to our discussion.

**Specific comments:**

Section 2.2 (lines 100–110): there is a missing information here about the organic carbon cycle. As far as I understand, the sediment component is run as an open system (with sediment loss through burial), and this sediment contains organic carbon generated by marine primary productivity (lines 255–256). Therefore, and given Eq. (1), setting Fvolc to half of the global silicate weathering flux (as indicated lines 136–138) would not result in a steady-state carbon cycle, because of this additional C sink (organic carbon burial), that would result in a net ocean-to-atmosphere flux lower than the remaining term "Fvolc – Fweath". Unless the organic carbon cycle is forced to work as a closed system (like silicate and phosphate, lines 106–107), and all buried organic carbon is put back into the atmosphere?
Many thanks for pointing out this critical issue! We have an open carbon cycle in CLIMBER-X but, indeed, a closed nutrient cycle. We recycle organic carbon in marine sediments along with nutrients, and sediment burial fluxes are returned in remineralized form to the surface ocean while compensating for the subduction of inorganic carbon by volcanic outgassing. Reviewer #1 also raised a similar concern, so we will provided a sentence clarifying the behaviour of phosphorus, silicate, and organic carbon in CLIMBER-X.

Lines 125–126: I do not understand why "carbonate sedimentary rock" should be different than "carbonate", in term of weathering (Eq. 2). Moreover, why not indicating the equations for "carbonate sedimentary rocks" weathering and loess weathering?
Thanks for your comment; we hope that we can clarify this. In Hartmann & Moosdorf (2012), there are three carbonate-rich sedimentary lithologies, which are mixed sedimentary rocks (sm), evaporites (ev), and carbonate sedimentary rocks (sc). The evaporites class (ev) is used only to compute phosphorus fluxes, and is therefore not considered here.
 However, other lithologies still maintain information on carbonate content, such as unconsolidated sediment (su) and metamorphics (mt). When we specify "carbonate sedimentary rocks", we mean that that the contribution of the lithology "sc" to carbonate weathering rates in a grid cell is not calculated using an Arrhenius equation. In addition to the 13 rock lithologies as listed in Table A2 in Hartmann & Moosdorf (2012), we also consider loess (lo) as another lithology. The contribution of "lo" to carbonate weathering rates in a grid cell is similarly not calculated using an Arrhenius equation. We included a table of the different lithologies (Table A1) in the "Additional material" section below, and will add this to the appendix of the revised manuscript.
Therefore, the Equation 2 presented in text is what is used for all other lithologies. As it is now, the "accounting for 16 different lithologies" (Page 6, Line 116) and sum over 14 is erroneous, and it should be 13 and 11 respectively. We agree it would be useful to show the equations that are used for the other two lithologies (loess and carbonate sedimentary rock), which is why we will incorporate them into the revised manuscript.

Lines 132–136: I think that orbital forcings could be mentioned here, among the "external forces" (line 132) excluded in the study, although it may be redundant with line 145.
We had a similar thought, and deliberated which section would be most appropriate for this information. However, we will repeat it here, especially in light of the recommended changes in the following comment.

Lines 145–146: It is not completely clear here whether the fixed orbital forcings concern only the spin-up run, or all simulations (including the spin-up).

On Page 7, Lines 152-153 we state that "All simulations run for 100,000 years with constant orbital parameters and without any climate acceleration technique". We will move the aforementioned Lines 145-146 before Line 152 to highlight that orbital parameters are constant in all simulations.

Lines 169–170: I think it would be useful here just to indicate that climate sensitivity is altered by rescaling the pCO2 seen by the radiative code as a function of the actual pCO2, and then refer to Appendix A.

Thanks for your suggestion. We will change this part as recommended.

Lines 185–186: This statement, "temperatures temporarily stabilize instead of decreasing due to the release of soil carbon into the atmosphere" seems erroneous. Temperature does stabilize during between 150yr and 1000yr in the 5000 PgC scenario (Fig. 2b), but pCO2 declines just as in the other scenarios (Fig. 2a). So how could it be an effect of the "release of soil carbon into the atmosphere"? It rather seems that there is a decoupling of CO2 and temperature, that is likely due to oceanic dynamics. Indeed, there is a small bump of global temperature at 700yr (without any pCO2 change), which coincides with abrupt AMOC recovery (Fig. 7e).

Many thanks for pointing this out. This is indeed an erroneous statement. Upon reviewing the data, we agree that the temperature stabilization in the 5000 PgC scenario within the first millennium is not explained by the release of soil carbon into the atmosphere. It does appear that the likely cause is oceanic dynamics and AMOC, as pointed out. The extended decline in AMOC results in a cooling in the Northern Hemisphere which prevents global mean temperature (GMT) from rising after year ~150. After some time, this cooling is offset by Southern Hemisphere warming via the bipolar seesaw, and explains why GMT stabilizes during the better part of the first millennium. This behaviour continues until the abrupt AMOC recovery, which triggers a rapid increase in GMT (and there is indeed the small bump in global temperature at this time). The role of AMOC on temperature, rather than $CO_2$ (radiative forcing, $\log(CO_2)$) is demonstrated in Fig. A1 of the "Additional material" section at the end of this document. We will revise this statement accordingly in the updated manuscript.

Lines 215–221: this non-monotonous behavior is interesting. Has it been already suggested, or is it a new finding of current study?

To the best of our knowledge, this has not yet been explicitly observed in a previous study on the long-term effects of anthropogenic $CO_2$. This is mostly because land carbon was often not considered (or the response unreported, as in Lenton & Britton 2006). However, we are not prepared to conclude that this is necessarily a new finding (e.g., a strong positive climate-carbon cycle feedback related to soil respiration has already been highlighted in studies such as Cox et al. (2000)). On a global level, the response of soil carbon to increasing emissions is generally dictated by (1) that which is gained from increases in primary production and litterfall, and (2) that which is lost from higher soil respiration, influenced by different competing feedbacks.

Line 222: This statement, "In our simulations, the land is a net carbon sink for the entire 100 kyr" also seems erroneous. From Fig. 3a, it appears that land becomes a (slight) net source of carbon at 200kyr in all simulations. Besides, I don't think that "land carbon" is defined anywhere in the manuscript. Is it simply "soil + vegetation" carbon?

Thanks for bringing this to our attention. We assume the reviewer here means 200 years, not 200 kyrs. Indeed this is an erroneous statement and it will be corrected to "the terrestrial storage of anthropogenic carbon is positive during the entire run". We will also make sure to explicitly define land carbon in the manuscript (as the sum of vegetation and soil carbon).

Line 364: The mention of "noLAND" comes quite abruptly here, given that the sensitivity experiments are only discussed in a later section (4). Could you remind "experiment with land carbon disabled", and refer to Table 1?

Thanks for the suggestion. Reviewer #2 also raised a similar point. We will introduce the term earlier in the experimental section of the text, clarify that this refers to the experiment with land carbon disabled, and include a reference to the experiment table.

Fig 13: It is difficult to visualize the trends of Ai and τi versus cumulative emission (trends that are discussed in the current section). I suggest adding two small panels in the figure, plotting Ai vs E and τi versus E.

That's a good idea, we will add two subplots for $A_i$ vs E and $\tau_i$ vs E in Figure 13 (see Fig. A2 in the "Additional material" section at the end of this document).

Lines 371–377: It might be useful to indicate here that Ai do not sum at 1 because the IRF does not start at 1, and that the initial value (= the sum of Ai) depends on the cumulative emission scenario.

Thank you for the suggestion. To some extent, this information was included in the Table 2 caption, but we will integrate it into the main text to ensure its visibility in the revised manuscript.

Lines 565–572: It seems that there is a positive feedback here: warmer temperature (for a same pCO2) generates higher pCO2, because of the warming-induced soil carbon release. It would be useful to indicate that it is a positive feedback.

We will add this to the revised manuscript.

Line 594: Is methane lost by converting it into CO2? Granted that 2200 ppb of methane should not generates more than 2 ppm of CO2, with is much less than the pCO2 anomaly reported in Fig. 15d.

Firstly, we would like to clarify that 2200 ppb is peak $CH_4$ concentration in the 5000 PgC scenario, whereas the sensitivity analysis in Section 4 is limited to 3000 PgC and less. This, of course, does not answer the reviewer's question, as peak $CH_4$ of 1600 in the 3000 PgC scenario alone cannot explain an additional 25 ppm of atmospheric $CO_2$.

Methane is oxidized assuming a constant lifetime of 9.5 years (Willeit et al. 2023). In reality, this results in $CH_4$ being converted into $CO_2$ in the atmosphere, but this flux is small. For simplicity (and for carbon conservation), we add carbon from methane to surface $CO_2$ flux in CLIMBER-X (e.g., soil $CO_2$ emission).

The reason why $CO_2$ is higher in the intCH4 experiments is because, as mentioned in the previous comment, $CH_4$ causes an additional positive climate-carbon cycle feedback (as additional warming enhances soil respiration; see Fig. A3-A4 in the "Additional material" section at the end of this document).
We originally had a larger discussion on this, but it was cut in our efforts to (already) shorten the paper. However, Reviewer #2 had a similar question, asking how temperatures evolve in the intCH4 experiments compared to the REF experiments (as a result of this large increase in $CH_4$ concentration), so we will elaborate on this more in the revised manuscript.

Lines 644–645: Would it really influence the ATMOSPHERIC lifetime of CO2? It seems to me that the longer weathering timescale is the just a delay because of carbon storage in land before it is stored through weathering, instead of being directly stored though weathering, and that this sink transfer has no consequence regarding carbon in the atmosphere.
You make a good point, and it is true that the longer (effective) weathering timescale is caused by the temporary storage of carbon on land. However, the land carbon pool on its own does increase the residence time of anthropogenic $CO_2$, as the land stores about 20-40% of anthropogenic carbon (Page 11, Fig. 4) before gradually releasing it into the atmosphere. This ultimately slows down the $CO_2$ decline on long timescales (Fig. 17a).

**Technical corrections:**

There are several occurrences where it should be more accurate to talk about weathering "flux", than weathering "rate", which rather refers to a specific flux (in mol/m2/yr): line 138, line 281, caption of Fig. 9, line 291, line 301...
Thanks, we will change the word "rate" to "flux" where appropriate.

Line 130: It seems that "run-off" should be spelled "runoff", to be consistent with the other occurrences of that word in the manuscript.
Thanks for pointing this out. It will be changed to "runoff" in the revised manuscript.

Caption of Fig. 6: The mean net annual NPP is in (a–c), not (a–b).
Thanks, this will be corrected.

Fig. 11: A mere suggestion: it feels more "natural" to use a colorscale with "wetter" colors (e.g., blue) for precipitation increase and "dryer" colors (e.g., red) for precipitation decrease.
Thanks for the suggestion. We will change Figure 11 h-i to a brown–bluegreen colormap, which is often used to indicate "drier" and "wetter" conditions.

Line 526: I believe that "begin" should here be a singular, "begins".
Thanks, this will be changed in the revised manuscript.

Line 638: Shouldn't "variation" be a plural here?
Thanks, we will correct this.

There are a few inconsistencies between US and British spelling. I noticed the use of "behavior" and "behaviour" in the text. Please check.

Thanks for pointing this out. The reviewer points out inconsistencies between US and British spelling (e.g., the use of "colour" but then at the same time "parametrize"). Some of these inconsistencies can be explained by the chosen variety of English, Canadian English, which is the first author's first language and is accepted by the EGU. However, we will change "behavior" to "behaviour" in text, and willl check for other inconsistencies (e.g., "…ise" → "…ize") to remain consistent with Canadian English.

Many DOIs link have duplicated "https://doi.org/https://doi.org/" in the reference list. Pleas check.

Many thanks for pointing this out. We will correct the DOI links in the references.

**Additional material:**

Fig. A1: Role of radiative forcing and AMOC on the evolution of global mean surface temperature. Trajectories have been plotted for the entire 100,000 years.

[Figure]

Fig. A2: Preliminary version of the revised Fig. 13 in the manuscript. This figure now includes subplots for (b) $A_i$ vs E and (c) $\tau_i$ vs E. The results for the PULSE experiment will also be shown here once available.

[Figure]

Fig. A3: Change in global mean surface temperature in the 0-3000 PgC emission scenarios. Colours here correspond to the cumulative emission scenarios shown in Fig. 2 of the manuscript. The response in temperature is shown here for two experiments: solid line for the intCH4 experiment and dashed lines for the REF experiment.

[Figure]

Fig. A4: Change in soil carbon inventory in the 0-3000 PgC emission scenarios. Colours here correspond to the cumulative emission scenarios shown in Fig. 2 of the manuscript. The response in soil carbon is shown here for two experiments: solid line for the intCH4 experiment and dashed lines for the REF experiment.

[Figure]

Table A1: Lithological classes in GLiM (Hartmann & Moosdorf 2012) and their parameters. These lithological classes are summed using the Arrhenius equation. The lithological classes of loess (lo) an carbonate sedimentary rock (sc) are not shown here. The evaporites class (ev) is used only to compute phosphorus fluxes, and is therefore not considered here.

| | Lithological code | Lithological class | Molality / weathering rate, b $((1/12)\times molC/kg$ water) | Activation energy of silicates, $E_{a,sil}$ (kJ/mol) | Fraction to weather as carbonate rocks, $\alpha$ |
|---|---|---|---|---|---|
| 1 | mt | Metamorphics | 0.007626 | 60 | 0.75 |
| 2 | pa | Acid plutonic rocks | 0.005095 | 60 | 0.42 |
| 3 | pb | Basic plutonic rocks | 0.007015 | 50 | 0 |
| 4 | pi | Intermediate plutonic rocks | 0.007015 | 60 | 0.42 |
| 5 | py | Pyroclastics | 0.0061 | 46 | 0 |
| 6 | sm | Mixed sedimentary rocks | 0.012481 | 60 | 0.76 |
| 7 | ss | Siliciclastic sedimentary rocks | 0.005341 | 60 | 0.36 |
| 8 | su | Unconsolidated sediments | 0.003364 | 60 | 0 |
| 9 | va | Acid volcanic rocks | 0.002455 | 60 | 0 |
| 10 | vb | Basic volcanic rocks | 0.007015 | 50 | 0 |
| 11 | vi | Intermediate volcanic rocks | 0.007015 | 50 | 0 |

**References:**

Cox, P. M., Betts, R. A., Jones, C. D., Spall, S. A. & Totterdell, I. J. (2000). Acceleration of global warming due to carbon-cycle feedbacks in a coupled climate model. *Nature*, 408(6809), 184–187. https://doi.org/10.1038/35041539

Hartmann, J. & Moosdorf, N. (2012). The new global lithological map database GLiM: A representation of rock properties at the Earth Surface. *Geochemistry, Geophysics, Geosystems*, 13(12). https://doi.org/10.1029/2012gc004370

Hartmann, J., Moosdorf, N., Lauerwald, R., Hinderer, M. & West, A. J.: Global chemical weathering and associated P-release — The role of lithology, temperature and soil properties, *Chemical Geology*, 363, 145–163, https://doi.org/10.1016/j.chemgeo.2013.10.025, 2014.

Willeit, M., Ilyina, T., Liu, B., Heinze, C., Perrette, M., Heinemann, M., Dalmonech, D., Brovkin, V., Munhoven, G., Börker, J., Hartmann, J., Romero-Mujalli, G. & Ganopolski, A. (2023). The Earth System Model CLIMBER-X v1.0 – Part 2: The global carbon cycle. *Geoscientific Model Development*, 16(12), 3501–3534. https://doi.org/10.5194/gmd-16-3501-2023

---

## Author Comment (AC4)

**Response to CC1**

Dear Paul Pukite,
Many thanks for your community comment; we hope we can clarify any misunderstandings. Please find our point-to-point responses to the individual comments below (community comment in black, our response in blue).

Why is it that the predominating mechanism behind the fat-tails of CO2 persistence in the atmosphere is never mentioned in the article? CO2 enters the ocean and only gradually diffuses downward, modeled as an infinite number of slabs according to conventional 1D physics.

The mechanisms behind the long ("fat") tail of the anthropogenic $CO_2$ anomaly in the atmosphere is well-understood and already discussed on Page 4 (Lines 63-64) where we cited relevant papers. Indeed, most of anthropogenic $CO_2$ (up to 80%) will dissolve and diffuse into ocean, where wind-driven mixing and large-scale ocean circulation (e.g., thermohaline circulation) will transport it into the deeper waters at the millennial time scale. The rest of anthropogenic $CO_2$ is removed from the atmosphere by interaction with marine sediments ($10^4$ yr) and silicate weathering ($10^5$ yr). Thus, the contribution of carbonate chemistry and weathering to the continued absorbtion by the oceans cannot be ignored (as done with a diffusion-only model).

This leads to an inverse power law tail, matching to the BERN heuristic of a set of damped exponentials and a fudge factor constant level representing the rest of the tail. The paper estimates that "75% of anthropogenic CO2 is removed within 197–1,820 years after emissions end". It would be useful to explain that statistical moments such as the mean adjustment time can only be expressed as such a range because the value will actually diverge with a fat tail.

We wish to clarify that the anticipated "long tail" of $CO_2$ uptake is due to the relatively slow process of silicate weathering, not ocean diffusion. This is illustrated in Figure 1 of Archer & Brovkin (2008). We are aware of the "BERN heuristic of a set of damped exponentials", but this is used exclusively to analyze our results (not used as a model for the oceanic uptake of anthropogenic $CO_2$). This is applied in Section 3.3, where such a set of exponential decay functions are used to estimate the removal timescales of ocean invasion, carbonate chemistry, and silicate weathering.

**References:**

Archer, D. & Brovkin, V. (2008). The millennial atmospheric lifetime of Anthropogenic CO2. *Climatic Change*, 90(3), 283–297. https://doi.org/10.1007/s10584-008-9413-1

---

## Author Response (AR1)

**General response:**

We sincerely appreciate the constructive and positive feedback provided by all the reviewers. Their recommendations have greatly enhanced the manuscript's structure and content. Please find our point-to-point responses to the individual comments given by the Reviewers below (reviewer comment in black, our response in blue). As an overview, and in response to all reviewers, we have addressed the following aspects in the revised manuscript. The ***line numbers*** and ***page numbers*** in this document refer to the revised mauscript file without tracked changes, unless otherwise stated.

1. Shortening the paper and re-organizing our results: We have made considerable effort to shorten our paper (and focus on the novelties of this study), by identifying content in Sections 3.1.2, 3.3, and 3.4 that does not signficantly contribute to our main points. For this, we have removed the following portions of the original manuscript: Lines 29-32, Lines 213-214, Lines 245-249, Lines 277-279, Lines 292-294, Lines 320-321, Lines 430-440, Lines 487-498, and Lines 503-506 (total $\simeq$ 38 Lines). We moved Table 2 to the appendix. We removed Fig. E2 as it was not referenced in text, and Fig. 14a from the original manuscript as it more or less shows the same as 14b-c. We made Fig. 2 a single column, similar to Fig. 3, 5, 7, 8 and 9. We have reorganized and rewritten certain statements as per the reviewers' specific comments (see individual responses).

2. Enhancing the discussion: We have rewritten large portions to account for a new PULSE experiment in Sections 3.2-3.3. We have attributed weathering flux changes to either temperature or runoff changes in Section 3.1.2 (Fig. E6). We have since revised our statement that the temperature plateau in the 5000 PgC scenario is not due to soil carbon, but rather, due to AMOC (***Lines 193-195***). We have expanded on the potential caveats in our study related to erosion limitation (***Lines 632-634, 676-680***) and tipping points (***Lines 689-691***) in Section 5. We have added information on the temperature evolution in the intCH4 experiment (***Lines 596-598***), and have clarified that increased temperatures in ECS4 are due to a positive climate—carbon cycle feedback in Section 4.4 (***Lines 558-559***).

3. Adding additional figures and tables: We added subplots to Fig. 13 for $A_i$ vs E and $\tau_i$ vs E (Fig. 13b,c), and a figure for temperature evolution in the intCH4 experiment in the appendix (Fig. E7). We have provided a table of lithology-specific weathering model parameters (e.g., activation energy of silicates for the different lithologies) in the appendix (Table E1). We have added another figure showing the fraction of a given lithology in a grid cell (Fig. E1). We have moved the fitting parameters for the multi-exponential decay fit, now including the PULSE experiment, in the appendix (Table E2). We have added another figure for attributing changes in the weathering fluxes to either temperature or runoff to the appendix (Fig. E6).

4. Clarifying certain aspects: We clarified the treatment of organic carbon in the revised manuscript (***Lines 108-113***). We have expanded on the weathering scheme by including a table of lithological values and a figure showing the fraction of a given lithology in a grid cell (Fig. E1, Table E1) for Equations 2 and 3. We have incorporated the equations for carbonate weathering specific to loess and carbonate sedimentary rocks. We have clarified in certain figure captions whether time is

counted from the beginning of the pulse, or from maximum $CO_2$ concentration (e.g., Fig. 3, Fig. 16, Fig. E3, Fig. E4, Fig. E5). We have added text to make it explicitly clear where we do not consider ramp up time of emissions (see individual responses).

**Erroneous changes:**
- An error was noticed in the fitting procedure in Fig. 10, as it did not reflect the response of weathering given by some equations (e.g., Uchikawa & Zeebe 2008) to increases in temperature. It has since been corrected (***Page 18***), although it does not change our findings.
- Before, we asserted that 5% of land area was responsible for about a third of carbonate and silicate weathering fluxes. This was erroneous, as we did not take into account latitudinal differences in grid-cell area. It has since been corrected to "About 10% of land area across the different emission scenarios is responsible for ~13-14% of the total carbonate weathering and ~33-37% of the total silicate weathering" (***Page 19, Lines 320-322***) and "As 10% of the land area in our simulations is responsible for more than a third of global silicate weathering fluxes" (***Page 35, Lines 674-676***).

**Response to RC1:**

In their paper 'Assessing the lifetime of anthropogenic CO2 and its sensitivity to different carbon cycle processes' Kaufhold et al. apply the CLIMBER-X Earth system model of intermediate complexity to investigate the atmospheric lifetime and the removal processes of cumulative CO2 emission in a set of 100 kyr experiments.

They include a variety of sensitivity experiments to further investigate the role of the landbiosphere and weathering feedbacks in removing atmospheric CO2 perturbations. Overall, the paper is well written and provides interesting results and nicely addresses the question of the landbiosphere and weathering feedbacks for the removal of atmospheric CO2 perturbations and provides and a wealth of figures. The spatially explicit weathering scheme of CLIMBER-X is an important addition to the investigation! In summary, the study is well suited for publication in Biogeosciences.

I have three more general aspects and a few minor comments the authors may address during revision.

We would like to thank the reviewer for their positive comments on our paper.

**General:**

1) Presentation of the C-perturbation

In the paper, the authors alternate between presenting the atmospheric C-perturbation (or the perturbation of other reservoirs) in absolute values (i.e. ppm CO2 or PgC) and as fraction of the maximum CO2 perturbation. For comparison with other studies and to address non-linearities it would, in my eyes, be much easier to show results as fractions of the CO2 perturbation rather than as absolute values (for example also in Fig. 1). The fact that the atmospheric CO2 perturbation (in ppm) is larger for larger cumulative emissions is not surprising and it could be interesting to investigate the non-linearities in more detail instead. An alternative could also be to normalize the results by the response to a certain pulse size to highlight the non-linearities (e.g. Fig. 2).

We acknowledge that alternating between absolute values (as in Section 3.1) and fractional values (as in Sections 3.2-3.4) could be confusing, and that expressing some responses as fractions of cumulative emissions may be more interesting for exploring nonlinearities. In principle, we could have provided a version of Fig. 1 that shows the change in atmospheric $CO_2$ as a fraction of cumulative emisions. However, the method used to determine atmospheric $CO_2$ concentration from other publications (i.e., visual inspection) is not accurate enough, meaning that the figure is semi-qualitative and mostly serves an illustrative purpose. We have provided a statement on this in the figure caption of the revised manuscript:

***Page 3****: Moved and revised in Fig. 1 caption "The data from previous studies shown here was acquired through visual inspection of graphs, meaning that the figure is semi-qualitative and mostly serves an illustrative purpose (small errors may be present)."*

Furthermore, we are cautious about modifying the current presentation of these results in Fig. 2 (either by normalizing it by cumulative emissions, or by a certain pulse size) as it could make it difficult for future comparisons with our work, especially given the ramp up period. We have aimed to remain consistent with how values have been presented in prior studies. For instance, atmospheric $CO_2$ concentrations have consistently been reported in absolute values (e.g., Archer 1998, Lenton & Britton 2006, Ridgwell & Hargreaves 2007, Archer & Brovkin 2008, Archer et al. 2009, Lord et al. 2015, etc.), and emissions removed or remaining given in such normalized values (e.g., Archer 1997, Eby et al. 2009, Lord et al.

2015, etc.). There are a few exceptions to this, but these are typically made for specific reasons. For instance, Joos et al. (2013) report atmospheric $CO_2$ only as a fraction of emitted emissions, but their study's objective was to compare the response of different models.

2) how emissions are prescribed
The way emissions are prescribed in this study (as Gaussian function) complicates the comparison with studies featuring a pulse-like release of carbon (as often done) to a certain degree. This leads in this study, for example, in the case of small total cumulative emissions, to a large fraction of the atmospheric CO2 perturbation already having been removed before reaching the maximum atm. CO2 perturbation and also to less timescales required when fitting the response as a sum of exponentials (section 3.3) as compared to studies with an instantaneous pulse-like emission. While this is acknowledged in the text, I think it should be made more clear, especially for the discussion of the timescales in sections 3.2-3.4.
This is a very good point. We agree that this complicates comparisons to other studies, especially as removing the ramp up of emissions may affect the calculated timescales in Section 3.3. As such, we have made the limitations of our analysis (for the REF experiment) more clear (see below). As per Reviewer #2's recommendation, we have also made it more clear in the Sections whether or not a figure or analysis includes the ramp up period of emissions.

*Page 22, Lines 393-396: Added "However, short-term processes (sub-centennial timescales) cannot be fit with our analysis after removing the ramp up period of emissions, meaning n=3 is already sufficient for examining the long-term uptake of anthropogenic $CO_2$. It should be noted that, due to this, the values for $A_i$ in the REF and noLAND experiments do not sum to 1 as a fraction of emissions was already removed via short-term processes (Fig 12b)."*
*Page 23: Added in Fig. 13 caption "Scenarios with smaller cumulative emissions typically result in lower $A_i$ values, as a larger proportion of emissions is taken up by short-term processes (sub-centennial timescales), which are not considered in our multi-exponential fitting procedure (see Lord et al., 2015b)."*

Further, it might be interesting to add one additional emission pathway sensitivity experiment, where all the carbon is emitted in the first timestep, as done in many of the studies discussed in the paper. In light of how fast the CLIMBER-X model is (10'000 years per day), this might be doable.
Originally, we chose to not use a pulse-like $CO_2$ perturbation due to concerns about model stability. However, following the reviewer's suggestion, we initiated such experiments for a new emissions pathway ensemble called 'PULSE.' In the revised manuscript, Sections 3.2 and 3.3 have been extensively rewritten to make the manuscript more concise, and include the results of the PULSE experiment. We believe this addition has enhanced the manuscript by providing greater robustness to the estimated fraction of emissions remaining and timescales.

*Addendum: The PULSE experiment is now incorporated in Table 1, Fig. 12, Fig. 13, Fig. 16, and Table E2, as well as in the disscusion in Sections 3.2, 3.3 and 5.*
*Page 21-22, Lines 367-371: Revised "This analysis was exclusively done on the REF experiment, the experiment with a pulse-like perturbation of $CO_2$ (PULSE, Table 1), and the experiment with land carbon disabled (noLAND, Table 1). The latter two experiments were included to (1) provide confidence in our estimated timescales by excluding the ramp up period of emissions as a factor (aligning with the procedure used in similar studies, e.g., Archer et al., 1997), and (2) facilitate a direct comparison with the findings of Lord et al. (2015b)."*

3) Length of the paper
While the paper does a very nice job in thoroughly describing processes and visualizing a lot in figures, I found it quite lengthy to read. Maybe during revisions this could be kept in mind. For example, in my opinion, sections 3.2-3.4 could be merged and shortened with a focus on the novelties of this study (timescale of the silicate weathering feedback).

In our efforts to make the manuscript more concise, we have removed large portions of Sections 3.1.2, 3.3, and 3.4 that did not significantly contribute to the discussion (see our "General response #1" for specific lines that were removed). However, we did not fully merge Sections 3.2–3.4, as we believe maintaining their separation helps preserve structure in the analysis. Considering the additional requests for more information about our weathering scheme, a PULSE experiment, the attribution of weathering fluxes to either temperature/runoff, an expanded discussion on erosion limitations, tipping elements, and temperature evolution/feedbacks in the intCH4 experiments (see our "General response #2-4"), it was not possible to significantly reduce the overall length despite these changes.

**Minor comments:**

- p. 5, l. 105: why is the conservation of phosphate and silicate enforced and how is it done?

As mentioned in the CLIMBER-X description paper (Willeit et al. 2022, 2023), the weathering module includes equations for silicate weathering fluxes. Riverine fluxes of silicate, phosphorus and organic carbon have been disabled in the model set-up here, as they would introduce additional challenges related to nutrient conservation in the ocean. Instead, the budgets for silicate and phosphorus (as well as organic carbon, not mentioned in the manuscript) are balanced by assuming that sediment burial fluxes are returned in remineralized form to the surface ocean. Spatially, these surface fluxes are distributed proportionally to annual runoff. This simplified approach ensures the conservation of phosphorus and silica inventories within the ocean–sediment system throughout the simulation.

We have provided more information clarifying this (as well as that of organic carbon) in the revised manuscript, as this issue was also raised by Reviewer #3:

***Page 5, Lines 108-113***: *Revised "In the open carbon cycle setup, a simplification is made to enforce that the budgets for silicate and phosphorus within the ocean—sediment system are balanced. This was done by disabling riverine fluxes of phosphorus and silicate, and assuming that organic carbon (which includes P) and opal (which includes Si) which are buried in the sediments (and, therefore, removed from the system) are instead returned to the surface ocean in remineralized form. The conservation of such inventories in the ocean—sediment system removes challenges related to nutrient conservation that would otherwise complicate the analysis and interpretation of model results."*

- p. 6, l. 127-130: please provide values for the parameters

We agree that it would be logical to provide values for $\alpha$, b, and $E_a$, and have done so in the revised manuscript (***Page 40***, Table E1). We have now specified in the revised manuscript that $\ell$ represents the different lithologies to sum over, and explicitly defined $b(\ell)$, $E_{a,sil}(\ell)$, and $\alpha(\ell)$ as functions of lithology (***Page 6, Lines 132-137***). Furthermore, we have also included information about the grid-cell lithology by including a figure showing the spatial distribution of the different rock lithologies in CLIMBER-X (***Page 39***, Fig. E1).

- p. 7, l. 149ff: looking forward to the interactive ice-sheet simulations!

Thank you for the positive comment, we are also excited to share these simulations in a follow up study.

- p. 7, l. 155: 'pulse' might be a misleading term, maybe replace with 'idealized CO2 emission histories'?
This is a good point, especially with the inclusion of the 'PULSE' experiment ensemble. We changed this to 'scenarios' in the revised manuscript:

***Page 7, Line 161***: *Revised "idealized $CO_2$ emission [scenarios]"*

- section 3.1.3: very nice!
Thank you for the positive comment on this section.

- Fig. 13: maybe clarify in the figure caption as well, why the smaller cumulative emissions lower fractions removed (-> more already taken up by other reservoirs before reaching the max. CO2 perturbation)
We have added a sentence clarifying this, as suggested:

***Page 23***: *Added in Fig. 13 caption "Scenarios with smaller cumulative emissions typically result in lower $A_i$ values, as a larger proportion of emissions is taken up by short-term processes (sub-centennial timescales), which are not considered in our multi-exponential fitting procedure (see Lord et al., 2015b)."*

- section 4: I really liked this section!
Thank you for the positive comment on this section.

- Fig. 16: check caption text, not fully clear
We assume that the reviewer is referring to the description of the cumulative stacked barplot in Figure 16. Our intention of this description was to communicate that the barplot is cumulative. By default, plotting a stacked barplot in Python would be non-cumulative, and the length of each bar would represent the magnitude of carbon uptake. For example, as it is shown now, Figure 16a REF shows that land in the 500 PgC scenario takes up ~130 PgC carbon whereas land in the 1000 PgC scenario takes up ~240 PgC. Should the stacked bar plot be non-cumulative, the 500 PgC scenario would still be the same, but the 1000 PgC bar would instead reach ~370 PgC. However, we understand how it is currently written may cause confusion, so have clarified this by rewriting the second sentence in the figure caption:

***Page 30***: *Revised in Fig. 16 caption "The stacked bar plot is cumulative, meaning that the height of the bar (rather than the bar length) in each emission scenario reflects the magnitude of carbon uptake or loss."*

- p. 32, l. 590: 'effect' -> 'affect'?
Thanks, we changed this in the revised manuscript.

***Page 31, Line 578***: *Revised "…concentration could therefore [affect] the capacity of different Earth…"*

- p. 35, l. 646ff: move this part to the other statements about silicate weathering before
Thanks, we have moved this to ***Page 34, Lines 614-629***.

**Response to RC2:**

This study assesses the lifetime of anthropogenic CO2 and its dependence on total emissions and sensitivity to different carbon cycle processes. The authors use the fast EMIC CLIMBER-X to simulate an ensemble of 100,000-year simulations. They thoroughly analyze which carbon reservoir takes how much carbon and when. Special attention is paid to the timescales of silicate weathering and this study provides a shorter estimate of this timescale compared to previous studies. Next sensitivity experiments are discussed to assess how sensitive the results are to several important processes.
The study is interesting and textually well written. I believe most of the reasoning, as well as the used methodology, are sound. I have a couple of major comments, minor comments and specific comments for the authors to address.
We would like to thank the reviewer for their positive comments on our paper.

**Major comments:**

My first main comment is that the paper is quite long and has a lot of figures. I think the paper would benefit from being a bit shorter. I will leave it to the authors to decide exactly what to change, but I have provided some suggestions below:
In our efforts to make the manuscript more concise, we have removed large portions of Sections 3.1.2, 3.3, and 3.4 that did not significantly contribute to the discussion (see our "General response #1" for specific lines that were removed). Considering the additional requests for more information about our weathering scheme, a PULSE experiment, the attribution of weathering fluxes to either temperature/runoff, an expanded discussion on erosion limitations, tipping elements, and temperature evolution/feedbacks in the intCH4 experiments  (see our "General response #2-4"), it was not possible to significantly reduce the overall length despite these changes.

Section 5 mainly focuses on the weathering processes, and I believe this is also the main novelty in this study. Section 3 is much more elaborate. Is it necessary to, for example, put 3.1.1 and 3.1.2 in the main text or could they also go into the supplementary?
Weathering processes are only one of the novelties of our work. The main novelty of our work is that we have carried out long-term simulations with a comprehensive Earth system model, and have analyzed the response of the carbon cycle on different time scales –not just for weathering processes. Therefore, we disagree that Sections 3.1.1 and 3.1.2 do not contribute significantly to the main text. While it is true that the weathering processes and their behaviour under different cumulative emission scenarios are a key novelty of our manuscript, the spatially explicit response of the land biosphere is also critical. In previous studies, the land has often been completely omitted (e.g., Archer et al. 1997; Lord et al. 2015; Köhler 2020), simulated with very low complexity and left unreported (e.g., Lenton & Britton 2006), or modelled with similar complexity but not over such long timescales (e.g., Brault et al. 2017). As highlighted by Reviewer #3, the non-monotonous response of soil carbon over long timescales observed in our study may even represent a novel finding. Furthermore, as ocean processes and carbonate chemistry are responsible for the majority of anthropogenic $CO_2$ removal, we also believe it would not be appropriate to completely omit this section. However, as we agree that this section in particular could be streamlined, we have made considerable efforts to shorten it where possible (see our "General response #1" for specific lines).

- Is it possible to combine some figures? E.g. Figures 7 and 8, and/or Figures 12 and 13.

We think it would be difficult to combine Fig. 7 and 8 without potentially omitting the response of a few variables. We also think it would be difficult to merge Fig. 12 and 13 considering the proposed changes to Fig. 13 suggested by Reviewer #3. However, we have removed Fig. 14a in the original manuscript, as it more or less shows the same as Fig. 14b, and presented Fig. 2 in a single column, similar to Fig. 3, 5, 7, 8 and 9.

- In the introduction the authors mention the next glacial cycle. Is it necessary to include this in the introduction?

While it is not necessary to mention the next glacial cycle, we believe it is important in discussing the current state of modelling the long-term future climate evolution. The inclusion of two examples may be excessive, however, so we have omitted Lines 29-32 of the original manuscript.

***Page 2***: Removed *"A consensus also remains to be seen regarding the timing and duration of the next glacial cycle; model realizations which satisfy paleoclimatic constraints in Talento and Ganopolski (2021) suggest that, under non-anthropogenic conditions, the next full glacial was expected to occur in approximately 90 to around 150 kyr."*

- Are all figures necessary to go into the main text or can they go into the supplementary? E.g. Figure 1?

We have made full use of the appendix for figures that are not immediately essential at first glance. Indeed, our manuscript has many figures, but we believe our manuscript is stronger when including the responses of the different components (both spatially explicit and timeseries), the associated analyses, and the sensitivity experiments. We have left Fig. 1 in the introduction as, like the previous comment, we believe it is important for establishing the current state of the field for long-term future climate modelling.

One thing that remains a bit unclear to me is how the time period where the emissions occur is treated in the analysis. I think it is necessary to mention this in a clear and explicit way. Is the period with emissions included in the 100,000 years? For what analyses is it included, and for what not?

Thank you for the suggestion. We agree that it would be helpful to clarify how the period with emissions is treated in the analysis. For figures showing the time evolution, t=0 corresponds to the beginning of the pulse. However, for all processing and further analysis (i.e., results not focused on the model response to emissions), the ramp-up period of emissions is excluded. We attempted to convey this by noting at the start of certain sections that the analyses were conducted for the time after the peak $CO_2$ concentration (i.e., $CO_2(t_0) = CO_2^{max}$), and to explicitly make a distinction between time after peak $CO_2$ concentration (yr) vs. time (yr) in the figures. In the revised manuscript, we identified areas where there could be ambiguities (e.g., caption of Fig. 3) and have made sure it is clear to the reader that the emission period is part of the simulated 100,000 years. Furthermore, as per Reviewer #1's suggestion, we have also incorporated results from the PULSE ensemble (where all the carbon is emitted in the first timestep), which avoids complications arising from the emission phase duration.

*Page 8, Lines 167-168: Added "It should be noted here that the 100 kyr simulation duration includes the ramp up and ramp down of emissions."*
*Page 20, Lines 338-339: Added "The ramp up of emissions is not included in this analysis."*
*Page 20, Lines 345-346: Added "...despite the potential complication from the ramp up and ramp down period of emissions in the REF experiment..."*
*Page 22, Lines 381-383: Added "Therefore, we decided to fit all data starting from peak concentration in atmospheric $CO_2$ (as done in Section 3.2), thereby omitting the ramp up period of emissions."*
*Page 26, Lines 482-483: Added "Like Sections 3.2 and 3.3, the ramp up period of emissions was excluded, meaning that..."*

Following on this comment, I also suggest including in the figures when there are CO2 emissions and/or when there is an atmospheric CO2 maximum. I think this will make the interpretation of the figures next to the text easier.

Thanks for the suggestion; we have made sure that this is explained in the figure captions where there could have been ambiguities:

*Page 10: Added to Fig. 3 caption "It should be noted that the ramp up of emissions is excluded here."*
*Page 30: Added to Fig. 17 caption "It should be noted that the time slices here are measured from the start of the simulations, which includes the emissions ramp up period."*
*Page 42: Added to Fig. E3 caption "The time slice here is measured from the start of the simulations, meaning it includes the emissions ramp up period."*
*Page 43: Added to Fig. E4 caption "The time slice here is measured from the start of the simulations, meaning it includes the emissions ramp up period."*
*Page 44: Added to Fig. E5 caption "The time slice here is measured from the start of the simulations, meaning it includes the emissions ramp up period."*

**Minor comments:**

Abstract: Is it possible to add how the range relates to the cumulative emission range?
To avoid ambiguities, we have added a few sentences linking our provided ranges to the range of cumulative emissions in the abstract of the revised manuscript.

*Page 1, Line 9: Added "Our findings indicate that, depending on the magnitude of the emission, 75% of anthropogenic $CO_2$ is removed within 197-1,820 years after peak $CO_2$ concentration [(with larger cumulative emissions taking longer to remove)]."*
*Page 1, Line 12: Added "Higher emission scenarios fall on the lower end of this range as increased soil respiration leads to greater carbon loss."*
*Page 1, Lines 17-18: Added "Furthermore, this timescale is shown to have a non-monotonic relationship with cumulative emissions."*

Line 51: this statement misses a reference.
We have added a reference to Archer et al. (2009), which corroborates our statement here (*Page 2, Line 52*).

Line 105, 106: and through emissions.
That's a good point, thanks. We have added that as suggested:

*__Page 5, Line 108__: Added "Carbon is not conserved in this setup; it is removed from the system through sediment burial and introduced to the system via weathering, volcanic outgassing, [and through anthropogenic emissions]."*

Line 118, 119: I suggest mentioning here explicitly what processes the weathering scheme depends on. Since the weathering plays a major role in the manuscript, I think it is important that the weathering scheme is as explicit and clear as possible which also makes it easier to compare it to previous studies using different weathering schemes.

This has already been explicitly mentioned on __Page 6, Lines 121-123__ of the revised manuscript: "PALADYN includes a rock weathering scheme influenced by runoff and temperature (Hartmann, 2009a; Börker et al., 2020), accounting for 16 different lithologies as described in Hartmann & Moosdorf (2012)." However, as per Reviewer #3's recommendation, we have also explicitly included the equations for loess and carbonate sedimentary rock weathering in Equation 2 (__Page 6, Line 126__), and as per Reviewer #1's recommendation, we added a table focused on weathering parameters for the different lithologies (__Page 40__, Table E1). We have now specified in the revised manuscript that $\ell$ represents the different lithologies to sum over, and explicitly defined $b(\ell)$, $E_{a,sil}(\ell)$, and $\alpha(\ell)$, as functions of lithology (__Page 6, Lines 132-136__). Furthermore, we have also included information about the grid-cell lithology by including a figure showing the spatial distribution of the different rock lithologies in CLIMBER-X (__Page 39__, Fig. E1). The "16 lithologies", which was erroneous, was fixed in the revised manuscript to 13.

Line 138: where do the numbers come from? Is there a source?

The reviewer here is asking where the values given for volcanic outgassing (0.0738 PgC yr$^{-1}$ or 6.15 TmolC yr$^{-1}$) come from. We set volcanic outgassing to half the simulated global silicate weathering rate at the pre-industrial time (this is already mentioned in the manuscript on __Page 6, Lines 142-144__). The numbers come from the 100,000 year equilibrium spin-up of the carbon cycle model as described in Willeit et al. (2023), which is mentioned a few lines later (__Page 7, Lines 149-150__). This value is not based on observations (although it is consistent with the range of observational estimates), but is determined from the condition ensuring that the carbon cycle model is in equilibrium at the pre-industrial climate state, which requires that volcanic outgassing is half the rate of simulated silicate weathering under pre-industrial climate conditions (Munhoven & François 1994; Willeit et al. 2023).

Line 185: I do not see a large response in atmospheric CO2 concentrations, whereas the response in temperature and land carbon are much clearer. I interpret this as that it is not the CO2 concentrations that cause the warming. What does cause this warming?

Many thanks for bringing this to our attention. Reviewer #3 also highlighted this issue. Upon re-evaluating the data, we recognize that this statement was erroneous. Our analysis suggests that the temperature stabilization observed in the 5000 PgC scenario during the first millennium is not driven by the release of soil carbon into the atmosphere, but rather by ocean dynamics and the AMOC. The extended decline in AMOC results in a cooling in the Northern Hemisphere which prevents global mean temperature (GMT) from rising after year ~150. After some time, this cooling is offset by Southern Hemisphere warming via the bipolar seesaw, and explains why GMT stabilizes during the better part of the first millennium. This behaviour continues until the abrupt AMOC recovery, which triggers a rapid increase in GMT (and there is a small bump in global temperature at this time). The role of AMOC on temperature, rather than $CO_2$ (radiative forcing, $\log(CO_2)$) is demonstrated in Fig. A1 of the "Additional material" section at the end of this document. We have revised this statement accordingly in the updated manuscript:

*Page 8, Lines 193-195: Revised "This is largely due to the extended decline in the Atlantic Meridional Overturning Circulation (AMOC; Fig. 7e), which results in a cooling in the Northern Hemisphere that prevents global mean temperature from further rising."*

2d, e are not referenced.
While Fig. 2d is already referenced in Section 3.1.2, we acknowledge that Fig. 2e was indeed not referenced in the text. We since have referenced Fig. 2e in the revised manuscript in the following way:

*Page 16, Lines 264-265: Added "…(contributing more to the total loss of sediment carbon inventory than sediment organic carbon, Fig. 2e)."*

[Fig.] 4 is referenced before Fig. 3.
Thanks, we have switched these two.

Section 3.1: The sediments are not treated as explicitly as the other reservoirs. Is this for a reason?
Section 3.1 largely introduces the general response of our experiments and the relative partition of anthropogenic carbon into the different reservoirs. The reviewer here asks why we did not treat sediments as their own reservoir.
This was intentionally done here as we performed our experiments with an open carbon cycle. As mentioned in the Fig. 3 caption (and in Appendix B), we focus on changes in cumulative carbon flux from the atmosphere relative to the pre-industrial, instead of changes in carbon inventory (as typically done), as a way to get around this issue. Given that there is no direct air–sediment flux, any anthropogenic $CO_2$ absorbed by sediments must first go through the ocean. This is why we conglomerated the two as one reservoir.

Line 214: Is it possible to give a one or two sentence summary of Kaufhold et al. (2024) here?
To enhance the conciseness of the revised manuscript, we have removed this sentence in its entirety.

*Page 12: Removed "Uncertainties in the land carbon response over the next millennium has been explored in more detail in Kaufhold et al. (2024)."*

Line 219: Fig. 4a is referenced, but there is not explicit treatment of soil carbon in Fig. 4a. Is the right figure referenced?
We realize that the reviewer was indeed correct that we cited the wrong figure here. Instead of the previously proposed changes, we have now referenced the correct figure:

*Page 11-12, Lines 225-226: Revised "…preventing the overall decline in soil carbon inventory (Fig. [5b])."*

Figure 7a and b show more or less the same thing. Is it necessary to show them both?
Indeed, the Revelle factor and pH are related to ocean chemistry and $CO_2$ absorption, but they focus on different aspects of the carbon cycle and acidification process. The Revelle factor indicates how easily the ocean can absorb $CO_2$, relative to the relative increase in atmospheric $CO_2$. Surface ocean pH indicates the acidity of sea water, which is affected by the actual amount of $CO_2$ absorbed. Although the Revelle factor is a critically important

metric for communicating buffering capacity, it is unfortunately often not reported in such long-term studies on the uptake of anthropogenic $CO_2$ (both in terms of peak magnitude and duration across the different emission scenarios). We thought that calculating this explicitly might be useful for future studies to reference.

Line 313: Is it possible to determine how much of the changes in weathering rates are because of changes in temperature, and how much due to changes in run off? I think this would make for a nice addition.
We agree that it would be a nice addition. Therefore, we have quantified the relative contribution of temperature and runoff to changes in the weathering rate with a new figure in the revised manuscript:

*Page 18, Line 313-314: Added "Runoff, which indirectly depends on precipitation through soil infiltration and drainage (Willeit and Ganopolski, 2016), drives some of these changes (Fig. 11g,h,i, [Fig. E6])"*
*Page 45: Added Fig. E6.*

Line 315-317: I do not fully understand this sentence.
The reviewer is referring to the following sentence: "For carbonate weathering, large changes are not only limited to the equatorial regions, although the highest and lowest weathering rates in South East Asia and Central Asia can also be explained by increases and decreases in precipitation (Fig. 11b,c)".
Here, we mean that large increases in carbonate weathering with increasing emissions is not only limited to the equatorial regions (as with silicate weathering), but it is more globally distributed. However, the change in carbonate weathering (where it increases and decreases) can also be explained by precipitation. We have clarified this in the revised manuscript:

*Page 18, Lines 317-319: Revsied "Large changes in carbonate weathering are more globally distributed than silicate weathering (Fig. 11e,f). However, like silicate weathering, large changes in carbonate weathering (e.g., increases and decreases in carbonate weathering in South East Asia and Central Asia compared to the pre-industrial ) can also be explained by increases and decreases in precipitation (Fig. 11b,c)."*

Section 4.1: I suggest mentioning the noLAND term earlier.
Thanks for the suggestion. Reviewer #3 also raised a similar point. We have introduced the term in the experimental section of the text, reminded the reader that this refers to the experiment with land carbon disabled, and included a reference to the experiment table.

*Page 8, Line 171: Added "…land carbon cycle response [(noLAND)]…"*
*Page 21, Line 368-369: Added "…and [the experiment with land carbon disabled (noLAND, Table 1)]."*

Figure 16: Would it be beneficial to also construct panels for weathering?
We considered this as well but ultimately decided against it, as weathering is not a true carbon pool like the others (land, ocean, and sediments). Fig. 16 effectively shows the magnitude of carbon uptake of the different pools across the different emission scenarios at different timeslices. The main novelty here is to distinguish how the different sensitivity experiments change the magnitude of carbon uptake. As weathering consumes more $CO_2$ over time, we decided that it would not provide additional information, as it would only increase in relative magnitude between 1 kyr and 100 kyr.

Line 592: Is permafrost treated in the methane model?
Permafrost is treated in the methane model through the representation of anaerobic decomposition in saturated soils (which can occur in permafrost regions). In these areas, carbon in the active layer may decompose under anaerobic conditions, leading to methane emissions. However, methane emissions from permafrost regions are generally small compared to emissions from low-latitude wetlands. We have added the following to our manuscript:

*Page 31-33, Lines 581-582: Added "Permafrost is treated in the methane model through the representation of anaerobic decomposition in saturated soils –a process that can take place in permafrost regions."*

Line 597: How does temperature evolve in intCH4 compared to REF as there is quite a strong increase in CH4 concentrations?
We initially included a more detailed discussion on this topic but later condensed it to maintain the manuscript's conciseness. In general, interactive methane increases peak temperature by up to approximately 0.5 °C, with higher emission scenarios exhibiting greater differences in peak temperature compared to the REF experiment. This was already noted in the original manuscript. However, temperatures in the intCH4 experiments tend to converge to the REF experiments around 1 kyr (similar to $CO_2$ concentrations, Fig. 15). While there are some differences in simulated temperature, the intCH4 and REF experiments largely follow each other over time. At what time temperature is larger/smaller in the intCH4 experiment compared to the REF experiment is primarily influenced by the non-monotonic response of soil carbon to increasing emissions (see Fig. A2-A4 in the "Additional material" section at the end of this document). We have added a figure to the appendix of our revised manuscript showing the temperature evolution in the intCH4 experiment, as well as a statement about how temperature evolves in the intCH4 experiment.

*Page 33, Lines 594-596: Added "Temperature differences between the intCH4 and REF experiments also tend to decrease around this time, and despite small variations in how temperature evolves, the experiments largely follow each other for the rest of the simulation (Fig. E7a)."*
*Page 46: Added Fig. E7.*

Line 674: I think it is good to explicitly mention that ESMs do not agree on centennial timescales.
Thanks, we have revised this as suggested:

*Page 35, Lines 668-669: Added "Although state-of-the-art Earth system models do not agree on the overall response of marine primary production to increasing $CO_2$ levels [on centennial timescales] ..."*

I suggest adding a discussion on how tipping points (might) affect the estimation of the timescales. The AMOC is already mentioned a couple of times in the text, but I think it would be good to reiterate that, and other tipping points, in Section 5.
Potential tipping points are inherently accounted for in our simulations, as CLIMBER-X already incorporates all potential "fast" tipping elements (e.g., AMOC, sea ice, permafrost, boreal forests, and Amazonian forests, etc.). The only exception to this are ice sheets, as the Greenland and Antarctic ice sheets are prescribed by their present day configurations in our

study. However, we know that the East Antarctic ice sheet will not entirely melt in our scenarios (Winkelmann et al. 2015). The West Antarctic Ice Sheet and the Greenland Ice Sheet will (likely) melt, although the net effect of their melt on ocean circulation remains uncertain (Wunderling et al. 2024). As such, it is difficult to predict how this might affect the removal timescale of ocean invasion, and its subsequent influence on other timescales. Furthermore, the Greenland Ice Sheet is projected to melt under the strongest scenario, which could influence weathering rates. This effect has been previously investigated using CLIMBER-2 (Munhoven et al. 2007), but it was shown to be nearly negligible. Without experiments explicitly simulating the crossing of tipping points (and all analyses therein), the influence of these events on the removal timescales of anthropogenic $CO_2$ cannot be assessed and remains highly uncertain. However, we have provided a statement acknowledging that the crossing of tipping points might affect the long-term capacity different components to absorb carbon over time.

*Page 35-36, Lines 689-691: Added "Moreover, significant global warming is expected to trigger the crossing of one (or more) critical thresholds in the Earth system (Lenton et al., 2008; Armstrong McKay et al., 2022; Wunderling et al., 2024), which could affect the long-term capacity of different components to absorb carbon over time."*

**Specific comments:**

Line 61, 62: It is not obvious how non-linear translates to the exponential functions. They are linked here through the word 'therefore', suggesting an obvious connection. I suggest either explaining why non-linear means exponential in this case or rewriting the second sentence a bit.
We rewrote the second sentence to the following:

*Page 4, Lines 63-65: Revised  "The decline in anthropogenic $CO_2$ is usually presented as a superposition of exponential decays (Maier-Reimer and Hasselmann, 1987; Archer et al., 1997; Archer and Brovkin, 2008; Colbourn et al., 2015; Lord et al., 2015b), with each function representing a different process in the carbon cycle that takes up carbon."*

Line 109: I suggest mentioning the units of Catm (i.e. PgC).
Thanks for the suggestion, we have added the unit for $C_{atm}$ as recommended (*Page 5, Line 115*).

Line 194: 'At peak CO2 concentrations, …'
We have changed it as suggested:

*Page 10, Line 202: Revised "At peak $CO_2$ concentration[s] …"*

Line 493: the double fraction does not look so nice in the text.
Although we removed the original sentence that this comment was referring to (deleted Lines 487-498 of the original manuscript, see our "General response #1"), we have since explained the behaviour of τ(t) on Lines 489-490 now with the following:

*Page 26, Lines 489-490: Added "This behaviour in removal timescales is due to multi-millennial processes at this time which temporarily stabilize atmospheric $CO_2$ concentrations (since $\tau(t) \propto \left(\frac{dCO_2(t)}{dt}\right)^{-1}$ )."*

Line 532: remaining where? In the atmosphere?
Thanks for pointing this out, we have changed it to the following:

*__Page 29, Lines 519-520__: Revised "...leads to a higher fraction of emissions remaining [in the atmosphere]".*

Line 566: I suggest rewriting this sentence a bit. I first thought that it meant that if a simulation has lower CO2 concentrations, it has a lower ECS.
We apologize for the miscommunication. We have changed "a lower ECS is associated with lower atmospheric concentrations of $CO_2$ (and vice versa)" to the following in the revised manuscript.

*__Page 31, Line 553__: Revised "...simulations using a lower ECS produce lower atmospheric concentrations of $CO_2$ (and vice versa)..."*

Line 642: 'the presence of land' feels a bit awkward here.
We have changed "the presence of land…" to the following in the revised manuscript:

*__Page 35, Line 651__: Revised "the inclusion of land carbon cycle processes effectively…".*

**Response to RC3:**

Kaufhold et al. manuscript addresses the question of the fate of anthropogenic CO2 and climate in the long-term future (100 thousand years). The authors used an Earth system model of intermediate complexity, which has several new implemented processes compared to previous similar studies. They clearly explain the novelties of their study, and the new findings. The manuscript is very well written, and well organized. It is well-suited for publication in Biogeosciences, with some minor revisions.

We would like to thank the reviewer for their positive comments on our paper.

**Major comments:**

I only have one major comment, which concerns the silicate weathering sensitivity to climate. The authors emphasize their re-estimation of the timescale of carbon removal by silicate weathering, to shorter values than previously thought. They partly attribute this finding to a stronger weathering feedback, which is compared to several estimations (Fig. 10b) and found to fall within the range, though on the upper part (doubling of weathering flux at +4°C, that is +18% per °C of warming).

Among the processes not represented in CLIMBER-X weathering model is the erosion limitation of weathering, or the "soil shielding effect", which is a different point of view of the exact same process. Soil shielding was extensively discussed in Hartmann et al. (2014) (cited in the manuscript), but wasn't yet implemented in Hartmann et al. (2009). Actually, soil shielding is not explicitly represented in any of the model presented in Fig. 10b.

I admit that there is no consensus on how this would affect the sensitivity of weathering to global climate (i.e., the weathering feedback strength), which is the point of interest here. Yet, there are several clues that it would significantly reduce the feedback strength:

- Godderis et al., Geoderma, 2008 (10.1016/j.geoderma.2008.01.020) showed that the sensitivity of tropical weathering to runoff is largely overestimated (~ 5-fold) if considered similar than for temperature climates. Indeed, in the present manuscript, tropical environments dominate the weathering flux, and its response to global warming.
- Maher & Chamberlain, Science, 2014 (10.1126/science.1250770), who also addressed the issue of erosion limitation, suggested a "maximal" weathering sensitivity, in actively eroding mountains, of +5% per °C of warming (which is lowest estimate presented on current Fig. 10b), and an average sensitivity of +1.2% per °C of warming.
- Another weathering model taking into account erosion limitation, and that is spatially explicit, Maffre et al., Clim. Past, 2023 (10.5194/cp-19-1461-2023), suggests a global weathering sensitivity of ~ +9% per °C of warming, though it is unclear if the best fitting functional form should be exponential or linear.

Given the absence of consensus on a value for weathering sensitivity, I do not consider that the present results should be revisited. Simply, I vividly recommend the authors to add more nuances on their statement about weathering timescale (which is one of their main conclusions), and to provide more discussion about weathering sensitivity, the large uncertainty that exists in the literature concerning its value, and how it should affect the weathering timescale.

We are aware of the soil shielding effect, and it was commented on in the CLIMBER-X carbon cycle description paper (Willeit et al., 2023): "The effect of soil shielding on the

weathering rate suggested by Hartmann et al. (2014) has not been considered since information on soil shielding is not readily available for periods beyond the recent past."
As the reviewer correctly identifies, the effect of soil shielding has not been considered by our model (and others) largely because there is no consensus on how it would effect the weathering feedback. However, we do not dismiss the possibility it could significantly change –and potentially weaken— the strength of the weathering feedback. In saying this, we have added a few sentences dedicated to this potential caveat. We also appreciate the compilation of references and they have been added to our manuscript.

*Page 34, Lines 632-634: Added "With this in mind, our estimates here do not address (potentially large) uncertainties related to the sensitivity of silicate weathering processes such as erosion limitation or net primary production on land."*
*Page 35, Lines 676-680: Added "Like many other models, CLIMBER-X also does not account for the potentially significant effect of soil shielding, given the limited quantity of data and ongoing uncertainties regarding its impact. However, there is some evidence to suggest that erosion limitation could weaken the strength of the weathering feedback (Goddéris et al., 2008; Maher and Chamberlain, 2014; Maffre et al., 2023), which may increase the estimated removal timescale of silicate weathering."*

**Specific comments:**

Section 2.2 (lines 100–110): there is a missing information here about the organic carbon cycle. As far as I understand, the sediment component is run as an open system (with sediment loss through burial), and this sediment contains organic carbon generated by marine primary productivity (lines 255–256). Therefore, and given Eq. (1), setting Fvolc to half of the global silicate weathering flux (as indicated lines 136–138) would not result in a steady-state carbon cycle, because of this additional C sink (organic carbon burial), that would result in a net ocean-to-atmosphere flux lower than the remaining term "Fvolc – Fweath". Unless the organic carbon cycle is forced to work as a closed system (like silicate and phosphate, lines 106–107), and all buried organic carbon is put back into the atmosphere?
Many thanks for pointing out this critical issue! We have an open carbon cycle in CLIMBER-X but, indeed, a closed nutrient cycle. We recycle organic carbon in marine sediments along with nutrients, and sediment burial fluxes are returned in remineralized form to the surface ocean while compensating for the subduction of inorganic carbon by volcanic outgassing. Reviewer #1 also raised a similar concern, so we have provided more information about the behaviour of phosphorus, silicate, and organic carbon in CLIMBER-X:

*Page 5, Lines 108-113: Revised "In the open carbon cycle setup, a simplification is made to enforce that the budgets for silicate and phosphorus within the ocean—sediment system are balanced. This was done by disabling riverine fluxes of phosphorus and silicate, and assuming that organic carbon (which includes P) and opal (which includes Si) which are buried in the sediments (and, therefore, removed from the system) are instead returned to the surface ocean in remineralized form. The conservation of such inventories in the ocean—sediment system removes challenges related to nutrient conservation that would otherwise complicate the analysis and interpretation of model results."*

Lines 125–126: I do not understand why "carbonate sedimentary rock" should be different than "carbonate", in term of weathering (Eq. 2). Moreover, why not indicating the equations for "carbonate sedimentary rocks" weathering and loess weathering?

Thanks for your comment; we hope that we can clarify this. In Hartmann & Moosdorf (2012), there are three carbonate-rich sedimentary lithologies, which are mixed sedimentary rocks (sm), evaporites (ev), and carbonate sedimentary rocks (sc). The evaporites class (ev) is used only to compute phosphorus fluxes, and is therefore not considered here.
However, other lithologies still maintain information on carbonate content, such as unconsolidated sediment (su) and metamorphics (mt). When we specify "carbonate sedimentary rocks", we mean that that the contribution of the lithology "sc" to carbonate weathering rates in a grid cell is not calculated using an Arrhenius equation. In addition to the 13 rock lithologies as listed in Table A2 in Hartmann & Moosdorf (2012), we also consider loess (lo) as another lithology. The contribution of "lo" to carbonate weathering rates in a grid cell is similarly not calculated using an Arrhenius equation. We agree it would be useful to show the equations that are used for the other two lithologies (loess and carbonate sedimentary rock), which is why we have incorporated them into the revised manuscript (***Page 6, Line 126-127*** in Equation 2).
The "16 lithologies", which was erroneous, was fixed in the revised manuscript to 13. We have expanded on the weathering scheme by including a table of lithological values and a figure showing the fraction of a given lithology in a grid cell (***Page 39-40***, Fig. E1 and Table E1).

Lines 132–136: I think that orbital forcings could be mentioned here, among the "external forces" (line 132) excluded in the study, although it may be redundant with line 145.
We had a similar thought, and deliberated which section would be most appropriate for this information. Ultimately, we decided to not include this information here, as it is indeed redundant with ***Page 7, Line 156-159***. Furthermore, we would have to modify the last part of the highlighted sentence (as orbital forcing is both predictable and relevant for long timescales).

Lines 145–146: It is not completely clear here whether the fixed orbital forcings concern only the spin-up run, or all simulations (including the spin-up).
On ***Page 7, Lines 158-159*** we state that "All simulations run for 100,000 years with constant orbital parameters and without any climate acceleration technique". However, we have moved what was previously Lines 145-146 in the original manuscript to ***Page 7, Lines 156-158*** to highlight that orbital parameters are constant in all simulations:

***Page 7, Lines 156-159****: Revised "To simplify interpretation and ensure consistency with previous studies, orbital forcing was fixed at present-day values, with the combined effects of anthropogenic and orbital forcing to be explored in a future study. All simulations run for 100,000 years with constant orbital parameters and without any climate acceleration technique."*

Lines 169–170: I think it would be useful here just to indicate that climate sensitivity is altered by rescaling the pCO2 seen by the radiative code as a function of the actual pCO2, and then refer to Appendix A.
Thanks for your suggestion. We have changed this part to the following:

***Page 8, Lines 176-178****: Revised "The sensitivity of the results to different equilibrium climate sensitivities (ECS) between 2-4°C was additionally investigated (ECS2, ECS4) by rescaling the equivalent $CO_2$ in the long-wave radiation scheme (further described in Appendix A)."*

Lines 185–186: This statement, "temperatures temporarily stabilize instead of decreasing due to the release of soil carbon into the atmosphere" seems erroneous. Temperature does stabilize during between 150yr and 1000yr in the 5000 PgC scenario (Fig. 2b), but pCO2 declines just as in the other scenarios (Fig. 2a). So how could it be an effect of the "release of soil carbon into the atmosphere"? It rather seems that there is a decoupling of CO2 and temperature, that is likely due to oceanic dynamics. Indeed, there is a small bump of global temperature at 700yr (without any pCO2 change), which coincides with abrupt AMOC recovery (Fig. 7e).

Many thanks for pointing this out. This is indeed an erroneous statement. Upon reviewing the data, we agree that the temperature stabilization in the 5000 PgC scenario within the first millennium is not explained by the release of soil carbon into the atmosphere. It does appear that the likely cause is oceanic dynamics and AMOC, as pointed out. The extended decline in AMOC results in a cooling in the Northern Hemisphere which prevents global mean temperature (GMT) from rising after year ~150. After some time, this cooling is offset by Southern Hemisphere warming via the bipolar seesaw, and explains why GMT stabilizes during the better part of the first millennium. This behaviour continues until the abrupt AMOC recovery, which triggers a rapid increase in GMT (and there is indeed the small bump in global temperature at this time). The role of AMOC on temperature, rather than $CO_2$ (radiative forcing, $log(CO_2)$) is demonstrated in Fig. A1 of the "Additional material" section at the end of this document. We have revised this statement accordingly in the updated manuscript:

*Page 8, Lines 193-195: Revised "This is largely due to the extended decline in the Atlantic Meridional Overturning Circulation (AMOC; Fig. 7e), which results in a cooling in the Northern Hemisphere that prevents global mean temperature from further rising."*

Lines 215–221: this non-monotonous behavior is interesting. Has it been already suggested, or is it a new finding of current study?

To the best of our knowledge, this has not yet been explicitly observed in a previous study on the long-term effects of anthropogenic $CO_2$. This is mostly because land carbon was often not considered (or the response unreported, as in Lenton & Britton 2006). However, we are not prepared to conclude that this is necessarily a new finding (e.g., a strong positive climate-carbon cycle feedback related to soil respiration has already been highlighted in studies such as Cox et al. (2000)). On a global level, the response of soil carbon to increasing emissions is generally dictated by (1) that which is gained from increases in primary production and litterfall, and (2) that which is lost from higher soil respiration, influenced by different competing feedbacks.

Line 222: This statement, "In our simulations, the land is a net carbon sink for the entire 100 kyr" also seems erroneous. From Fig. 3a, it appears that land becomes a (slight) net source of carbon at 200kyr in all simulations. Besides, I don't think that "land carbon" is defined anywhere in the manuscript. Is it simply "soil + vegetation" carbon?

Thanks for bringing this to our attention. Indeed this is an erroneous statement and it has been corrected as the following the revised manuscript. We have also made sure that land carbon is explicitly defined in the manuscript as the sum of vegetation and soil carbon.

*Page 11, Line 215: Added "The land carbon pool [(as the sum of vegetation and soil carbon)] is…"*
*Page 12, Lines 231-233: Revised "In our simulations, the [terrestrial storage of anthropogenic carbon is positive] during the entire 100 kyr due to…"*

Line 364: The mention of "noLAND" comes quite abruptly here, given that the sensitivity experiments are only discussed in a later section (4). Could you remind "experiment with land carbon disabled", and refer to Table 1?

Thanks for the suggestion. Reviewer #2 also raised a similar point. We have introduced the term in the experimental section of the text, remind the reader that this refers to the experiment with land carbon disabled, and include a reference to the experiment table.

*__Page 8, Line 171__: Added "...land carbon cycle response [(noLAND)]..."*
*__Page 21, Line 368-369__: Added "...and [the experiment with land carbon disabled (noLAND, Table 1)]."*

Fig 13: It is difficult to visualize the trends of Ai and τi versus cumulative emission (trends that are discussed in the current section). I suggest adding two small panels in the figure, plotting Ai vs E and τi versus E.

We have added two subplots for $A_i$ vs E and $\tau_i$ vs E in Fig. 13 (*__Page 23__*).

Lines 371–377: It might be useful to indicate here that Ai do not sum at 1 because the IRF does not start at 1, and that the initial value (= the sum of Ai) depends on the cumulative emission scenario.

Thank you for the suggestion. To some extent, this information was included in the caption of Table 2 of the original manuscript (now Table E2), but we have now integrated it into the main text to ensure its visibility in the revised manuscript:

*__Page 22, Lines 393-396__: "However, short-term processes (sub-centennial timescales) cannot be fit with our analysis after removing the ramp up period of emissions, meaning n=3 is already sufficient for examining the long-term uptake of anthropogenic $CO_2$. It should be noted that, due to this, the values for $A_i$ in the REF and noLAND experiments do not sum to 1 as a fraction of emissions was already removed via short-term processes (Fig 12b)."*

Lines 565–572: It seems that there is a positive feedback here: warmer temperature (for a same pCO2) generates higher pCO2, because of the warming-induced soil carbon release. It would be useful to indicate that it is a positive feedback.

We have added this to the revised manuscript:

*__Page 31, Lines 558-559__: Added "This behaviour is indicative of a positive feedback as a result of warming-induced soil carbon release."*

Line 594: Is methane lost by converting it into CO2? Granted that 2200 ppb of methane should not generates more than 2 ppm of CO2, with is much less than the pCO2 anomaly reported in Fig. 15d.

Firstly, we would like to clarify that 2200 ppb is peak $CH_4$ concentration in the 5000 PgC scenario, whereas the sensitivity analysis in Section 4 is limited to 3000 PgC and less. This, of course, does not answer the reviewer's question, as peak $CH_4$ of 1600 in the 3000 PgC scenario alone cannot explain an additional 25 ppm of atmospheric $CO_2$.

Methane is oxidized assuming a constant lifetime of 9.5 years (Willeit et al. 2023). In reality, this results in $CH_4$ being converted into $CO_2$ in the atmosphere, but this flux is small. However, for simplicity (and for carbon conservation), we add carbon from methane to surface $CO_2$ flux in CLIMBER-X (e.g., soil $CO_2$ emission).

The reason why $CO_2$ is higher in the intCH4 experiments is because, as mentioned in the previous comment, $CH_4$ causes an additional positive climate-carbon cycle feedback (as

additional warming enhances soil respiration; see Fig. A2-A4 in the "Additional material" section at the end of this document).
We originally had a larger discussion on this, but it was cut in our efforts to (already) shorten the paper. However, Reviewer #2 had a similar question, asking how temperatures evolve in the intCH4 experiments compared to the REF experiments (as a result of this large increase in $CH_4$ concentration), so have elaborated on this more in the revised manuscript.

*Page 33, Lines 596-598: Added "Temperature differences between the intCH4 and REF experiments also tend to decrease around this time, and despite small variations in how temperature evolves, the experiments largely follow each other for the rest of the simulation (Fig. E7a)."*
*Page 46: Added Fig. E7.*

Lines 644–645: Would it really influence the ATMOSPHERIC lifetime of CO2? It seems to me that the longer weathering timescale is the just a delay because of carbon storage in land before it is stored through weathering, instead of being directly stored though weathering, and that this sink transfer has no consequence regarding carbon in the atmosphere.
You make a good point, and it is true that the longer (effective) weathering timescale is caused by the temporary storage of carbon on land. However, the land carbon pool on its own does increase the residence time of anthropogenic $CO_2$, as the land stores about 20-40% of anthropogenic carbon (Page 11, Fig. 4) before gradually releasing it into the atmosphere. This ultimately slows down the $CO_2$ decline on long timescales (Fig. 17a).

**Technical corrections:**

There are several occurrences where it should be more accurate to talk about weathering "flux", than weathering "rate", which rather refers to a specific flux (in mol/m2/yr): line 138, line 281, caption of Fig. 9, line 291, line 301...
Thanks, we have changed the word "rate" to "flux" in the following lines of the revised manuscript: *Line 130, Line 144, Line 283, Fig. 9 caption, Line 293, Line 301, Line 525, and Line 627*.

Line 130: It seems that "run-off" should be spelled "runoff", to be consistent with the other occurrences of that word in the manuscript.
Thanks for pointing this out. It was changed to "runoff" in the revised manuscript (*Page 6, Line 136*).

Caption of Fig. 6: The mean net annual NPP is in (a–c), not (a–b).
Thanks, this has been corrected (Fig. 6, *Page 13*).

Fig. 11: A mere suggestion: it feels more "natural" to use a colorscale with "wetter" colors (e.g., blue) for precipitation increase and "dryer" colors (e.g., red) for precipitation decrease.
Thanks for the suggestion. We changed Figure 11 h-i to a brown–bluegreen colormap, which is often used to indicate "drier" and "wetter" conditions (Fig. 11, *Page 19*).

Line 526: I believe that "begin" should here be a singular, "begins".
Thanks, this was changed in the revised manuscript (*Page 27, Line 514*).

Line 638: Shouldn't "variation" be a plural here?
Thanks, this was corrected (*Page 34, Line 647*).

There are a few inconsistencies between US and British spelling. I noticed the use of "behavior" and "behaviour" in the text. Please check.

Thanks for pointing this out. The reviewer points out inconsistencies between US and British spelling (e.g., the use of "colour" but then at the same time "parametrize"). Some of these inconsistencies can be explained by the chosen variety of English, Canadian English, which is the first author's first language and is accepted by the EGU. However, we changed "behavior" to "behaviour" as suggested in text (***Page 25, Line 455***).

Many DOIs link have duplicated "https://doi.org/https://doi.org/" in the reference list. Pleas check.

Many thanks for pointing this out. This was corrected in the revised manuscript.

**Additional material:**

Fig. A1: Role of radiative forcing and AMOC on the evolution of global mean surface temperature. Trajectories have been plotted for the entire 100,000 years.

[Figure]

Fig. A2: Change in global mean surface temperature in the 0-3000 PgC emission scenarios. Colours here correspond to the cumulative emission scenarios shown in Fig. 2 of the manuscript. The response in temperature is shown here for two experiments: solid line for the intCH4 experiment and dashed lines for the REF experiment.

[Figure]

Fig. A3: Change in vegetation carbon inventory in the 0-3000 PgC emission scenarios. Colours here correspond to the cumulative emission scenarios shown in Fig. 2 of the manuscript. The response in vegetation carbon is shown here for two experiments: solid line for the intCH4 experiment and dashed lines for the REF experiment.

[Figure]

Fig. A4: Change in soil carbon inventory in the 0-3000 PgC emission scenarios. Colours here correspond to the cumulative emission scenarios shown in Fig. 2 of the manuscript. The response in soil carbon is shown here for two experiments: solid line for the intCH4 experiment and dashed lines for the REF experiment.

[Figure]

**References:**

Archer, D. & Brovkin, V. (2008). The millennial atmospheric lifetime of anthropogenic $CO_2$. *Climatic Change*, 90(3), 283–297. https://doi.org/10.1007/s10584-008-9413-1

Archer, D., Eby, M., Brovkin, V., Ridgwell, A., Cao, L., Mikolajewicz, U., Caldeira, K., Matsumoto, K., Munhoven, G., Montenegro, A. & Tokos, K. (2009). Atmospheric lifetime of fossil fuel carbon dioxide. *Annual Review of Earth and Planetary Sciences*, 37(1), 117–134. https://doi.org/10.1146/annurev.earth.031208.100206

Archer, D., Kheshgi, H. & Maier-Reimer, E. (1997). Multiple timescales for neutralization of fossil fuel $CO_2$. *Geophysical Research Letters*, 24(4), 405–408. https://doi.org/10.1029/97gl00168

Archer, D., Kheshgi, H. & Maier-Reimer, E. (1998). Dynamics of fossil fuel $CO_2$ neutralization by marine $CaCO_3$. *Global Biogeochemical Cycles*, 12(2), 259–276. https://doi.org/10.1029/98gb00744

Brault, M.-O., Matthews, H. D. & Mysak, L. A. (2017). The importance of terrestrial weathering changes in multimillennial recovery of the global carbon cycle: A two-dimensional perspective. *Earth System Dynamics*, 8(2), 455–475. https://doi.org/10.5194/esd-8-455-2017

Cox, P. M., Betts, R. A., Jones, C. D., Spall, S. A. & Totterdell, I. J. (2000). Acceleration of global warming due to carbon-cycle feedbacks in a coupled climate model. *Nature*, 408(6809), 184–187. https://doi.org/10.1038/35041539

Eby, M., Zickfeld, K., Montenegro, A., Archer, D., Meissner, K. J. & Weaver, A. J. (2009). Lifetime of anthropogenic climate change: Millennial time scales of potential $CO_2$ and surface temperature perturbations. *Journal of Climate*, 22(10), 2501–2511. https://doi.org/10.1175/2008jcli2554.1

Hartmann, J. & Moosdorf, N. (2012). The new global lithological map database GLiM: A representation of rock properties at the Earth Surface. *Geochemistry, Geophysics, Geosystems*, 13(12). https://doi.org/10.1029/2012gc004370

Hartmann, J., Moosdorf, N., Lauerwald, R., Hinderer, M. & West, A. J.: Global chemical weathering and associated P-release — The role of lithology, temperature and soil properties, *Chemical Geology*, 363, 145–163, https://doi.org/10.1016/j.chemgeo.2013.10.025, 2014.

Joos, F., Roth, R., Fuglestvedt, J. S., Peters, G. P., Enting, I. G., von Bloh, W., Brovkin, V., Burke, E. J., Eby, M., Edwards, N. R., Friedrich, T., Frölicher, T. L., Halloran, P. R., Holden, P. B., Jones, C., Kleinen, T., Mackenzie, F. T., Matsumoto, K., Meinshausen, M., … Weaver, A. J. (2013). Carbon dioxide and climate impulse response functions for the computation of Greenhouse Gas Metrics: A multi-model analysis. *Atmospheric Chemistry and Physics*, 13(5), 2793–2825. https://doi.org/10.5194/acp-13-2793-2013

Köhler, P. (2020). Anthropogenic $CO_2$ of high emission scenario compensated after 3500 years of ocean alkalinization with an annually constant dissolution of 5 Pg of Olivine. *Frontiers in Climate*, 2. https://doi.org/10.3389/fclim.2020.575744

Lenton, T. M. & Britton, C. (2006). Enhanced carbonate and silicate weathering accelerates recovery from fossil fuel $CO_2$ perturbations. *Global Biogeochemical Cycles*, 20(3). https://doi.org/10.1029/2005gb002678

Lord, N. S., Ridgwell, A., Thorne, M. C. & Lunt, D. J. (2015). An impulse response function for the "long tail" of excess atmospheric $CO_2$ in an Earth system model. *Global Biogeochemical Cycles*, 30(1), 2–17. https://doi.org/10.1002/2014gb005074

Munhoven, G. & François, L.M. (1994). Glacial-Interglacial Changes in Continental Weathering: Possible Implications for Atmospheric CO2 . In: Zahn, R., Pedersen, T.F., Kaminski, M.A., Labeyrie, L. (eds) Carbon Cycling in the Glacial Ocean: Constraints on the Ocean's Role in Global Change. NATO ASI Series, vol 17. Springer, Berlin, Heidelberg. https://doi.org/10.1007/978-3-642-78737-9_3

Munhoven, G., Brovkin, V., Ganopolski, A. & Archer, D. (2007). Impact of future Greenland deglaciation on global weathering fluxes and atmospheric $CO_2$ [Paper presentation]. 17th V. M. Goldschmidt Conference 2007, Cologne, Germany.

Ridgwell, A. & Hargreaves, J. C. (2007). Regulation of atmospheric $CO_2$ by deep-sea sediments in an Earth system model. *Global Biogeochemical Cycles*, 21(2). https://doi.org/10.1029/2006gb002764

Willeit, M., Ganopolski, A., Robinson, A. & Edwards, N. R. (2022). The Earth System Model CLIMBER-X v1.0 – Part 1: Climate model description and validation. Geoscientific Model Development, 15(14), 5905–5948. https://doi.org/10.5194/gmd-15-5905-2022

Willeit, M., Ilyina, T., Liu, B., Heinze, C., Perrette, M., Heinemann, M., Dalmonech, D., Brovkin, V., Munhoven, G., Börker, J., Hartmann, J., Romero-Mujalli, G. & Ganopolski, A. (2023). The Earth System Model CLIMBER-X v1.0 – Part 2: The global carbon cycle. *Geoscientific Model Development*, 16(12), 3501–3534. https://doi.org/10.5194/gmd-16-3501-2023

Winkelmann, R., Levermann, A., Ridgwell, A. & Caldeira, K. (2015). Combustion of available fossil fuel resources sufficient to eliminate the Antarctic Ice Sheet. *Science Advances*, 1(8). https://doi.org/10.1126/sciadv.1500589

Wunderling, N., von der Heydt, A. S., Aksenov, Y., Barker, S., Bastiaansen, R., Brovkin, V., Brunetti, M., Couplet, V., Kleinen, T., Lear, C. H., Lohmann, J., Roman-Cuesta, R. M., Sinet, S., Swingedouw, D., Winkelmann, R., Anand, P., Barichivich, J., Bathiany, S., Baudena, M., Bruun, J. T., Chiessi, C. M., Coxall, H. K., Docquier, D., Donges, J. F., Falkena, S. K. J., Klose, A. K., Obura, D., Rocha, J., Rynders, S., Steinert, N. J. & Willeit, M. (2024). Climate tipping point interactions and cascades: A Review. *Earth System Dynamics*, 15(1), 41–74. https://doi.org/10.5194/esd-15-41-2024

---

## Author Response (AR2)

**General response:**

We would like to express our gratitude to all of the reviewers for the ongoing support and insightful feedback. Their thoughtful and constructive comments have played a key role in refining both the structure and content of the manuscript. Please find our point-to-point responses to the individual comments given by the Reviewers below (reviewer comment in black, our response in blue). The ***line numbers*** and ***page numbers*** in this document refer to the revised mauscript file *with* tracked changes.

**Response to RC2:**

I thank the authors for their clear and elaborate response to my, and the other reviewers, remarks. All my concerns have been addressed. I have some short new remarks on the revision which should be easy and quick to address. Congratulations on an interesting, well written, paper.

We would like to sincerely thank the reviewer for their positive feedback and thoughtful comments on our paper. We are glad to hear that all of your concerns, as well as those of the other reviewers, have been addressed. We appreciate your additional remarks on the revision, and made sure to incorporate them. Thank you again for your time and constructive input, and for recognizing the work presented in our paper.

Line 152: REF is now introduced in line 182 but could also already be introduced in this line.
Thanks for the suggestion, we have introduced it in the following way as suggested:

***Page 7, Line 152***: Added "In the reference [(REF)] experiment, we…"

Line 195: '… towards to …' I suppose 'to' can be removed here.
Thanks for catching that, it has been changed as suggested:

***Page 10, Line 200***: Removed "…temperatures slowly decrease towards [to] pre-industrial levels…"

Line 228: I'm not sure what is meant with 'generally globally distributed'. Should I interpret this sentence as: 'Most of the globe sees an increase in vegetation carbon with increasing emissions mainly because of increases in NPP. However, there are a few regions that see a decline in vegetation carbon due to a reduction in carbon'. If so, I suggest rewriting the sentence a bit to make it clearer (feel free to use my suggestion, paraphrase it or come up with something yourself).

We have changed the text based on what was suggested, but have replaced "…due to a reduction in [carbon]" with "…due to a reduction in [precipitation]", since we believe that is what the reviewer meant here.

***Page 12, Line 233-235***: Changed "Most of the globe sees an increase in vegetation carbon with increasing emissions mainly because of increases in NPP (Fig. 6b,c,e,f). However, there are a few regions that see a decline in vegetation carbon due to a reduction in precipitation (Fig. 11h,i)."

Line 280: to is missing behind proportional, I think.

Thanks for catching that, it has been changed as suggested:

*__Page 16, Line 286__: Added "…as it is proportional [to] organic detritus production…"*

Line 339: Now an instantaneous pulse is used in the revision with the PULSE experiments, right?
Indeed, you are correct. We have revised this sentence to better reflect our suite of experiments:

*__Page 20, Line 345-346__: Added "Unlike previous studies (e.g., Archer et al., 2009a; Joos et al., 2013), we do not deploy an instantaneous pulse [in the REF ensemble], and our fractions of emissions remaining do not start at 1 as carbon sequestration already began during the ramp up of emissions."*

Line 656: 'long timescales' is mentioned several times without a clear definition of what is meant with long. I don't think that's necessarily a problem. However, I do suggest defining it in line 656 as it is a major conclusion. A suggestion on how to do it: '… on long timescales ($> 10^3$ years).' (or whatever timescale you want to put there).
This is a very good point. In addition to defining "long timescale(s)" in the line as suggested, we have also provided a definition at the beginning of the manuscript.

*__Page 2, Line 24__: Added "…on the long-term [($>10^3$ years)] future."*
*__Page 35, Line 670__: Added "The methane cycle will likely have a negligible effect on long timescales [($>10^3$ years)]."*

Generally: in most of the text the simulations corresponding to the REF experiment are referred to as one ('REF experiment'). I understand that the simulations all fall under one 'experiment', but this is a bit confusing since there are several simulations. I suggest referring in the text to 'REF experiments', so plural. I suggest also doing this for the other experiments. I think this might make it a bit clearer.
This is a good point, thanks. We have changed "experiment" to "experiment[s]", "ensemble", and "ensembles" where appropriate in the revised manuscript.

**Response to RC3:**

The authors have adequately addressed all the remarks I have raised during the first round of review. Their choice regarding which information (and figures) to add, and which paragraph to remove, in order not to excessively lengthen the manuscript, is relevant. Although there are a few more details I would recommend them to rectify, as far as I concerned, I consider that it can be achieved without needing another round of review.
We would like to sincerely thank the reviewer for their positive feedback and for recognizing the efforts made in addressing the remarks raised during the first round of review. We appreciate the understanding regarding the balance between adding relevant information and maintaining the manuscript's length. We took note of your additional recommendations and made the necessary adjustments accordingly. Thank you once again for your valuable input, and we are pleased to hear that you find that a further review is not required.

Line 165: to be fully consistent, I suggest to change here also "pulse" to "scenario".
Thanks for catching that, it has been changed as suggested:

***Page 7, Line 165**: Changed "explore the sensitivity of the results to the duration of the $CO_2$ emission [scenarios], ...".*

Lines 190-195: the cause identified by the authors for the CO2/temperature decoupling in the 500 PgC appears consistent. But the description the authors provide is somewhat awkward, and reads as "global temperature decreases at t = 300yr, except in the 5000 PgC scenario where AMOC declines, resulting in a cooling, preventing further warming".
I believe what the author meant is that the AMOC starts declining around t = 150 yr (not t = 300 yr and the resulting cooling effect truncates the warming peak otherwise observed in all other scenarios (and although CO2 keeps increasing until t = ~300 yr). Hence, there is a global temperature plateau, and global cooling is delayed to t = ~700 yr starts.
This can be explained a bit clearer than currently done.
Thanks for bringing this to our attention. After rereading this section, we agree that it can be interpreted incorrectly, so we have since changed it to the following in the revised manuscript:

***Page 8, Line 191-199**: Changed "However, the extended decline in the Atlantic Meridional Overturning Circulation (AMOC; Fig. 7e) in the 5000 PgC scenario results in a cooling effect in the Northern Hemisphere that prevents global mean temperature from further rising, thereby truncating peak warming. After emissions cease (~150–200 years after the start of the simulation, Fig E2), both atmospheric $CO_2$ concentration and global temperatures decrease as carbon is taken up by the land and ocean pools (Fig. 2a-d). The only exception to this is the 5000 PgC scenario, where there is a global temperature plateau, and global cooling is delayed until approximately ~700 years after the start of the simulation, coinciding with a substantial recovery of the AMOC (Fig. 2b, 7e)."*

The caption of Fig. 12 now gives the impression that panel a (main one) is for the whole time-series, including the ramp up emission, while panels b and c are after the peak CO2 concentration. Whereas the main text says that "The ramp up of emissions is not included in this analysis" (line 338).
Thanks, we understand how this could be interpreted incorrectly. We have since changed this to the following:

***Page 21, Fig. 12 caption**: Changed "(a) Fraction of emissions remaining in the atmosphere [after peak $CO_2$ concentration] for the different emission scenarios in the REF experiment".*

Lines 374-384, Fig 13 and Table E2 : the inclusion of PULSE experiment add some complexity to the understanding of the fit. I am surprise to see that Ai do not sum to 1 for the PULSE case (Fig. 13 and Table E2), and are actually are very similar than for the REF case. It looks as if the fit was performed starting around t = 200yr, and not at peak CO2 concentration, which, in all likelihood, must happen at t = 1yr, where the fraction of emission remaining is still ~ 1 (Fig. 12b). Or maybe is it due to a poorness of the fit for the the very first years, as suggested in caption of Table E2? Meaning that the "n=3" exponential fit cannot reproduce a curve reaching 1 at "t after peak CO2 = 0yr"?
It would be beneficial to say a word or to about why Ai does not sum to 1 in the PULSE case. In this paragraph (lines 374-384) or in the caption of Table E2. Another suggestion would be to specify which year the peak CO2 concentration in the 3 cases (REF, noLAND and PULSE).
As the reviewer correctly noted, the sum of $A_i$ in the PULSE experiment not equalling 1 is due to the relatively poor fit during the first ~100 years. We attempted to address this in the caption of Table E2 (as the reviewer also pointed out). Achieving a more accurate fit would require a

higher number of exponential decay terms to better capture "short-term" ($<10^2$ years) carbon removal processes such as air–sea $CO_2$ exchange, dissolution, and land carbon dynamics, as suggested by Lord et al. (2015) and Jeltsch-Thömmes & Joos (2020). By using n=3 for our fit, it does appear as if we have started fitting around t ≈ 200 years. However, we do fit the entire time series, starting from t=0 (time after peak $CO_2$ concentration), as one can see in Fig. A1 in the "Additional materials" section at the end of this document. This is also reflected (to a very small extent) in the lower $R^2$-values as compared to the REF/noLAND experiments, as shown in Table E2 of the revised manuscript.

In our preliminary tests on the PULSE experiment, we found that n=4 to 5 provided the best fit depending of the magnitude of $CO_2$ emissions, consistent with the findings of Colbourn et al. (2015) and Lord et al. (2015). However, we kept n=3 because (1) ultimately, we are focused on the long-term carbon cycle, and (2) we wanted to remain consistent in our analysis across the different experiments. In saying this, we have clarified in the text that achieving a better fit would require a higher number of exponentials, while also acknowledging the drawbacks of this approach. For the record (since the response is public and this information will be accessible to others), we have included an example of fitting the 500 and 5000 PgC scenarios with a higher-order exponential fit as Fig. A2 the "Additional Materials" section.

***Page 22-24, Lines 399-406***: Changed "Short-term processes (i.e., sub-centennial timescales) cannot be fit for the REF and noLAND experiments given potential complications with and the removal of the ramp-up period. However, n=3 is already sufficient for examining the long-term uptake of anthropogenic $CO_2$. It should be noted that, due to this decision, the values for $A_i$ in the REF, PULSE, and noLAND experiments do not sum to 1 (Fig 12b). This is because, for the REF and noLAND experiments, a fraction of emissions was already removed via short-term processes during the excluded ramp-up period. A more accurate fit could be achieved for the PULSE experiment by increasing the number of fitted exponentials, however, we kept n=3 as to remain consistent in our analysis.".

**Additional material:**

Fig. A1: Multi-exponential fit (n=3) for atmospheric $CO_2$ concentration in the PULSE ensemble, as shown in the revised manuscript. Colours here correspond to cumulative emission scenarios of 500-5000 PgC (as shown in Fig. E1 of the revised manuscript).

[Figure]

Fig. A2: Multi-exponential fit when using a higher order (n=4 to 5) of fitted exponentials for the PULSE experiment. Colours here correspond to cumulative emission scenarios of 500-5000 PgC (as shown in Fig. E1 of the revised manuscript).

[Figure]

**References:**

Colbourn, G., Ridgwell, A., & Lenton, T. M. (2015). The time scale of the silicate weathering negative feedback on atmospheric $CO_2$. *Global Biogeochemical Cycles*, 29(5), 583–596. https://doi.org/10.1002/2014gb005054

Jeltsch-Thömmes, A & Joos, F. (2020). Modeling the evolution of pulse-like perturbations in atmospheric carbon and carbon isotopes: the role of weathering–sedimentation imbalances. *Climate of the Past*, 16, 423–451. https://doi.org/10.5194/cp-16-423-2020

Lord, N. S., Ridgwell, A., Thorne, M. C. & Lunt, D. J. (2015). An impulse response function for the "long tail" of excess atmospheric $CO_2$ in an Earth system model. *Global Biogeochemical Cycles*, 30(1), 2–17. https://doi.org/10.1002/2014gb005074

---

## Author Response (AR3)

**General response:**

Dear Olivier Sulpis,

Thank you for overseeing the review of our manuscript. We are pleased that it has been accepted for publication as is, and look forward to its publication.

Best regards,

Christine Kaufhold, on behalf of all coauthors